# Deep Learning with Plausible Deniability

**Wenxuan Bao**[1]* **Shan Jin**[2] **Hadi Abdullah**[2] **Anderson C. A. Nascimento**[2]
**Vincent Bindschaedler**[1] **Yiwei Cai**[2]
[1]University of Florida      [2]Visa Research

## Abstract

Deep learning models are vulnerable to privacy attacks due to their tendency to memorize individual training examples. Theoretically-sound defenses such as differential privacy can defend against this threat, but model performance often suffers. Empirical defenses may thwart existing attacks while maintaining model performance but do not offer any robust theoretical guarantees.

In this paper, we explore a new strategy based on the concept of plausible deniability. We introduce a training algorithm called **P**lausibly **D**eniable **S**tochastic **G**radient **D**escent (PD-SGD). The core of this approach is a rejection sampling technique, which probabilistically prevents updating model parameters whenever a mini-batch cannot be plausibly denied. We provide theoretical results showing that PD-SGD effectively mitigates privacy leakage from individual data points. Experiments demonstrate the scalability of PD-SGD and the favorable privacy-utility trade-off it offers compared to existing defense methods.

## 1 Introduction

Deep learning models are susceptible to privacy attacks such as membership inference [38, 45, 7, 4, 50, 14, 22, 41] that compromise the confidentiality of training data. Although mitigation strategies exist, the current state of affairs forces practitioners to choose between strong privacy guarantees and high-quality performant models. Differential privacy (DP)-based approaches [16] such as DP-SGD [1] offer strong mathematical privacy guarantees but often substantially degrade model quality. Empirical defenses [29, 40, 9] better preserve model quality but come without any mathematical guarantees and thus may ultimately prove vulnerable to future (yet-to-be-discovered) attacks.

This paper introduces a new approach for model training inspired by the concept of *plausible deniability*. The central privacy goal is to ensure that the trained model could be obtained from different data instead (and therefore model developers can plausibly deny using specific data subsets). To satisfy this privacy desideratum, we propose Plausibly Deniable Stochastic Gradient Descent (PD-SGD), a new gradient-based learning algorithm that leverages an efficient *privacy test* in each learning iteration that scrutinizes mini-batch gradients before they are used to update model parameters. The test discards anomalous gradients — that are not consistent with the desired deniability level — thereby mitigating potential adverse privacy consequences from such updates (as illustrated in Fig. 2).

We formalize batch-level plausible deniability as a privacy game focused on a single learning iteration. We prove that achieving batch PD also protects individual examples' privacy in the sense of membership inference. With proper instantiation of the privacy test, PD-SGD satisfies batch PD and $(\varepsilon, \delta)$-differential privacy. Through composition, we obtain a guarantee for the entire training process.

In summary, we provide a new way to conceptualize privacy for model training through plausible deniability and a new algorithm for it that uses rejection-sampling based privacy testing. Compared to DP-SGD, our approach does not require per-example clipping (which reduces computational and

---

*Work partially done during internship at Visa Research

39th Conference on Neural Information Processing Systems (NeurIPS 2025).

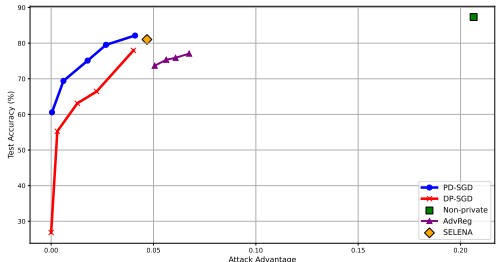

Figure 1: **Privacy–Utility Trade-off** for different methods: We train WRN-16-4 on CIFAR-10 from scratch with different defense methods. PD-SGD offers a **better** trade-off than all other defenses. Attack Advantage is $2 \times$ (Balanced Attack Accuracy $- 0.5$).

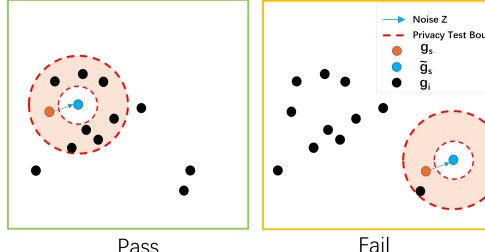

Figure 2: **Privacy Test Illustration.** Gradients are shown in 2-D. *Pass:* noisy seed gradient $\tilde{g}_s$ (blue) has $T$ alternatives/neighbors $g_i$ (black) inside privacy-test bound (red dashed circles). *Fail:* Too few gradients fall inside the region, so the update is rejected.

memory overhead) and it is applicable to any loss function (even non-decomposable ones). Compared to empirical defenses (i.e., Adversarial Regularization [29], SELENA [40], and HAMP [9]), our approach not only offers a theoretical guarantee but also provides a superior privacy-utility tradeoff. In experiments, we find that models trained with PD-SGD often match or exceed the test accuracy of empirical defenses while providing lower attack success rates against state-of-the-art membership inference attacks [45, 7]. We also show that PD-SGD scales to large models by fine-tuning LLaMA-2-7B [44] on SST-2 [39].

## 2 Background & Related works

**Deep learning.** We consider a supervised model represented by a function $f_\theta$ where $\theta$ denotes the model's weights/trainable parameters. The model is trained using a dataset $D$ of $n$ data points $(x_i, y_i)$, $i \in [1, n]$ and solving for the vector $\theta$ that minimizes the loss function $\mathcal{L}(\cdot)$ on $D$. This is typically done using Stochastic Gradient Descent (SGD) [20] or a variant [42]. We focus on mini-batch SGD, which we refer to as (vanilla) SGD. In each iteration, the algorithm partitions the training set into (roughly) equal-sized mini-batches, randomly picks a mini-batch, and updates the parameters according to the mini-batch's gradient. Specifically given a mini batch $B_j$, we let $g_j = \nabla_\theta \mathcal{L}(\theta, B_j) \in \mathbb{R}^d$ denote the gradient of the loss on $B_j$ with respect to the model parameters $\theta \in \mathbb{R}^d$. The update at step $k$ is therefore: $\theta_k = \theta_{k-1} - \eta g_j$, where $\eta$ is the learning rate.

**Membership inference attacks (MIAs).** MIAs have been extensively studied in recent years [38, 35, 46, 33, 27, 10, 7, 45, 28, 4, 50, 14]. These are privacy attacks where the adversary aims to determine if a target example $(x, y)$ was included in the model's training set. Specifically, the adversary seeks to discern between two competing hypotheses: $H_0$ ("non-member" or "out"): $(x, y) \notin D$, or $H_1$ ("member" or "in"): $(x, y) \in D$.

Membership inference attacks were first introduced by Shokri et al. [38], employing shadow models trained on data similar to the target's to emulate its behavior and generate attack data. Recent works like Ye et al. [45] propose different attack variants aimed at reducing adversarial uncertainty to improve attack effectiveness. Carlini et al. [7] propose a Likelihood Ratio Attack (LiRA) and advocate for increasing true positive rates at low false positive rates.

**Defenses with a provable guarantee.** Differentially Private Stochastic Gradient Descent (DP-SGD—Abadi et al. [1]) provably satisfies differential privacy [17]. DP-SGD updates the model parameters iteratively like SGD, except that it bounds privacy leakage through (1) per-example clipping and (2) noise addition. Each mini-batch gradient is computed as the average over the batch's per-example gradients, but the *per-example gradients* are first clipped to have bounded $l_2$-norm. This ensures that each example has a bounded influence on the mini-batch gradient that decreases with the size of the mini-batch. Further, the mini-batch gradient is noised with isotropic Gaussian noise before being used to update the parameters.

Given a clipping threshold $C > 0$, the noisy gradient is:

$$\bar{g}_j = \frac{1}{b}\left[\sum_i g_{j,i}\,\min(1, \frac{C}{||g_{j,i}||}) + \mathcal{N}(0, \sigma^2 C^2 I)\right]\,,$$

where $b$ is the number of examples in the mini-batch, $g_{j,i}$ is the gradient vector of example $i$ in batch $B_j$, and $\sigma$ is the noise level.

The prediction accuracy of models trained this way often suffers significantly due to the impact of the noise [13] and gradient clipping [8, 31, 24]. Careful tuning of hyperparameters, and (or) use of techniques such as data augmentation [12, 3] is critical to obtain favorable utility, especially when the amount of training (or fine-tuning) data is limited [43]. Another drawback is increased training time, and larger memory requirements, although recent research attempts to mitigate these issues [6].

**Empirical defenses.** To address the problem of low utility while still effectively thwarting membership inference, several empirical defense mechanisms have been proposed. These include Adversarial Regularization (AdvReg) [29], SELENA [40] and HAMP [9]. We select AdvReg, SELENA, and HAMP because they are well-known and widely used as baselines [40, 2]. These defense mechanisms are applied during the training phase, like DP-SGD.[2]

These approaches typically employ strategies such as regularization to lower the attack score, or applying knowledge distillation to mitigate the attacks. While these empirical defense mechanisms can preserve the model's utility and offer some level of privacy protection, they lack provable guarantees. Consequently, it is unclear to what extent they truly eliminate sensitive information leakage or the degree to which they will be effective against future (possibly adaptive) attacks.

To the best of our knowledge, no existing defense mechanism simultaneously offers a theoretically justified guarantee and maintains good model utility. Our proposed method, PD-SGD, is designed to help bridge this gap.

**Plausible deniability.** It is often said that differential privacy provides plausible deniability. Differential privacy ensures that the probabilities of any output on neighboring datasets (datasets that differ in exactly one example) are tightly bounded in terms of the privacy budget $\varepsilon$. Thus, in a sense, one can plausibly deny the membership of the differing example.

There are existing attempts at formalizing plausible deniability notions for machine learning such as [32, 5]. Rass et al. [32] point out that since the same supervised model can be obtained from multiple datasets (including purely random ones), then one can deny the dataset used. Bindschaedler et al. [5] focus on the problem of synthesizing tabular microdata where a synthetic row is produced from a single row of a database as a "seed." The authors propose that a synthetic row is *plausibly deniable* if the original database contains more than $T$ (integer parameter) alternative rows that could generate the synthetic with similar probability.

## 3 Plausible Deniability for Deep Learning

### 3.1 Batch-Level Plausible Deniability

In an epoch of a mini-batch learning algorithm such as SGD, the dataset is first randomly shuffled and partitioned into $m$ mini-batches of roughly equal size. Then, $m$ iterations are performed and in each: (1) a mini-batch is selected, (2) the gradient of the loss function (with respect to the parameters) is computed on this mini-batch, and (3) this gradient is used to update the parameters. After the last iteration, the epoch ends.

Let $\mathcal{T}$ denote a *one iteration* learning algorithm that represents steps (1) and (2) as described above. The algorithm takes as input a sequence of batches $\mathfrak{B} = (B_1, \ldots, B_m)$ and the current model parameter vector $\theta$, and it outputs a gradient vector $g$. Suppose we invoke this algorithm on batches $\mathfrak{B}$, it selects batch $B$ (such that $B = B_i$ for some $i \in [1, m]$), and outputs gradient vector $g = \text{grad}(B_i; \theta)$, which is a function of $B_i$ and $\theta$.

---

[2]There are inference phase defenses such as MemGuard [23]. We do not consider them, since PD-SGD is a training algorithm.

```
Game G_0 — Batch PD
1: (B_1, B_2, ..., B_m), B_t ← A'(T_θ)    ▷ adversary pick
   batches and the target batch
2: b ~ {0,1}                              ▷ sample random bit b
3: if b = 1 then
4:    g ← T_θ(B_1, B_2, ..., B_m, B_t)    ▷ gradient from T
      with B_t included
5: else
6:    g ← T_θ(B_1, B_2, ..., B_m)         ▷ gradient from T
      without B_t
7: end if
8: b' ← A(g, T_θ, B_1, B_2, ..., B_m, B_t)
```

```
Game G_1 — Singleton-Batch PD
1: (B_1, B_2, ..., B_m), z ← A'(T_θ)      ▷ adversary pick
   batches and target data point z
2: B_t = {z}                              ▷ batch with only z in it
3: b ~ {0,1}                              ▷ sample random bit b
4: if b = 1 then
5:    g ← T_θ(B_1, B_2, ..., B_m, B_t)    ▷ gradient from T
      with B_t included
6: else
7:    g ← T_θ(B_1, B_2, ..., B_m)         ▷ gradient from T
      without B_t
8: end if
9: b' ← A(g, T_θ, B_1, B_2, ..., B_m, B_t)
```

```
Game G_2 — Chosen Data, Random Batches (≈ Strong MI)
1: S, z ← A'(T_θ)  ▷ adversary pick dataset S (|S| = n) and
   target data point z
2: (B_1, B_2, ..., B_m) ← Partition(S)    ▷ randomly shuffle
   and partition
3: B_t = {z}                              ▷ batch with only z in it
4: b ~ {0,1}                              ▷ sample random bit b
5: if b = 1 then
6:    g ← T_θ(B_1, B_2, ..., B_m, B_t)
7: else
8:    g ← T_θ(B_1, B_2, ..., B_m)
9: end if
10: b' ← A(g, T_θ, S, B_t)
```

```
Game G_3 — Average Membership Inference
1: S ~ D^n              ▷ sample n data points i.i.d. from the data
   distribution
2: z ← A'(T_θ)          ▷ adversary picks target data point z
3: (B_1, B_2, ..., B_m) ← Partition(S)    ▷ randomly shuffle
   and partition
4: B_t = {z}                              ▷ batch with only z in it
5: b ~ {0,1}
6: if b = 1 then
7:    g ← T_θ(B_1, B_2, ..., B_m, B_t)
8: else
9:    g ← T_θ(B_1, B_2, ..., B_m)
10: end if
11: b' ← A(g, T_θ, B_t)
```

Figure 3: One step privacy games relating batch plausible deniability to average membership inference. The goal of the adversary $(A, A')$ is to guess bit $b$. We have that $\mathrm{adv}_{G_i} \geq \mathrm{adv}_{G_{i+1}}$ for $i = 0, 1, 2$.

Intuitively, we can plausibly deny that batch $B$ was used (or was even a batch available to be selected) if the same (or similar) gradient $g$ could have been obtained from a different (disjoint) mini-batch $B' \neq B$. In other words, if there exists $B' \neq B$ among the set of batches such that $g = \mathrm{grad}(B'; \theta)$. If the gradient is computed deterministically, it would be extremely unlikely that two different batches $B, B'$ have the exact same gradient. But if the process is randomized (e.g., if a small amount of noise is added to the gradient) then observing a gradient $g$ that could have been produced by multiple batches does not reveal which batch was used (or if all were even available in the training data).

We formalize this privacy desideratum using a game-based definition (in the style used in prior works [34, 46, 45]) where the adversary's behavior is captured by two algorithms $A$ and $A'$. Specifically, $A'$, denotes the procedure by which the adversary chooses the sequence of $m > 1$ (disjoint) batches $(B_1, B_2, \ldots, B_m)$ and a target batch $B_t$, whereas $A$ denotes the procedure by which the adversary guesses whether $B_t$ is included.

The batch PD game $(G_0)$ is shown in the top left of Fig. 3. The adversary determines the set of batches $\mathfrak{B} = (B_1, \ldots, B_m)$ and the target batch $B_t$. Then, a random bit $b$ is sampled. If $b = 1$ then the learning algorithm $T$ is given the adversarially chosen of batches *and* the target batch. Otherwise, it is given only $\mathfrak{B}$ but not the target batch. In either case, the algorithm produces some output $g$ that is then provided to the adversary procedure $A$. The adversary guesses bit $b'$ and wins the game if $b' = b$. Intuitively, if no adversary can win at this game with higher probability than random chance, then worlds in which $b = 1$ (and the target batch could be utilized) and in which $b = 0$ (and the target batch is not available) are indistinguishable. Said differently, the model developer (who runs $T$) can plausibly deny inclusion/exclusion of *any* batch.

The advantage of adversary $(A, A')$ for a game $G$ is $\mathrm{adv}_G(A, A') = 2\Pr\{b' = b\} - 1$ and we omit $A, A'$ when clear from the context. The probability is over the randomness in the learning algorithm $T$ (and the choice of bit $b$). We denote the advantage of the best adversary for a game $G$ as $\mathrm{adv}_G$. Informally, we say that a learning algorithm $T$ provides *(batch-level) plausible deniability* if the adversary's advantage at the batch PD game is bounded. Crucially, batch PD is a property of the algorithm. It does *not* depend on the data and must hold regardless of the set of batches (which the adversary is allowed to choose).

## 3.2 Relationship to Membership Inference

Batch PD implies protection of individual examples in the sense of membership privacy. The intuition is that protecting "membership" of a batch among the available batches also protects membership of individual examples within that batch.

To show this, we construct a sequence of games $G_0$ (batch PD), $G_1$, $G_2$, $G_3$ (average MIA) where the advantage of the best adversary for $G_i$ is at least as large as that of $G_{i+1}$. From this, we conclude that batch PD implies resilience to membership inference attacks. These games are shown in Fig. 3 and we discuss their construction and relationship in full detail in Appendix B.

## 3.3 Indistinguishability & PD Criterion

Let $p_{\mathcal{T}}(\cdot)$ denote the probability distribution over the output (gradient) from algorithm $\mathcal{T}$. With this notation, $\Pr_{\mathcal{T}_\theta}(g|(B_1,\ldots,B_m))$ is the probability that $g$ is produced when the input of $\mathcal{T}$ consists of the batches $\mathfrak{B} = (B_1,\ldots,B_m)$ and the parameters are $\theta$. For conciseness, we will omit $T_\theta$ since we will only compare probabilities in the cases where $\mathcal{T}$ and $\theta$ are fixed (and clear from the context). So we write $\Pr(g|\mathfrak{B})$. We define plausible deniability of batches by the "closeness" of the distributions $\Pr(\cdot|\mathfrak{B})$ and $\Pr(\cdot|\mathfrak{B}')$ for two sets of batches $\mathfrak{B}$ and $\mathfrak{B}'$ that differ in exactly one batch (e.g., $\mathfrak{B} = (B_1,\ldots,B_m)$ and $\mathfrak{B}'(B_1,\ldots,B_m,B')$ for some batch $B'$).

For distributions $p$ and $q$ over the same domain $X$, we write $p \simeq_{\lambda,\lambda'} q$ to denote that $p$ and $q$ are $(\lambda, \lambda')$-*indistinguishable*. Distributions $p$ and $q$ are $(\lambda, \lambda')$-indistinguishable for $\lambda \geq 1$ and $0 \leq \lambda' \leq 1$ iff for all $x \in X$ and $S \subseteq X$ we have:

$$p(x) \leq \lambda q(x) + \lambda'(x) \quad \text{and} \quad q(x) \leq \lambda p(x) + \lambda'(x) \quad \text{with} \int_S \lambda'(x) \leq \lambda' . \tag{1}$$

Eq. (1) is a pointwise condition that relates the two distributions by a multiplicative factor $\lambda$ and some slack $\lambda'(x) \geq 0$ but the slack has total mass at most $\lambda'$ over the domain.

**Definition 1.** *A (one step) learning algorithm $\mathcal{T}$ satisfies the* PD criterion *if for any two sets of batches $\mathfrak{B}$ and $\mathfrak{B}'$ that differ in exactly one batch, we have $\Pr_{\mathcal{T}}(\cdot|\mathfrak{B}) \simeq_{\lambda,\lambda'} \Pr_{\mathcal{T}}(\cdot|\mathfrak{B}')$ for some $\lambda \geq 1$ and $0 \leq \lambda' < 1$.*

Definition 1 is a sufficient (but not necessary) condition to get an advantage bound on batch PD, i.e.: on $\mathrm{adv}_{G_0} \leq \frac{\lambda-1}{\lambda+1} + \frac{\lambda'}{\lambda+1}$. We provide more details in Appendix B.

**Further theory.** In Appendix C.3 we discuss the relationship between differential privacy and plausible deniability. In brief, the notions have similarities but are not directly compatible as they operate at different levels. (DP operates on entire datasets and reasons about adding/removing data points; PD operates on sets of batches and reasons about adding/removing batches.) However, we show later in the paper that the PD-SGD algorithm satisfies $(\varepsilon, \delta)$-DP, although the guarantee is looser than for DP-SGD (we leave improving bounds for future work).

## 4 Plausibly Deniable Stochastic Gradient Descent

We propose a simple modification to SGD to achieve batch PD. The idea is to combine (1) randomization of the gradient through noise addition with (2) a privacy test that enables discarding (noisy) gradients that are not *plausibly deniable* given the available set of batches. When gradients are not plausibly deniable, they are simply discarded. Otherwise, they are used to update model parameters as usual. Intuitively, this process is a way to smooth out the output distribution over the gradient to make it largely insensitive to the availability of any given batch.

### 4.1 Randomizing Gradients with Gaussian Noise

Adding noise to the gradient in SGD is a well-known technique that has benefits for convergence [30, 52]. In our case, we add isotropic Gaussian noise to the gradient vector $g$ as $\tilde{g} = g + Z$, where $Z \sim \mathcal{N}(0, \sigma^2 I)$. We can now view each (noisy) mini-batch gradient $\tilde{g}$ as a random variable and the probability of producing a fixed gradient vector $\tilde{g}$ from a batch $B$ with gradient $g = \mathrm{grad}(B; \theta)$ is denoted $p(\tilde{g}|B) = p(\tilde{g} - g)$ where $p$ denotes the Gaussian pdf for $Z \sim \mathcal{N}(0, \sigma^2 I)$.

For a fixed $\tilde{g}$, we say that two batches $B_1$, $B_2$ are $\alpha$-*similar* iff

$$\frac{1}{\alpha} \leq \frac{p(\tilde{g}|B_1)}{p(\tilde{g}|B_2)} \leq \alpha \,, \tag{2}$$

where $\alpha \geq 1$ is a privacy parameter. We write $B_1 \simeq_\alpha B_2$ to denote this (and omit $\tilde{g}$ when clear from the context). Informally, if $B_1 \simeq_\alpha B_2$, then adversaries observing $\tilde{g}$ cannot establish that $\tilde{g}$ is more likely to have originated from $B_1$ than from $B_2$ (other than allowed by $\alpha$).

The larger the number of distinct $\alpha$-similar batches, the harder it is for the adversary to link observing $\tilde{g}$ to a specific batch. The idea of privacy testing is to count the number of $\alpha$-similar batches and compare the count to a threshold $T$. We call such batches "alternatives" since if some batch $B$ originated $\tilde{g}$ then these $\alpha$-similar batches are alternative explanations (that do not involve $B$) for the adversary observing $\tilde{g}$.

## 4.2 Privacy Testing

We construct several privacy tests by considering alternative ways of counting $\alpha$-similar batches and randomizing the probability that the test passes given a specific count. In Appendix C.1 we discuss test variants and their properties in detail.

The privacy test acts as a local Lipschitz condition that bounds the change due to any one batch. There are important properties of a privacy test that influence the achieved level of plausible deniability: (1) stability, i.e., sensitivity of the count to adding/removing/substituting a batch; and (2) bounded increase in passing probability for increasing counts.

In addition, tests that have a ceiling (i.e., a maximum probability of passing the test no matter how large the count of alternatives) are desirable. This is because they limit the information leakage from gradient rejections (failing to pass the test). However, ceilings may not be needed in practical scenarios as realistic adversaries may not observe rejections.

Given a partition of the dataset into batches $\mathfrak{B} = (B_1, B_2, \ldots, B_m)$. We pick a seed batch $B \in \mathfrak{B}$ randomly and produce a noisy gradient $\tilde{g}$. Let $\tau = \tau(g, B, \mathfrak{B})$ denote the count of alternatives. The privacy test takes as input the noisy gradient $\tilde{g}$, the gradient $g$ from chosen "seed" batch $B$, and the set of batch gradients $G = (g_1, \ldots, g_m)$ (where $g_i = \mathrm{grad}(B_i; \theta)$). The privacy parameters are: $\alpha \geq 1$ (similarity factor), $T \geq 1$ (count threshold), $\beta \geq 1$ (count noise scale), and $\psi \geq 0$ (floor/ceiling).

PrivacyTest$(\tilde{g}, g, G; \alpha, T, \beta, \psi)$:

      1. $\tau \leftarrow$ CountSimilar$(\tilde{g}, g, G; \alpha)$
      2. Sample $c \sim \mathrm{Geom}(\beta)$
      3. If $\tau + c \geq T$: PASS with probability $1 - \psi$
      4. Else: FAIL

The test first computes the number of alternatives $\tau$, then it randomizes the threshold by adding an integer from the symmetric Geometric distribution with shape $\beta$. If the count matches or exceeds the threshold, then the ceiling probability $1 - \psi$ is applied and the test passes or fails accordingly. If the count is below the threshold, then the test fails. For $\beta \to \infty$ and $\psi = 0$ we recover a deterministic test that passes or fails based on whether the number of alternatives is at least $T$.

We propose three ways to count alternatives (CountSimilar). Let $B_1, \ldots, B_m$ be disjoint batches and let $B_s$ be the chosen "seed" batch.

- **Simple counting**: $\tau$ is the number of distinct batches that are $\alpha$-similar to $B_s$ (Eq. (2)).
- **Integer ("bins") counting**: $\tau$ is the number of distinct batches $B'$ such that: $\lfloor \log_\alpha(p(\tilde{g}|B_s)) \rfloor = \lfloor \log_\alpha(p(\tilde{g}|B')) \rfloor$.
- **Clique counting**: $\tau$ is the largest $k$ such that there exists distinct batches $B_1, B_2, \ldots, B_k$ that include $B_s$, where for any pair $i, j \in 1, 2, \ldots, k$: $B_i \simeq_\alpha B_j$ (Eq. (2)).

In experiments, we find that all three variants perform similarly (up to slight tuning of privacy parameters). The simple counting method performs well in experiments (which is why we often use it). However, the other two counting methods have better theoretical properties: when adding or removing a batch, the count for any noisy gradient $\tilde{g}$ (originating from any batch) can change by at most 1 (i.e., sensitivity is 1).

---

**Algorithm 1** Plausibly Deniable Stochastic Gradient Descent (PD-SGD)

---

**Input:** Training dataset $D$, number of batches $m$, number of training steps $K$, loss function $\mathcal{L}(\cdot)$, privacy parameters $(\alpha, T, \beta, \psi)$.
**Initialize:** $\theta_0$ randomly
**for** $i = 1, 2, \ldots, K$ steps **do**
    **Randomly split** $D$ **into** $\{B_1, \ldots, B_m\}$
    **Pick seed batch** $B_s$ **uniformly at random**
    $g_s \leftarrow \nabla_\theta \mathcal{L}(\theta_{i-1}, B_s)$    `// Compute gradient on seed batch`
    $\tilde{g} \leftarrow g_s + Z$ where $Z \sim \mathcal{N}(0, \sigma^2 I)$    `// Compute noisy gradient`
    **Privacy testing and parameter updates**
    $G \leftarrow \emptyset$
    **for** $j \in [1, m]$ **do**
        $g_j \leftarrow \nabla_\theta \mathcal{L}(\theta_{i-1}, B_j)$    `// Compute gradient on batch` $B_j$
        $G \leftarrow G \cup \{g_j\}$
    **end for**
    **if** $\text{PrivacyTest}(\tilde{g}, g, G; \alpha, T, \beta, \psi)$ **then**
        $\theta_i \leftarrow \theta_{i-1} - \eta\,\tilde{g}$
    **else**
        $\theta_i \leftarrow \theta_{i-1}$
    **end if**
**end for**

---

Evaluating the test is efficient (for all ways of counting). Let $\alpha = \exp(\gamma)$ for some $\gamma > 0$ (we can think of $\gamma$ as the privacy parameter instead of $\alpha$).

Take the log of the pdf. Checking for $\alpha$-similarity (Eq. (2)) is equivalent to checking if:

$$|\text{logpdf}(Z) - \text{logpdf}(\tilde{g} - g_i)| \leq \gamma\,, \tag{3}$$

which is easily testable for all batches' gradients $g_i$ for $i = 1, 2, \ldots, m$ since the log-pdf of isotropic Gaussian can be computed efficiently.

### 4.3 Algorithm

Algorithm 1 provides a description of the proposed method. We initialize $\theta_0$ randomly and iterate for up to $K$ learning steps. In each step, we randomly partition the training data $D$ into $m$ roughly equal size batches $B_1, \ldots, B_m$. We pick a single seed batch $B_s$ among them uniformly at random. We then compute the gradient vector of the loss with respect to the model parameters under seed batch $B_s$, which results in $g_s$, and add isotropic Gaussian noise with scale $\sigma$ on it to obtain noisy gradient $\tilde{g}$.

Then we evaluate the privacy test, which first involves the computation of the other batches' gradients. If the test passes, then we update the model parameters $\theta_i$ with the noisy gradient $\tilde{g}$. Otherwise, the update is never applied (keep $\theta_i = \theta_{i-1}$) (i.e., we discard the update) and continue to the next step.

It can be seen that we can instantiate the test so as to recover (vanilla) SGD: take $T \leq 1$ $\psi = 0$, $\beta \to \infty$ and $\sigma = 0$ (no noise on the gradient).

### 4.4 Algorithmic Complexity

The computational complexity of SGD, DP-SGD and Algorithm 1 depends on the number of gradient calculations. SGD computes one gradient per step to update the parameters. DP-SGD computes $b$ gradients per step where $b$ is the batch size since it needs to compute per data point gradients for clipping. PD-SGD computes the seed batch gradient (and adds noise to it), and then (up to) $m - 1$ gradients for other batch gradients to run the privacy test.

However, since PD-SGD updates can fail, the complexity for a fixed number $k$ of successful gradient updates is $O(mk(1 - \rho)^{-1})$, where $\rho$ is the rejection rate. This is because $(1 - \rho)^{-1}$ is the expected number of iterations to get one successful gradient update. By comparison, performing $k$ gradient updates requires $O(k)$ gradient computations with SGD and $O(kb)$ with DP-SGD. Thus, for large batches ($b \gg m$) we expect PD-SGD to be faster than DP-SGD. We observe this in experiments (Appendix G.9). PD-SGD also consumes less memory than DP-SGD, albeit more than SGD. Note that evaluating the privacy test does not require keeping $O(m)$ batches' gradients simultaneously in memory, since after evaluating $\alpha$-similarity (e.g., through Eq. (3)) we can discard them.

Aside from lower computation and memory usage, not having to compute per data point gradients has other benefits. PD-SGD (unlike DP-SGD) is compatible with any loss function, not only decomposable ones. Non-decomposable are those where the batch gradient cannot be written as a sum of the individual example gradients. Kong et al. [25] discuss this issue for DP-SGD and how to mitigate it.

### 4.5 Privacy-Utility Tradeoff & Batch PD

Due to space constraints, we defer a full privacy analysis of PD-SGD to Appendix C. The key results are that: (1) PD-SGD (with suitable privacy parameters) satisfies the batch PD criterion (Definition 1) for $\lambda, \lambda'$ such that for any $1 \leq t < T$: $\lambda \leq \alpha\beta(1 + \frac{1}{t})$ (not the exact expression) and $\lambda' \leq m^{-1}\beta^{-(T-t)}$; and (2) PD-SGD (again with suitable privacy parameters) satisfies $(\varepsilon, \delta)$-differential privacy for $\varepsilon \approx \ln \lambda$ (slightly different than above) and $\delta \leq \lambda'$. These results are for one iteration of PD-SGD, but with DP composition results, we then obtain guarantees for the full training process.

## 5 Experiments

### 5.1 Experimental Setup

**Threat Model.** For the purposes of comparing PD-SGD against existing membership inference defenses, we assume a *black-box membership adversary* who knows the complete PD-SGD algorithm and its privacy parameters — and the entire pool of candidate training records, but can only interact with the **final trained model** (or its API); it does not see per-iteration mini-batches, noisy gradients, or acceptance decisions.

**Setup.** We use three of the most commonly used datasets for evaluating membership inference attacks [38, 45, 40] and DP-SGD [12, 3]: CIFAR-10, CIFAR-100, and Purchase-100. For the models, we fine-tune ViT-B-16 for CIFAR-10 and CIFAR-100, linear model for Purchase-100, and Wide ResNet for CIFAR-10 and CIFAR-100 training from scratch. Unless otherwise stated, we instantiate the privacy test using simple counting without randomizing the threshold. We found empirically that counting variants performs similarly up to parameter tuning variations. We tune privacy parameters $T$, $\gamma$ ($= \ln \alpha$), and $\sigma$ according to Appendix E. In cases where we randomized the threshold and used a ceiling, we set $\beta = e$ and $\psi = 0.2$.

To evaluate the robustness of our defense mechanisms against such adversaries, we employ black-box membership inference attacks using the Privacy Meter.[3] We use the Population Attack (P-Attack), Reference Attack (R-Attack), Shadow model Attack (S-Attack) based on [45] and Carlini et al. Attack (C-Attack) based on [7]. We provide full details in Appendix F.

### 5.2 Evaluations

**Privacy-Utility Trade-off.** We primarily evaluate utility using the trained models' test accuracies, although we include results on computational overhead in Appendix G.9. We evaluate privacy using our selected set of membership inference attacks, namely P-Attack, R-Attack, S-Attack, and C-Attack. For the first three, we report the attack AUC score. For C-Attack, we report TPR at 0.1% FPR as advocated for by [7].

We use two sets of hyperparameters for PD-SGD. Parameter Setting 1 (PS 1) is designed to optimize utility while maintaining reasonable privacy, while Parameter Setting 2 (PS 2) prioritizes better privacy at the cost of lower accuracy. Appendix E provides full details of how we tune parameters, and Table 4 shows the details of hyperparameters we used in experiments.

Our findings, summarized in Table 1, show that PD-SGD achieves a superior privacy-utility trade-off, surpassing both empirical and DP-based defenses across all evaluated tasks.

PD-SGD, particularly with PS1, achieves **comparable utility** to the non-private setting, with a 96.18% test accuracy on CIFAR-10 fine-tuning and robust performance on CIFAR-100 and Purchase-100.

---

[3]https://github.com/privacytrustlab/ml_privacy_meter/tree/173d4ad80f183ae6e1867b2793dfffe0633107d0/benchmark

Table 1: **Evaluations for PD-SGD**: We evaluate PD-SGD on three datasets with four different attacks. We report the average results and standard deviation among three independent runs. FS represents Training from scratch, and FT represents Finetuning. PS 1 and PS 2 represent parameter setting 1 and 2. We can observe that PD-SGD can achieve a better privacy-utility trade-off than other empirical defense mechanisms and DP-SGD. In this table, we set $\varepsilon = 8$ for DP-SGD, and we provide all experiment setups and full results in the Appendix.

| Task | Method | Test acc | P-Attack | R-Attack | S-Attack | C-Attack |
|---|---|---|---|---|---|---|
| CIFAR-10 (FS) | Non-private | 87.22% (±0.13%) | 0.60 (±0.01) | 0.60 (±0.01) | 0.58 (±0.01) | 0.22% (±0.03%) |
| | AdvReg | 75.38% (±0.09%) | 0.53 (±0.00) | 0.54 (±0.01) | 0.53 (±0.01) | 0.19% (±0.02%) |
| | SELENA | 81.04% (±0.07%) | 0.53 (±0.01) | 0.53 (±0.01) | 0.53 (±0.01) | 0.19% (±0.01%) |
| | DP-SGD | 63.31% (±0.15%) | 0.51 (±0.01) | 0.50 (±0.00) | 0.51 (±0.01) | 0.13% (±0.02%) |
| | PD-SGD (PS 1) | 82.22% (±0.11%) | 0.53 (±0.01) | 0.52 (±0.01) | 0.51 (±0.01) | 0.19% (±0.01%) |
| | PD-SGD (PS 2) | 79.69% (±0.25%) | 0.53 (±0.00) | 0.50 (±0.01) | 0.51 (±0.01) | 0.15% (±0.01%) |
| CIFAR-10 (FT) | Non-private | 96.09% (±0.02%) | 0.57(±0.01) | 0.69(±0.01) | 0.56 (±0.01) | 0.37% (±0.03%) |
| | AdvReg | 95.96% (±0.06%) | 0.56 (±0.01) | 0.59 (±0.01) | 0.55 (±0.00) | 0.31% (±0.01%) |
| | SELENA | 96.01% (±0.04%) | 0.55 (±0.01) | 0.51 (±0.01) | 0.56 (±0.02) | 0.33% (±0.02%) |
| | DP-SGD | 94.22% (±0.09%) | 0.54 (±0.00) | 0.59 (±0.01) | 0.54 (±0.01) | 0.23% (±0.02%) |
| | PD-SGD (PS 1) | 96.18% (±0.06%) | 0.54 (±0.01) | 0.49 (±0.01) | 0.55 (±0.01) | 0.27% (±0.02%) |
| | PD-SGD (PS 2) | 94.73% (±0.07%) | 0.53 (±0.01) | 0.49 (±0.01) | 0.53 (±0.01) | 0.20%(±0.03%) |
| CIFAR-100 (FT) | Non-private | 74.22% (±0.03%) | 0.73(±0.01) | 0.68(±0.01) | 0.73(±0.01) | 0.38% (±0.03%) |
| | AdvReg | 72.08% (±0.03%) | 0.70(±0.01) | 0.68(±0.01) | 0.72(±0.01) | 0.33% (±0.02%) |
| | SELENA | 68.46% (±0.04%) | 0.63(±0.00) | 0.60(±0.01) | 0.65(±0.01) | 0.19% (±0.02%) |
| | DP-SGD | 27.12% (±0.05%) | 0.51 (±0.01) | 0.52 (±0.01) | 0.51 (±0.01) | 0.13% (±0.03%) |
| | PD-SGD (PS 1) | 72.56% (±0.06%) | 0.67(±0.01) | 0.62(±0.01) | 0.64(±0.01) | 0.18% (±0.02%) |
| | PD-SGD (PS 2) | 68.79% (±0.05%) | 0.62(±0.01) | 0.59 (±0.01) | 0.62 (±0.01) | 0.14% (±0.02%) |
| Purchase-100 (FS) | Non-private | 68.56%(±0.12%) | 0.76(±0.01) | 0.78(±0.01) | 0.77(±0.01) | 0.12%(±0.02%) |
| | AdvReg | 57.56%(±0.07%) | 0.70(±0.01) | 0.70(±0.01) | 0.66(±0.01) | 0.08% (±0.02%) |
| | SELENA | 64.31% (±0.09%) | 0.63(±0.00) | 0.73(±0.01) | 0.66(±0.01) | 0.07%(±0.01%) |
| | DP-SGD | 47.61% (±0.12%) | 0.56(±0.00) | 0.56(±0.01) | 0.56(±0.01) | 0.08% (±0.01%) |
| | PD-SGD (PS 1) | 64.83% (±0.05%) | 0.63(±0.01) | 0.72(±0.01) | 0.64(±0.01) | 0.06% (±0.01%) |
| | PD-SGD (PS 2) | 61.16% (±0.07%) | 0.61(±0.01) | 0.59(±0.02) | 0.60(±0.01) | 0.06% (±0.01%) |

This high-utility performance extends to training from scratch. To demonstrate generalizability, we trained a Wide ResNet (WRN-16-4) model from scratch on CIFAR-10, with results shown in the row of CIFAR-10 (FS) Table 1. Here, PD-SGD (82.22%) surpasses empirical methods like SELENA (81.04%) and AdvReg (75.38%). In all cases, PD-SGD exhibits stronger membership inference attack resilience than these empirical defenses, with C-Attack performance being among the lowest recorded.

Furthermore, PD-SGD provides a favorable privacy-utility tradeoff even when privacy is paramount (PS2). For instance, on Purchase-100, there is only approximately a 7% decrease in test accuracy to obtain a reduction in attack AUC of nearly 0.15 compared to the non-private baseline. This is mirrored in the CIFAR-10 from-scratch setting, where the R-Attack AUC score shows a marked decrease from 0.60 to 0.50.

Compared to DP-SGD, the method provides comparable or better membership privacy but with **much higher test accuracy**. For instance, PD-SGD provides both higher test accuracy and better MIA defense than DP-SGD for $\varepsilon = 8$ for CIFAR-10. This superior trade-off holds in further experiments, including when fine-tuning with more data (Appendix G.11) and training from scratch on CIFAR-100 (Appendix G.12), where we find PD-SGD still achieves better utility and comparable membership privacy even when using a large $\varepsilon$ like 500 for DP-SGD.

We further illustrate the privacy-utility tradeoff between methods visually in Fig. 1. The x-axis shows the (empirical) attack advantage, and the y-axis shows the test accuracy for the WRN-16-4 model trained on CIFAR-10. Compared to DP-SGD, PD-SGD provides **higher** test accuracy for the same attack advantage, even for high privacy cases i.e., attack advantages close to 0. Compared to empirical defenses, PD-SGD not only can provide better utility with comparable attack advantage but also offers a way to navigate the tradeoff (through the privacy parameter) and not (only) a fixed point on the privacy-utility landscape.

**Scalability and Efficiency.** To show the scalability of PD-SGD, we fine-tune LLaMA-2-7B [44] on SST-2 [39] with PD-SGD. Our run reaches 94.76% test accuracy while maintaining stable memory use (peak allocated memory 6,753.48 MB). It is comparable to 94.8% reported in Table 12 of Zhao et al. [51] with normal training and 92.2% with DP-Lora [48]. This shows PD-SGD is practical for LLM fine-tuning and does not incur prohibitive memory overhead.

Table 2: **Impact of Privacy Test and Noise:** We keep all hyperparameters the same, only changing the threshold $T$ to control the privacy test. ✓means the presence of noise or the application of a privacy test, × means the absence of these components, and ⊗ represents the use of random rejection for gradient updates instead of standard privacy testing.

| Method | Noise | Privacy Test | Test acc | P-Attack | R-Attack | S-Attack | C-Attack |
|---|---|---|---|---|---|---|---|
| Non Private | × | × | 96.08% | 0.56 | 0.68 | 0.56 | 0.35% |
| Only Noise | ✓ | × | 94.99% | 0.54 | 0.57 | 0.55 | 0.30% |
| Only Privacy Test | × | ✓ | 96.01% | 0.55 | 0.56 | 0.56 | 0.32% |
| Random Rejection | ✓ | ⊗ | 94.78% | 0.55 | 0.54 | 0.54 | 0.28% |
| **PD-SGD** | ✓ | ✓ | **94.70%** | **0.53** | **0.48** | **0.53** | **0.20%** |

Although the privacy test introduces overhead compared to SGD, PD-SGD is significantly more efficient than DP-SGD. This is because PD-SGD operates at the batch level, avoiding the costly computation and storage of per-example gradients. A single training step is substantially faster, and GPU memory consumption is drastically lower — remaining comparable to non-private training and using less than half the memory required by DP-SGD in our experiments. Full details on computational time and memory usage are available in Appendix G.9.

**Privacy Test and Gradient Noise.** Compared to SGD, PD-SGD includes two components: (1) noise addition to the seed batch's gradient, and (2) a plausible deniability-based privacy test. We create a set of principled experiments to isolate the effect of these two components.

- **Only Noise:** we set $T = 1$ ($\psi = 0, \beta \to \infty$), ensuring the privacy test will always pass.
- **Only Privacy Test:** use privacy test normally, but update parameters using the un-noised gradient.
- **Random Rejection:** seed batches' gradients are randomly rejected at the same rate as PD-SGD.

Table 2 shows the results. Adding noise to the gradient without the privacy test does not effectively defend against membership inference. The R-Attack success rate decreases substantially, but there is no substantial decrease for P-Attack, S-Attack, and C-Attack. Similarly, if the privacy test is used but the gradient is un-noised or if updates are randomly rejected, we again see no major decrease in membership inference attack success rates. By contrast, PD-SGD exhibits the largest effect in mitigating membership inference attacks. The R-Attack success rate drops further to 0.48, and other attack vectors like P-Attack, S-Attack, and C-Attack are similarly reduced.

These results demonstrate that it is the combination of both noise addition and privacy testing that results in the observed privacy protection of PD-SGD.

**Additional Experiments.** In Appendix G, we explore trade-offs between the privacy parameters, show the impact of the batch size/number of batches, rejection rate for PD-SGD, and provide additional experiments such as training from scratch on CIFAR-100. We also evaluate how PD-SGD rejects anomalous batches, its computation time and GPU usage per training step, frequency of examples used in PD-SGD, compared to SoTA DP-SGD and new defense mechanism (e.g., HAMP [9]), privacy-test variants, and vulnerable data points identified by PD-SGD.

## 6    Conclusions

We proposed a new approach for training models with privacy, inspired by the concept of plausible deniability. Our construction, PD-SGD, is based on a rejection sampling approach using a privacy test. We show that PD-SGD limits membership inference attack success rate and show that PD-SGD can be instantiated to meet differential privacy guarantees. In experiments, we find that PD-SGD provides a superior privacy-utility trade-off compared to existing defense methods. This makes PD-SGD a promising solution for enhancing privacy protection in practical deep-learning applications.

There are limitations that we plan to alleviate in future work. The relationships between batch PD, membership inference, PD-SGD, and differential privacy are per iteration. They can be extended to the entire training process, but we believe that the resulting bounds are loose and that stronger composition results can be derived. Also, we focused on the centralized learning setting, and more work is required for application to collaborative or federated learning.

## Acknowledgments

We thank the anonymous NeurIPS reviewers and area chair for their helpful feedback and suggestions. This work was supported in part by the National Science Foundation under CNS-2055123. Any opinions, findings, and conclusions or recommendations expressed in this material are those of the authors and do not necessarily reflect the views of the National Science Foundation.

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

## Road Map

This appendix is structured as follows:

- **Appendix A** — *Notation*: a compact table of symbols used throughout the paper.
- **Appendix B** — *Batch PD Analysis*: more on the relationship between batch PD and other privacy games such as membership privacy, and derivation of bounds on adversary's advantage.
- **Appendix C** — *Privacy-Test Analysis*: detailed study of the privacy tests and their properties, and the derivation of the $(\varepsilon, \delta)$-DP guarantee.
- **Appendix D** — *What Batch Pass the Test*: analysis of gradient rejection and its implications.
- **Appendix E** — *Hyper-parameter Tuning*: practical guidance on choosing the noise scale $\sigma$, threshold $\gamma$, and neighbor count $T$, illustrated with privacy–utility curves.
- **Appendix F** — *Implementation Details*: datasets, model architectures, training settings, and configurations for four black-box membership-inference attacks.
- **Appendix G** — *Extended Experiments*: additional results and ablation studies, including training with larger datasets, training from scratch on CIFAR-100, rejection of anomalous batches, training-time efficiency, parameter sensitivity analyses, frequency of examples used of PD-SGD, comparison to SoTA DP-SGD, privacy-test variants, and vulnerable data points identified by PD-SGD.

## A  Symbols

Table 3: Table of Symbols.

| Symbol | Meaning | Where |
|--------|---------|-------|
| $(x, y)$ | Individual Example From Training Set | Section 2 |
| $\theta$ | Model Parameter Vector — $\theta \in \mathbb{R}^d$ | Section 2 |
| $B_i$ | SGD Mini-Batch $i$ | Section 3.1 |
| $\mathfrak{B}$ | A set of batches | Section 3.1 |
| $g_s$ | Gradient (of the Loss wrt $\theta$) of Batch $i$ | Section 4.1 |
| $B_s$ | Chosen "Seed" Batch | Section 4.1 |
| $g_s$ | Gradient of Seed Batch | Section 4.1 |
| $\tilde{g}$ | Noisy Gradient | Section 4.1 |
| $Z$ | Gaussian Noise — $\mathcal{N}(0, \sigma^2 I)$ | Section 4.1 |
| $\tau(B, \mathfrak{B})$ | The number of $\alpha$-similar batches to $B$ in $\mathfrak{B}$ | Section 4.2 |
| $\sigma$ | Privacy Parameter — Noise Scale | Section 4.1 |
| $\gamma$ | Privacy Parameter — Log-PDF Threshold | Section 4.2 |
| $\alpha$ | Privacy Parameter — $\alpha = \exp(\gamma)$ | Section 4.2 |
| $T$ | Privacy Parameter — Count Threshold | Section 4.2 |
| $\beta$ | Privacy Parameter — Threshold Randomization | Section 4.2 |
| $\psi$ | Privacy Parameter — Test Ceiling | Section 4.2 |
| $\lambda$ | PD Indistinguishability / LR Bound | Section 3.3 |
| $\lambda'$ | PD Indistinguishability / LR Slack | Section 3.3 |

## B  More on Batch PD

In this section, we provide more details on the game based construction of Section 3.2 and derive advantage bounds based on the PD criterion Section 3.3.

### B.1  From Batch PD to Membership Privacy

Consider the series of games from Fig. 3. These games are provided in the style of Salem et al. [34] and connect batch PD to (one-step) MIA. We assume there are at least $m > 2$ batches that are disjoint and of roughly equal size. We let $\mathcal{D}$ denotes the data distribution (used only in some games).

**Difference between Batch PD ($G_0$) and $G_1$:** In $G_1$ the adversary does not get to pick a full target batch, but only a single data point.

Observe that: $\mathrm{adv}_{G_0} \geq \mathrm{adv}_{G_1}$. Any adversary that wins $G_1$ with some advantage can play $G_0$ and output the same $z$ as $B_t$ as it would when playing $G_1$.

**Difference between $G_1$ and $G_2$:** In $G_2$ the adversary picks the dataset $S$ but does not determine the partitioning into batches (or learn it).

We have: $\mathrm{adv}_{G_1} \geq \mathrm{adv}_{G_2}$. If adversarial partitioning provides any benefit then adversaries for $G_1$ can use that but those for $G_2$ cannot.

**Difference between $G_2$ and $G_3$:** in $G_3$ the adversary picks only the target data $z$, not the dataset $S$ (which is sampled randomly).

We have: $\mathrm{adv}_{G_2} \geq \mathrm{adv}_{G_3}$. We conclude: $\mathrm{adv}_{G_0} \geq \mathrm{adv}_{G_3}$. A one-step algorithm $\mathcal{T}$ that achieves batch PD also has a bounded (one-step average) membership inference advantage.

**Relationship to other privacy games.** The Salem et al. [34] SoK establishes the relationship between a number of data inference games. However, the games in Salem et al. [34] are (1) not one step; (2) do not model batches and partition; and (3) operate "replace-one" setting (i.e., adversary chooses $z_0, z_1$ but $z_b$ is included; e.g., Game10 in the SoK) not in the "add-remove" setting.

Differences (1) and (2) are a consequence of working with a batch notion. We need the games to explicitly model the batches. We do not believe (3) is a consequential difference. It is well-known that the "replace-one" and "add-remove" settings can be related (usually a constant factor is the difference between the two). To be consistent with the rest of the paper, we follow the "add-remove" setting.

## B.2 Bounding Adversarial Success Rate

We show that the PD criterion is a sufficient (but not necessary) condition to bound the advantage of any adversary at the batch PD game. Recall that a learning algorithm $\mathcal{T}$ satisfies the PD criterion (Definition 1) iff for any two sets of batches $\mathfrak{B}$ and $\mathfrak{B}'$ that differ in exactly one batch, the distributions $\Pr_{\mathcal{T}}(\cdot | \mathfrak{B})$ and $\Pr_{\mathcal{T}}(\cdot | \mathfrak{B}')$ are $(\lambda, \lambda')$-indistinguishable for $\lambda \geq 1$ and $0 \leq \lambda' < 1$.

Consider the batch PD game $G_0$ in Fig. 3. Let $H_b$ be the hypothesis corresponding to bit $b$. The likelihood ratio is $r(g) = \frac{\Pr(g|H_1)}{\Pr(g|H_0)}$. For conciseness, we write $l_b(g) = \Pr(g|H_b)$. In this game (and all the ones we consider in this paper), we have equal priors since $\Pr(b = 1) = 1/2$ and so the posterior odds are exactly the likelihood ratio, i.e., the best adversary guesses based on the likelihood ratio ($b' = 1$ iff $r(g) \geq 1$ else $b' = 0$).

Advantage is the total variation distance between the two likelihoods, i.e., $\mathrm{adv} = \mathrm{adv}_{G_0} = \frac{1}{2}|l_1 - l_0|_1$. Alternatively, we can reason in terms of type I and type II errors. Define $A = \{g : r(g) \geq 1\} = \{g : l_1(g) \geq l_0(g)\}$ the region under which the adversary decides in favor of $H_1$. Let $P_D = \Pr(A|H_1)$ (probability of detection), $P_{FA} = \Pr(A|H_0)$ (false alarm), and $P_{MD} = \Pr(A^c|H_1) = 1 - P_D$. Since: $\mathrm{adv} = \Pr(A|H_1) - \Pr(A|H_0) = P_D - P_{FA} = 1 - (P_{FA} + P_{MD})$, maximizing advantage is the same as minimizing the sum of missed detections and false alarms.

**Lemma 1.** *Let $G_0$ denote the batch PD game. If a learning algorithm $\mathcal{T}$ satisfies the batch PD criterion (Definition 1) for some $\lambda \geq 1$ and $\lambda' \geq 0$. Then the advantage of any adversary in the batch PD game is bounded:*
$$\mathrm{adv}_{G_0} \leq \frac{\lambda - 1}{\lambda + 1} + \frac{\lambda'}{\lambda + 1} \leq \frac{\lambda - 1}{\lambda + 1} + \frac{\lambda'}{2} .$$

We provide the proof in Appendix H.1. As we show in Appendix C.2, Lemma 1 applies to the PD-SGD construction for $\lambda, \lambda'$ depending only on the privacy parameters (and the number of batches).

## B.3 Multiple Steps

The definitions and games discussed so far focus on a single learning step. We can extend the batch PD game to $k > 1$ iterations by keeping $b$ fixed across iterations, but letting the adversary choose different sets of batches and target batch across iterations. The adversary gets all of the gradients produced and guesses bit $b'$ at the end.

**Game $G^{(k)}$ — $k$-steps Batch PD**

1: $\theta_0$ initialized randomly
2: $b \sim \{0, 1\}$          ▷ sample random bit $b$
3: **for** $i = 1, 2, \ldots, k$ **do**
4:      $(B_1, B_2, \ldots, B_m), B_t \leftarrow A'(\mathcal{T}_{\theta_{i-1}})$      ▷ adversary pick batches and the target batch
5:      **if** $b = 1$ **then**
6:          $g_i \leftarrow \mathcal{T}_{\theta_{i-1}}(B_1, B_2, \ldots, B_m, B_t)$      ▷ gradient from $\mathcal{T}$ with $B_t$ included
7:      **else**
8:          $g_i \leftarrow \mathcal{T}_{\theta_{i-1}}(B_1, B_2, \ldots, B_m)$      ▷ gradient from $\mathcal{T}$ *without* $B_t$
9:      **end if**
10:      Update $\theta_i$
11: **end for**
12: $b' \leftarrow A(g_1, \ldots, g_k, \mathcal{T})$

Note that the algorithm depends on the current parameters $\theta$ at each iteration and therefore the distribution across iterations, even for the same batches, is not identical.

Because the batches (adversarially chosen) may change at every iteration, not every possible run of this game maps onto a realistic training process. In an actual training process with an SGD-like algorithm, the set of batches changes but is constrained to always be a valid partition of the training dataset. However, a bound on the advantage of the best adversary for this game is a bound on any adversary $(A, A')$, including those adversaries that are restricted to choose batches so as to mirror an actual training process (by restriction on $A'$). This allows the game to capture numerous scenarios.

For example, this can model the case where there is a fixed dataset $D$ and the adversary wants to determine whether a small set of "correlated" examples $S = (x_1, y_1), \ldots, (x_l, y_l)$ was part of the training data. In this case, $A'$ can select any partition $B_1, \ldots, B_m$ (different at each iteration) such that their union equals $D$ and as the target batch $B_t$ any batch that includes all of $S$ (i.e., $S \subseteq B_t$). This then captures the case where (for whatever reason) the dataset may include all of $S$ in one batch (e.g., the training process is poisoned in some way).

## C    Privacy: Tests, Batch PD & Differential Privacy

In this section, we explore the relationship between privacy testing and batch PD and between PD-SGD and differential privacy.

### C.1    Privacy Testing

Recall from Section 4.2 the different ways of counting alternatives in the privacy test. The integer and clique variants have the property that adding or removing any batch changes the count of alternatives by at most 1. Also recall that we considered a randomized version of the test that also caps the probability of passing the test.

**Randomized Thresholds.** Given the count $\tau$ of alternatives, we randomize the test based on noise $z \sim \mathrm{Geom}(\beta)$ so that the test passes iff $\tau + z \geq T$. We define $z \sim \mathrm{Geom}(\beta)$ for $\beta > 1$ so that $\Pr(z = i) = \frac{\beta - 1}{\beta + 1} \beta^{-|i|}$ for any integer $i$. This type of geometric noise was initially proposed by Ghosh et al. [19].

The probability of passing the test is $\mathbb{E}[1_{\tau + z \geq T}]$, where we can think of $\tau + z$ as a noisy count. For conciseness, we write $p_\tau = \Pr(z \geq T - \tau)$ to denote the probability that the test passes given the count $\tau$. All our privacy test variants are such that the probability of passing the test only depends on $\tau$. When using a test such that the sensitivity of $\tau$ is 1 then the probability of passing the test when adding or removing one batch changes only by a bounded amount (e.g., by at most a factor of $\beta$ when increasing $\tau$ by 1).

**Adding a Ceiling.** We also have a ceiling on the probability of passing the test. If there are enough alternatives (i.e., $\tau + z \geq T$) then we flip a coin with probability of heads $1 - \psi$ (for some $\psi > 0$). If the coin lands on heads then the test passes. Otherwise the test fails. This means the probability of passing the test is $p_\tau = (1 - \psi)\Pr(\tau + z \geq T)$. For example, if $\psi = 0.2$ then the test never passes with probability higher than 0.8.

**Properties.** With this setup, we obtain the following properties that describe how the probability of passing the test changes for changing number of alternatives.

**Lemma 2.** *Let $p_\tau$ be the probability of passing the test with $\tau \geq 0$ alternatives for a test with randomization $\mathrm{Geom}(\beta)$ and ceiling $1 - \psi$. We have:*

$$1 \leq \frac{p_{\tau+1}}{p_\tau} \leq \beta \quad and \quad 1 \geq \frac{1 - p_{\tau+1}}{1 - p_\tau} \geq \frac{\psi}{1 - \frac{1-\psi}{\beta}} \ .$$

The lemma states that $p_\tau$ is non-decreasing as a function of alternatives $\tau$ and then when increasing the count by one (e.g., if adding a batch), the probability of passing the test increases by a factor of at most $\beta$. Further, the change in probability of failing the test due to increasing the count by one is lower bounded as a function of $\beta$ and $\psi$.

Moreover, when $\tau$ is far from the threshold $T$, the test passes with exponentially small probability.

**Lemma 3.** *Let $p_\tau$ be the probability of passing the test with $\tau \geq 0$ alternatives for a test with randomization $\mathrm{Geom}(\beta)$ and ceiling $1 - \psi$. For any $1 \leq t < T$:*

$$p_t \leq (1 - \psi) \exp\left(-\varepsilon_0 (T - t)\right),$$

*where $\varepsilon_0 = \ln \beta$.*

The proofs of Lemmas 2 and 3 are in Appendix H.2.

We provide experiments that compare different variants of the privacy test in Appendix G.8.

## C.2 PD Criterion

Recall from Lemma 1 that any (one-step) learning algorithm $\mathcal{T}$ that satisfies the PD criterion has bounded advantage (for batch PD game, membership inference, etc.). We show that the properties of the test in combination with $\alpha$-similarity yield Definition 1.

**Lemma 4.** *Let $\mathcal{T}$ denote a single training iteration of PD-SGD (Algorithm 1) with a privacy test with parameters $T > 1$, $\alpha > 1$, randomization $\mathrm{Geom}(\beta)$ and ceiling $1 - \psi$ (for $\beta \geq 1$ and $1 > \psi > 0$). For any integer $1 \leq t < T$: $\mathcal{T}$ satisfies the PD criterion (Definition 1) (on any partition of $m \geq T$ batches) for:*

$$\lambda = \lambda(t) = \max\left\{\frac{m}{m+1}\beta\left[1 + \frac{\alpha}{t}\right], \frac{m+1}{m}\frac{\beta - 1 + \psi}{\beta\psi}, \frac{m}{m+1}(1 + \frac{1}{m\psi})\right\}$$

*and*

$$\lambda' = \lambda'(t) = \frac{1 - \psi}{m+1}\beta^{-(T-t)} \ .$$

In this lemma, $t$ can be freely chosen to trade off between $\lambda$ and $\lambda'$. The proof is in Appendix H.3. Combining Lemmas 1 and 4 yields the advantage bound on batch PD.

The $\lambda$ term in Lemma 4 can be simplified for some constraints on the privacy parameters, in which case we expect the term that depends on $\alpha$, $\beta$ and $t$ to denominate. For example, we can eliminate the dependence on $\psi$ by choosing $\psi = (\beta + 1)^{-1}$ in which case the other terms are at most $\frac{m+1}{m}\beta$. For a large enough number of batches, $\frac{m}{m+1}$ approaches 1 so behavior is driven by the term $\beta(1 + \alpha t^{-1})$.

## C.3 Differential Privacy

For reasons analogous to PD-SGD satisfying the PD criterion (Lemma 4), we can show that the algorithm satisfies $(\varepsilon, \delta)$-differential privacy.

However, we stress that batch PD and differential privacy are not directly compatible in the sense that the input to the learning algorithm $\mathcal{T}$ for batch PD is a partition of batches, whereas for differential privacy it is a dataset $D$. This means that for differential privacy, we need a learning algorithm $\mathcal{T}'$ to first randomly partition $D$ into a set of batches before passing those batches to $\mathcal{T}$. This straightforward conceptually, but it highlights an important consequence: the batch PD guarantee has to hold for the worst case partitioning (since it's adversarially chosen). In the differential privacy case, the adversary does not know the partition of the dataset into batches.

**Lemma 5.** *Let $\mathcal{T}$ be as in Lemma 4 with privacy parameters $(T, \alpha, \sigma, \beta, \psi)$ and $\mathcal{M}$ the algorithm that first randomly partitions the dataset into $m \geq T > 1$ batches (of roughly equal size) before invoking $\mathcal{T}$. For any two neighboring datasets $D_1$, $D_2$, any output set $S \subseteq \mathrm{Range}(\mathcal{M})$ and any integer $1 \leq t < T$, $\mathcal{M}$ satisfies $(\varepsilon, \delta)$-differential privacy. That is:*

$$\Pr(Y|\mathcal{M}(D_1)) \leq e^\varepsilon \Pr(Y|\mathcal{M}(D_2)) + \delta \,,$$

*where $\varepsilon = \ln \beta \left(1 + \frac{1}{t}\alpha\right)$ and $\delta \leq \frac{1}{m}(1 - \psi) \exp\left(-\varepsilon_0 (T - t)\right)$. Here $\varepsilon_0 = \ln \beta, \alpha = \exp(\gamma)$.*

We provide a proof in Appendix H.4.

**Parameter Tuning.** It is possible to tune the parameters so the guarantee is fairly stringent. Similarly to earlier, we may want to set a relatively low ceiling, i.e., $\psi = (\beta + 1)^{-1}$. To ensure a small $\delta$, it is desirable to set $t$ such that $T - t$ is relatively large. For example, we can set $T$ and $t$ such that $\exp(-\varepsilon_0(T - t)) = \frac{1}{|D|}$ which requires $T = \frac{\ln |D|}{\varepsilon_0} + t$ and ensures that $\delta \leq \frac{1}{m}(1 - \psi)\frac{1}{|D|}$. That is $\delta$ is asymptotically smaller than $|D|^{-1}$ if we consider a fixed batch size (so that as $|D|$ increases so does $m$). In such a case, we get $\varepsilon \leq \gamma + \varepsilon_0 + \ln\left(1 + \frac{1}{t}\right) = \gamma + \varepsilon_0 + \ln\left(1 + (T - \frac{\ln |D|}{\varepsilon_0})^{-1}\right)$. Note that $T \leq m$ so everything else equal, it is more challenging to get good privacy with a smaller number of batches.

**Composition.** We can apply advanced composition [18] (Theorem 3.20) to obtain an overall guarantee for the training process. So for $K$ steps we get:

$$\varepsilon' = \sqrt{2K \ln(\frac{1}{\delta''})}\varepsilon + K\varepsilon(e^\varepsilon - 1) \text{ and } \delta' = K\delta + \delta'',$$

where $(\varepsilon', \delta')$ is the privacy budget for an entire training run and $1 \gg \delta'' > 0$ can be freely chosen to control the tradeoff between $\varepsilon'$ and $\delta'$.

As previously mentioned, these DP bounds may be overly pessimistic in the sense that the adversary even knows the partition of data into batches. We leave to future work the task of deriving tighter bounds in more realistic settings.

## D   What Batch Pass the Privacy Test?

So far, we have analyzed the privacy of our approach from the lens of batch PD and differential privacy. Another lens we can adopt is to ask what kind of batches get rejected? In this section, we show that gradient updates from anomalous batches are rejected with high probability (even for the simplest variant of the privacy test).

Consider a seed batch $B_s$, its associated gradient $g_s$, and another batch $B_i$ with gradient $g_i$. Recall that a noisy gradient $\tilde{g} = g_s + Z$ is plausibly deniable with respect to batch $B_i$ iff Eq. (2) holds. In other words, we denote plausibility (of $\tilde{g}$ with respect to some $g_i$) as the probability that Eq. (2) holds:

$$q(s, i) = \Pr\left[\alpha^{-1} \leq \frac{p(\tilde{g} - g_s)}{p(\tilde{g} - g_i)} \leq \alpha\right],$$

where the probability $q(s, i)$ is taken over the randomness of $Z \sim \mathcal{N}(0, \sigma^2 I)$. This probability only depends on batches $B_s$ and $B_i$. The following result shows that it only depends on the $l_2$-distance between the two gradients, i.e., $||g_s - g_i||_2$.

**Lemma 6.** *For any seed batch with gradient $g_s$ and any mini-batch with gradient $g_i$, let $d = ||g_s - g_i||_2^2$. The probability that Eq. (2) holds depends only $d$ and we have:*

$$q(d) = q(s, i) = \Pr\left(Y \in \left[\frac{d - \tilde{\gamma}}{2\sigma\sqrt{d}}, \frac{d + \tilde{\gamma}}{2\sigma\sqrt{d}}\right]\right), \tag{4}$$

*where $Y \sim \mathcal{N}(0, 1)$ and $\tilde{\gamma} = 2\sigma^2 \gamma$.*

Lemma 6 shows that $q(d)$ is exactly the probability that a standard normal variable takes a value in $[\frac{d - \tilde{\gamma}}{2\sigma\sqrt{d}}, \frac{d + \tilde{\gamma}}{2\sigma\sqrt{d}}]$ where $\tilde{\gamma} = 2\sigma^2 \gamma$. We provide a proof in Appendix H.5.

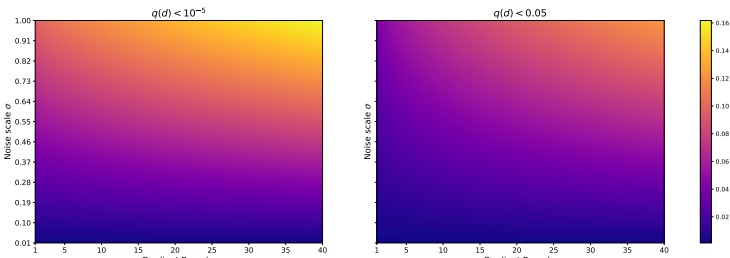

Figure 4: **Normalized Distance $d^*$ for varying $\sigma$ and $\gamma$ under different** $q(d)$. We observe that for a fixed probability of passing the test $q(d)$, the larger the product of $\sigma$ and $\gamma$ the larger the normalized distance $d^*$ can be, meaning that more anomalous batches pass the privacy test. Note that $d^* = \sqrt{d/k}$ where $k$ is the dimension of the gradient vector (we set $k = 7680$ for this case).

Intuitively, for $a \gg b > 0$ the probability $\Pr(a - b \le Y \le a + b)$ can be reasonably approximated as $2b\phi(a)$ where $\phi(\cdot)$ is the standard normal pdf, and thus the probability falls exponentially fast with $a$.

The following results derived from tail bounds on Lemma 6 show that plausibility falls off **exponentially** fast with the $l_2$-norm $d$ whenever $d$ is sufficiently large with respect to $\tilde{\gamma}$. This immediately implies that **any highly anomalous candidate gradient** (i.e., gradient with large $l_2$-norm to all other mini-batch gradients) **will be rejected with high probability.**

**Lemma 7.** *For any seed batch with gradient $g_s$ and any mini-batch with gradient $g_i$, and let $d$ be defined as in Lemma 6. If $d > 2\sigma^2\gamma$, we have that:*

$$q(d) < C_{d,\gamma,\sigma} \cdot \exp\left(-\left[\frac{d^2 + \tilde{\gamma}^2}{8d\sigma^2}\right]\right). \tag{5}$$

*where $C_{d,\gamma,\sigma} = \frac{\sqrt{2d}\sigma}{\sqrt{\pi}} \cdot \left[\exp\left(\frac{\gamma}{2}\right)(d - \tilde{\gamma})^{-1} - (d + \tilde{\gamma}) \cdot \exp\left(-\frac{\gamma}{2}\right)\left[(d + \tilde{\gamma})^2 + 4\sigma^2 d\right]^{-1}\right].$*

We provide the proof of Lemma 7 in Appendix H.5.

## E  Parameter Tuning

There are two main strategies to approach parameter tuning.

**Theory-based strategy.** We can tune parameters based on the rejection rate theory from Lemma 7. By tuning $\sigma$ and $\gamma$, we can make $q(d)$ arbitrarily small. If we have a desired bound on $d$, then we can find combinations of $\sigma$ and $\gamma$ that achieve the desired effects. This can, for example, be done through a grid search.

To provide intuition and guide parameter tuning, we plot the minimum $d$ such that $q(d)$ is at most some $\delta > 0$ as a function of $\gamma$ and $\sigma$. This is shown in Fig. 4 for $\delta = 0.05$ and $\delta = 10^{-5}$, which plots $\sqrt{d/k}$, where $k$ is the dimension of the gradient vector (i.e., $g \in \mathbb{R}^k$) used here for normalization. We observe that (as expected) we require larger $d^*$ for the same $\sigma$ and $\gamma$ for $q(d) < 10^{-5}$ compared to $q(d) < 0.05$. Moreover, for a fixed $q(d)$, the normalized distance $d^*$ appears to grow with the product of $\sigma$ and $\gamma$. This is consistent with Lemma 7, which suggests that the asymptotic behavior is driven by the product $\sigma^2\gamma$. Wen tuning the privacy parameters, exploring combinations of $\sigma$ and $\gamma$ such that $\sigma^2\gamma$ remains roughly constant is a sensible strategy.

Alternatively, we can tune parameters based on the connection between PD-SGD and differential privacy (Appendix C.3). In particular, we can set $T \ge \varepsilon_0^{-1} \ln|D| + t$ to ensure a low enough $\delta$. In that case $\varepsilon = \gamma + \varepsilon_0 + \ln\left(1 + \frac{1}{t}\right)$ so minimizing $\gamma$ maximizes privacy. However, if the chosen pair $(\gamma, \sigma)$ does not allow passing the test often enough, then the utility suffers. Keeping $\sigma^2\gamma$ constant to ensure a reasonable rejection rate and then tuning other parameters is also sensible.

**Practical Parameter Tuning.** We propose and evaluate an empirical parameter tuning strategy. We conducted additional experiments on an NVIDIA B200 GPU.

---

**Algorithm 2** Hyperparameter Search

---

**Input:**
1: $\sigma_{\text{list}}$        $\leftarrow$ list of candidate noise scales
2: $\gamma_{\text{list}}$        $\leftarrow$ list of candidate gamma values
3: $T_{\text{list}}$        $\leftarrow$ list of candidate thresholds
4: $E_{\text{full}}$        $\leftarrow$ number of epochs for full training
5: $E_{\text{short}}$      $\leftarrow$ number of steps for short trial (e.g., 200)
6: $ACC_{\text{thresh}}$ $\leftarrow$ accuracy threshold to keep $\sigma$ (e.g., 96%)
7: train_fn     $\leftarrow$ training function that returns (reject_rate, final_accuracy)
**Output:**
8: BestConfigs $\leftarrow$ list of viable $(\sigma, \gamma, T)$ configurations

9:                                               $\triangleright$ Phase 1: $\sigma$ Screening with $T = 1$
10: $\sigma_{\text{filtered}} \leftarrow []$
11: **for each** $\sigma$ in $\sigma_{\text{list}}$ **do**
12:      acc $\leftarrow$ train_fn$(\sigma, \gamma = \text{default}, T = 1, \text{epochs} = E_{\text{full}})$.accuracy
13:      **if** acc $\geq ACC_{\text{thresh}}$ **then**
14:          append $\sigma$ to $\sigma_{\text{filtered}}$
15:      **end if**
16: **end for**

17:                                          $\triangleright$ Phase 2: Coarse Filtering over $(\sigma, \gamma, T)$
18: CandidateConfigs $\leftarrow []$
19: **for each** $\sigma$ in $\sigma_{\text{filtered}}$ **do**
20:      **for each** $\gamma$ in $\gamma_{\text{list}}$ **do**
21:          **for each** $T$ in $T_{\text{list}}$ **do**
22:              (rej, acc) $\leftarrow$ train_fn$(\sigma, \gamma, T, \text{epochs} = E_{\text{short}})$
23:              **if** $0 < \text{rej} < 1$ **then**
24:                  append $(\sigma, \gamma, T)$ to CandidateConfigs
25:              **end if**
26:          **end for**
27:      **end for**
28: **end for**

29: BestConfigs $\leftarrow []$
30: **for each** $(\sigma, \gamma, T)$ in CandidateConfigs **do**
31:      (rej, acc) $\leftarrow$ train_fn$(\sigma, \gamma, T, \text{epochs} = E_{\text{full}})$
32:      record $(\sigma, \gamma, T, \text{rej}, \text{acc})$ in BestConfigs
33: **end for**

34: **return** BestConfigs

---

As a baseline, we first performed a full grid search over the three main privacy parameters ($\sigma$, $\gamma$, and $T$), covering 180 combinations in total. This exhaustive search required 110.41 GPU hours to complete.

We then applied our empirically guided two-phase strategy, which significantly reduces tuning cost:

Phase 1 ($\sigma$ screening): We fix the threshold $T = 1$ and run full training for each candidate $\sigma$. We retain only those $\sigma$ values that achieve high utility (final test accuracy $\geq 96\%$).

Phase 2 (coarse filtering of $\gamma$ and $T$): For each surviving $\sigma$, we run shortened training (200 steps) across the $\gamma$ and $T$ grid. We discard any configuration that results in degenerate behavior (i.e., reject rate reaches 0% or 100%) early. The remaining viable combinations are then trained to convergence. We provide full pseudo-code in Algorithm 2.

This procedure reduces total tuning time from 110.41 GPU hours to just **4.15** GPU hours, while still identifying high-performing configurations. In practice, this makes PD-SGD much more efficient to tune than standard grid search. Notably, we used the full grid primarily to ensure fair comparison across baselines, not because PD-SGD requires it. We believe that the combination of theory-informed constraints, practical heuristics, and structured filtering makes PD-SGD both scalable and practical.

# F Experiments Setup

## F.1 Datasets

We use three of the most commonly used datasets for evaluating membership inference attacks [38, 45, 40] and DP-SGD [12, 3].

**CIFAR-10** [26] contains 60,000 images with 10 classes. We use 50,000 as the full training set and 10,000 as the test set as most papers do. Each example has three RGB channels and size $32 \times 32$ pixels. For fine-tuning tasks, we only use 500 data samples for training and for training from scratch tasks, we use 30,000 for training.

**CIFAR-100** is a well-known benchmark in the field of computer vision, also collected by [26]. CIFAR-100 contains 60,000 color images, each with a resolution of $32 \times 32$ pixels. It is more complex than the CIFAR-10 dataset; the images are organized into 100 distinct classes. The dataset allocation includes 50,000 images for training purposes and 10,000 for testing. For the fine-tuning task, we only use 1000 data samples for training and the rest of training data examples are used for MIA evaluation. For training from scratch, we use 25,000 data samples as the same setting in [50].

**Purchase-100** is based on Kaggle's "acquire valued shoppers" challenge[4] and processed and simplified as introduced in [38]. The dataset contains shopping records for thousands of individuals and includes 197,324 data entries. For training, we use 25,000 samples and the rest for testing. For MIAs, we use 25,000 samples from test set as shadow dataset.

## F.2 Models

**ViT-B-16** are pre-trained on the LAION-2B dataset [37]. We obtain the model from Open Clip[5] and add a linear layer as a classification head. We only fine-tune this last layer and freeze the weights of other layers. We utilize this model for CIFAR-10 and CIFAR-100 fine-tuning tasks.

**Wide ResNet (WRN)** [49] is a popular variant of the ResNet (Residual Network) model [21]. The architecture increases the number of channels in convolutional layers (width) rather than the number of layers (depth). We use WRN-16-4 in experiments which is also commonly used in many DP-SGD related work [3, 12, 36]. We train the model from scratch on CIFAR-10. We use WRN-28-2 for training from scratch on CIFAR-100.

**Linear model** is commonly used for tabular data such as Purchase-100. We use this one-layer linear model for experiments on Purchase-100.

## F.3 Setups

We implemented PD-SGD using PyTorch. For DP-SGD, we use Opacus [47]. For other empirical defense mechanisms, we reproduce them using SELENA's [40] original code-base [6] and HAMP's original code-base [7]. For membership inference attack, we use the Privacy Meter toolbox.[8] From it, we use Population Attack (P-Attack), Reference Attack (R-Attack), Shadow model Attack (S-Attack) based on [45] and Carlini et al. Attack (C-Attack) based on [7]. We employ these four widely used attacks to comprehensively evaluate empirical privacy leakage and make fair comparisons between different methods. Note that our goal here is not to use the most exotic or recent attack, but to establish a fair empirical comparison between different defense methods, and thus we use a well-understood set of popular recent membership inference attacks.

**Details for Attacks:** We keep the same attack setting for all defense mechanisms for a fair comparison. For all datasets, other than the part we used for training the target models, the rest of training samples are used as shadow datasets for shadow models or reference models. For all shadow models or reference models, we sample the same amount of data samples as target dataset for training. We use

---

[4]`https://kaggle.com/c/acquire-valued-shoppers-challenge/data`
[5]`https://github.com/mlfoundations/open_clip`
[6]`https://github.com/inspire-group/MIAdefenseSELENA`
[7]`https://github.com/DependableSystemsLab/MIA_defense_HAMP`
[8]`https://github.com/privacytrustlab/ml_privacy_meter/tree/173d4ad80f183ae6e1867b2793dfffe0633107d0`

Table 4: Hyperparameters setting for experiments in Table 1

| Dataset | Param setting | $\sigma$ | $\gamma$ | T | Step | Reject Rate |
|---|---|---|---|---|---|---|
| CIFAR-10(FT) | PS 1 | 0.1 | 40 | 2 | 20000 | 27.78% |
| | PS 2 | 0.3 | 2 | 3 | 20000 | 30.31% |
| CIFAR-100(FT) | PS 1 | 0.1 | 50 | 3 | 20000 | 44.08% |
| | PS 2 | 0.2 | 10 | 3 | 20000 | 46.35% |
| Purchase-100(FS) | PS 1 | 0.01 | 1000 | 3 | 100000 | 3.91% |
| | PS 2 | 0.01 | 750 | 3 | 100000 | 87.76% |
| CIFAR-10 (FS) | PS 1 | 0.01 | 40000 | 3 | 100000 | 1.06% |
| | PS 2 | 0.02 | 7000 | 3 | 100000 | 3.70% |
| CIFAR-100 (FS) | PS 1 | 0.01 | 100000 | 3 | 10000 | 0.49% |
| | PS 2 | 0.01 | 9000 | 3 | 10000 | 32.81% |

Table 5: **Evaluations for PD-SGD**: We evaluate PD-SGD on three datasets with four different attacks. We report the average results and standard deviation among three independent runs. We can observe that PD-SGD can achieve a better privacy-utility trade-off than other empirical defense mechanisms and DP-SGD.

| Dataset | Method | Test acc | P-Attack | R-Attack | S-Attack | C-Attack |
|---|---|---|---|---|---|---|
| CIFAR-10 (FS) | Non-private | 87.22% (±0.13%) | 0.60 (±0.01) | 0.60 (±0.01) | 0.58 (±0.01) | 0.22% (±0.03%) |
| | AdvReg | 75.38% (±0.09%) | 0.53 (±0.01) | 0.54 (±0.01) | 0.53 (±0.01) | 0.19% (±0.02%) |
| | SELENA | 81.04% (±0.07%) | 0.53 (±0.01) | 0.53 (±0.01) | 0.53 (±0.01) | 0.19% (±0.01%) |
| | DP-SGD ($\varepsilon = 1$) | 26.53% (±0.48%) | 0.50 (±0.00) | 0.49 (±0.01) | 0.50 (±0.01) | 0.07% (±0.02%) |
| | DP-SGD ($\varepsilon = 4$) | 55.46% (±0.28%) | 0.50 (±0.01) | 0.49 (±0.01) | 0.50 (±0.01) | 0.10% (±0.01%) |
| | DP-SGD ($\varepsilon = 8$) | 63.31% (±0.15%) | 0.51 (±0.01) | 0.50 (±0.00) | 0.51 (±0.01) | 0.13% (±0.02%) |
| | PD-SGD (param setting 1) | 82.22% (±0.11%) | 0.53 (±0.01) | 0.52 (±0.01) | 0.51 (±0.01) | 0.19% (±0.01%) |
| | PD-SGD (param setting 2) | 79.69% (±0.25%) | 0.53 (±0.00) | 0.50 (±0.01) | 0.51 (±0.01) | 0.15% (±0.01%) |
| CIFAR-10 (FT) | Non-private | 96.09% (±0.02%) | 0.57 (±0.01) | 0.69 (±0.01) | 0.56 (±0.01) | 0.37% (±0.03%) |
| | AdvReg | 95.96% (±0.06%) | 0.56 (±0.01) | 0.55 (±0.01) | 0.55 (±0.00) | 0.31% (±0.01%) |
| | SELENA | 96.01% (±0.04%) | 0.55 (±0.00) | 0.51 (±0.01) | 0.56 (±0.02) | 0.33% (±0.02%) |
| | DP-SGD ($\varepsilon = 1$) | 68.97% (±0.11%) | 0.52 (±0.01) | 0.50 (±0.01) | 0.52 (±0.01) | 0.17% (±0.01%) |
| | DP-SGD ($\varepsilon = 4$) | 93.53% (±0.07%) | 0.54 (±0.01) | 0.56 (±0.02) | 0.54 (±0.01) | 0.20% (±0.03%) |
| | DP-SGD ($\varepsilon = 8$) | 94.22% (±0.09%) | 0.54 (±0.00) | 0.59 (±0.01) | 0.54 (±0.01) | 0.23% (±0.02%) |
| | PD-SGD (param setting 1) | 96.18% (±0.06%) | 0.54 (±0.01) | 0.49 (±0.01) | 0.55 (±0.01) | 0.27% (±0.02%) |
| | PD-SGD (param setting 2) | 94.73% (±0.07%) | 0.53 (±0.01) | 0.49 (±0.01) | 0.53 (±0.01) | 0.20% (±0.03%) |
| CIFAR-100 (FT) | Non-private | 74.22% (±0.03%) | 0.73 (±0.01) | 0.68 (±0.01) | 0.73 (±0.01) | 0.38% (±0.03%) |
| | AdvReg | 72.08% (±0.03%) | 0.70 (±0.01) | 0.68 (±0.01) | 0.72 (±0.01) | 0.33% (±0.02%) |
| | SELENA | 68.46% (±0.04%) | 0.63 (±0.00) | 0.60 (±0.01) | 0.65 (±0.01) | 0.19% (±0.02%) |
| | DP-SGD ($\varepsilon = 1$) | 4.46% (±0.13%) | 0.50 (±0.01) | 0.50 (±0.00) | 0.50 (±0.01) | 0.10% (±0.01%) |
| | DP-SGD ($\varepsilon = 4$) | 18.37% (±0.06%) | 0.50 (±0.01) | 0.50 (±0.01) | 0.51 (±0.01) | 0.12% (±0.02%) |
| | DP-SGD ($\varepsilon = 8$) | 27.12% (±0.05%) | 0.51 (±0.01) | 0.52 (±0.01) | 0.51 (±0.01) | 0.13% (±0.03%) |
| | PD-SGD (param setting 1) | 72.56% (±0.06%) | 0.67 (±0.01) | 0.62 (±0.01) | 0.64 (±0.01) | 0.18% (±0.02%) |
| | PD-SGD (param setting 2) | 68.79% (±0.05%) | 0.62 (±0.01) | 0.59 (±0.01) | 0.62 (±0.01) | 0.14% (±0.02%) |
| Purchase-100 (FS) | Non-private | 68.56% (±0.12%) | 0.76 (±0.01) | 0.78 (±0.01) | 0.77 (±0.01) | 0.12% (±0.02%) |
| | AdvReg | 57.56% (±0.07%) | 0.70 (±0.01) | 0.70 (±0.01) | 0.66 (±0.01) | 0.08% (±0.02%) |
| | SELENA | 64.31% (±0.09%) | 0.63 (±0.00) | 0.73 (±0.01) | 0.66 (±0.01) | 0.07% (±0.01%) |
| | DP-SGD ($\varepsilon = 1$) | 22.51% (±0.22%) | 0.53 (±0.01) | 0.54 (±0.01) | 0.54 (±0.00) | 0.04% (±0.01%) |
| | DP-SGD ($\varepsilon = 4$) | 43.46% (±0.15%) | 0.56 (±0.01) | 0.55 (±0.01) | 0.56 (±0.01) | 0.07% (±0.02%) |
| | DP-SGD ($\varepsilon = 8$) | 47.61% (±0.12%) | 0.56 (±0.00) | 0.56 (±0.01) | 0.56 (±0.01) | 0.08% (±0.01%) |
| | PD-SGD (param setting 1) | 64.83% (±0.05%) | 0.63 (±0.01) | 0.72 (±0.01) | 0.64 (±0.01) | 0.06% (±0.01%) |
| | PD-SGD (param setting 2) | 61.16% (±0.07%) | 0.61 (±0.01) | 0.59 (±0.02) | 0.60 (±0.01) | 0.06% (±0.01%) |

8 shadow models for S-Attack, R-Attack and C-Attack. For the C-Attack, we use the online version of it and adopted from privacy meter.[9] When evaluating attacks, we always use balanced evaluation dataset (50% member and 50% non-member). When reporting (balanced) accuracy, we always select the threshold with the highest attack accuracy.

**Details for Defenses:** We keep the same parameter setting for all other empirical defense mechanisms as SELENA's original code-base and HAMP original code-base. For DP-SGD, we set the clipping threshold to 1 and use the same batch size as PD-SGD and SGD. We also perform a hyperparameter search to identify the best learning rate for every run.

# G    Additional Experiments

## G.1    Hyperparameter settings and Full Experimental Results

We show hyperparameter settings in Table 4 and the full experimental results in Table 5.

---

[9]`https://github.com/privacytrustlab/ml_privacy_meter/tree/`
`173d4ad80f183ae6e1867b2793dfffe0633107d0/benchmark`

## G.2 Understanding parameters of PD-SGD

Table 6: **Impact of $\gamma$**

| $\gamma$ | Test Acc | Reject Rate | Best Attack |
|---|---|---|---|
| 1 | 92.78% | 99.54% | 0.52 |
| 2 | 94.70% | 30.31% | 0.53 |
| 3 | 94.71% | 13.70% | 0.56 |
| 4 | 94.74% | 5.78% | 0.57 |
| 6 | 94.80% | 2.25% | 0.59 |

Table 7: **Impact of $\sigma$**

| $\sigma$ | Test Acc | Reject Rate | Best Attack |
|---|---|---|---|
| 0.1 | 17.19% | 99.95% | 0.52 |
| 0.15 | 96.02% | 0.15% | 0.54 |
| 0.2 | 95.70% | 0.03% | 0.55 |
| 0.4 | 93.67% | 0.00% | 0.55 |
| 1.0 | 85.23% | 0.00% | 0.56 |

Table 8: **Impact of $T$**

| $T$ | Test Acc | Reject Rate | Best Attack |
|---|---|---|---|
| 1 | 64.78% | 0.00% | 0.76 |
| 2 | 64.81% | 10.17% | 0.75 |
| 3 | 64.76% | 18.86% | 0.71 |
| 5 | 62.66% | 84.68% | 0.64 |
| 7 | 3.21% | 99.90% | 0.50 |

Recall that PD-SGD has three primary parameters — $\sigma$, $\gamma$, and $T$ — that control the privacy-utility trade-off. In this section, we discuss how these parameters impact the performance of PD-SGD.

We first fine-tune the ViT model on CIFAR-10 with different $\gamma$ values while keeping all other parameters fixed. The results are presented in Table 6. We observe that as $\gamma$ decreases, the model's test accuracy experiences a slight decline. However, the Best Attack AUC diminishes substantially. Notably, when $\gamma$ decreases from 2 to 1, even though the Best Attack AUC decreases slightly, the reject rate increases sharply to 99.54%, and the test accuracy drops to 92.78%. This suggests that $\gamma = 2$ may be the optimal choice for this parameter setting.

We perform similar experiments with different $\sigma$ values and present the results in Table 7. We observe that when $\sigma$ is large (i.e., $\sigma > 0.2$), the gradients can easily pass the Privacy Test, but the Best Attack AUC remains high, and the model fails to achieve good test accuracy due to the large noise introduced during training. When $\sigma$ is relatively small, although some gradients are rejected, it provides better defense performance (lower Attack AUC). However, if $\sigma$ is too small, such as 0.1, under the same $\gamma$ and $T$, it becomes very difficult for gradients to pass the privacy test, resulting in low test accuracy.

We also test different $T$ values while keeping all other parameters fixed. We train the linear model on Purchase-100 and present the results in Table 8. We observe that as $T$ increases, it becomes harder for gradients to pass the privacy test. Consequently, the reject rate increases, test accuracy decreases, but better defense performance is achieved (lower Attack AUC).

Therefore, based on these tables and results, we find that the observations corroborate our findings in Fig. 4. This demonstrates that PD-SGD can provide a wide range of privacy-utility trade-offs through different parameter settings. On the other hand, to achieve a better privacy-utility trade-off, it is advisable to tune all three parameters together rather than adjusting only one parameter.

Table 9: **Impact of batch size on Purchase-100 and CIFAR-10**

| Dataset | Batch size | Number of Batches | Test Acc | Reject Rate | Best Attack |
|---|---|---|---|---|---|
| Purchase-100 | 1024 | 24 | 0% | 100% | 0.5 |
| | 2048 | 12 | 60.10% | 88.94% | 0.62 |
| | 3072 | 8 | 64.76% | 10.41% | 0.73 |
| | 4096 | 6 | 64.80% | 9.06% | 0.74 |
| | 5120 | 4 | 64.73% | 0% | 0.77 |
| CIFAR-10 | 1024 | 29 | 60.24% | 55.85% | 0.51 |
| | 2048 | 14 | 74.35% | 37.63% | 0.51 |
| | 3072 | 9 | 80.40% | 20.07% | 0.53 |
| | 4096 | 7 | 80.59% | 14.07% | 0.53 |
| | 5120 | 5 | 81.57% | 7.41% | 0.54 |

## G.3 Understanding Batch size in PD-SGD

The batch size plays an important role in terms of privacy. There are extreme edge cases that are unrealistic, where the batch size is the entire training set or the batch size is a single example. For more realistic batch sizes, there are several tradeoffs, and ultimately, the behavior also depends on the chosen privacy parameters.

We conduct experiments on Purchase-100 and CIFAR-10 and report results in Table 9. Results show that as batch size increases, the rejection rate typically decreases — larger batches more easily pass the privacy test (for fixed privacy parameters). However, this does not necessarily translate into better privacy protection, as the potential for individual sample contributions to still be inferred remains.

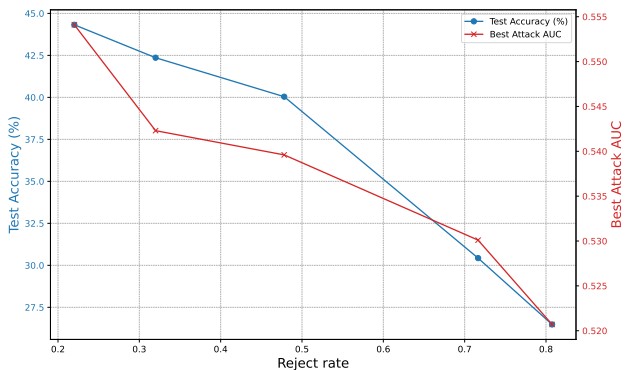

Figure 5: **Privacy-utility trade-off with fixed Reject rate:** We show the privacy utility trade-off with fixed reject rate by using adaptive $\gamma$ during training. We can observe that with a higher reject rate, defense performance is improved, but utility is lower.

Moreover, we found that adjusting other parameters — e.g., $\sigma$, $\gamma$, and threshold can help mitigate these effects, maintaining a balance between utility and privacy across varying batch sizes. For example, for a batch size of 1024, if we double $\gamma$, we can decrease the reject rate to 56.98% and achieve a test accuracy of 63.87% with Best Attack AUC of 0.68.

### G.4 Understanding Reject Rate of PD-SGD

The rejection rate is strongly correlated with the level of privacy protection achievable through PD-SGD. In the extreme case where PD-SGD rejects all updates, we gain perfect privacy at the cost of zero utility. Conversely, if no updates are ever rejected, utility may be high but at the risk of increased privacy leakage. Therefore, it is crucial to find an appropriate rejection rate.

In our previous experiments, the rejection rate was determined solely by the privacy parameters and could not be predicted before training. To better understand its relationship with privacy, we adaptively adjust $\gamma$ to maintain a desired rejection rate set prior to running the experiments. Specifically, during training, we update $\gamma$ based on the rejection rate observed over the most recent updates, using an exponential moving average to increase or decrease $\gamma$.

Furthermore, instead of counting every update, we only track the number of successful updates, ensuring that the model trains for the same number of effective steps regardless of the rejection rate. We apply this approach under the same experimental settings described in Appendix G.12, training a WRN-28-2 model from scratch on CIFAR-100. We show the results in Fig. 5. We observe that, as expected, a higher rejection rate improves defense performance, reflected in a lower best-attack AUC, but also reduces utility (test accuracy). Notably, even when operating at lower accuracy levels, PD-SGD surpasses a conventional DP-SGD baseline with $\varepsilon = 500$. While that baseline achieves 30.55% test accuracy and an attack AUC of 53.84%, certain PD-SGD configurations (e.g., a reject rate near 0.6) maintain higher accuracy while further reducing the attack AUC.

### G.5 Rejection of Anomalous Batches

How do we know that PD-SGD rejects gradient updates from anomalous batches and only those from anomalous batches? We intentionally generate anomalous batches to evaluate this by flipping the labels of a subset of examples ("poisoned examples") and grouping them into a single batch with other normal samples. We ensure that throughout training, the poisoned examples are in the same "anomalous" batch. We then record the rejection rates when the anomalous batch is chosen as seed and when other batches are chosen as seed, for varying proportions of poisoned examples.

Results are shown in Fig. 6, where we observe that for normal batches the rejection rate remains consistently low, as expected and desired. This means that the privacy test does not discard updates unnecessarily. However, when the anomalous batch is selected as seed, the rejection rate increases significantly and quickly plateaus near 100% as the proportion of poisoned examples increases. This

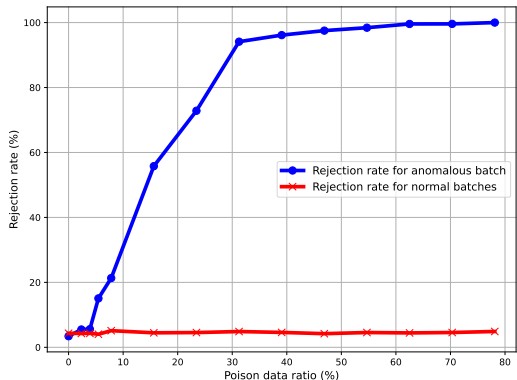

Figure 6: **Rejection rate for anomalous and normal batches.** Rejection rate for the anomalous batch increases to close to 100% as the proportion of poisoned examples increases, while the rate for normal batches remains stable. This suggests that, as desired for privacy and utility, only those gradient updates that may cause privacy leaks are rejected.

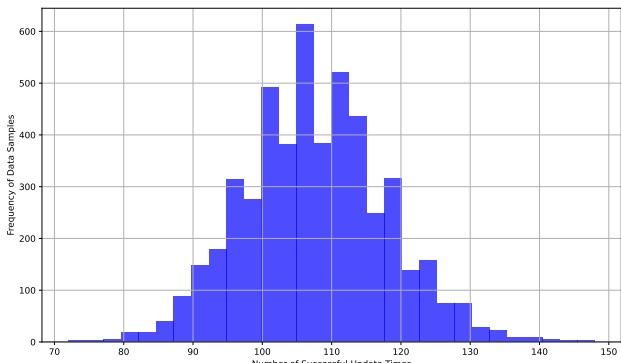

Figure 7: **Distribution of data samples' successful update:** Histogram of all training data samples' successful update. The average count is 107.53 ($\pm 10.27$) and the min and max are 72 and 148, respectively.

indicates that PD-SGD effectively identifies and rejects anomalous batches, preventing the model parameters from being updated in such cases.

### G.6 Frequency of examples used of PD-SGD

Since PD-SGD works by rejecting implausible gradient updates, some training set examples may be used more frequently to update parameters than others. To investigate this, we record the successful update counts for each data sample in the training set in a case where parameters are set to achieve roughly 15% reject rate. We show this distribution in Fig. 7. We can observe that, as expected, there is a range of update frequencies. However, no training set example is used fewer than 72 times, so no example is systematically excluded from influencing the final model.

### G.7 Comparison to SoTA DP-SGD and HAMP

Although this study is the first to propose PD-SGD and it may not be directly comparable with state-of-the-art (SoTA) DP-SGD methods that have undergone extensive optimization, we nonetheless provide comparative evaluation results. Specifically, we compare PD-SGD against recent SoTA DP-SGD approaches such as those by De et al. [12] and Bu et al. [6]. The detailed results are presented in Table 10. Our findings demonstrate that PD-SGD achieves high utility than these established methods under comparable privacy regimes (as measured by MIA success rates).

We also conducted experiments using a recent empirical defense mechanism, HAMP [9] on the Purchase-100 dataset. We show the results in Table 11. PD-SGD (PS 1) attains higher utility than

Table 10: **Compared to SoTA DP-SGD:** Evaluation of PD-SGD and SOTA DP-SGD: Train WRN-16-4 from scratch on CIFAR-10.

| Method | Test Acc | P-Attack | R-Attack | S-Attack | C-Attack |
|---|---|---|---|---|---|
| Non-private | 87.22% | 0.60 | 0.60 | 0.58 | 0.22% |
| DP-SGD | 63.31% | 0.51 | 0.50 | 0.51 | 0.13% |
| Bu et al. [6] | 63.56% | 0.51 | 0.50 | 0.50 | 0.13% |
| De et al. [12] | 72.17% | 0.52 | 0.50 | 0.51 | 0.12% |
| PD-SGD(PS 1) | 82.22% | 0.53 | 0.52 | 0.51 | 0.19% |
| PD-SGD(PS 2) | 79.69% | 0.53 | 0.50 | 0.51 | 0.15% |

HAMP while delivering comparable privacy. Also PD-SGD (PS 2) provides stronger privacy than HAMP at similar utility.

Table 11: **Evaluations for new baseline HAMP [9]**: Evaluation of PD-SGD and new baseline HAMP on Purchase-100 dataset.

| Method | Test Acc | P-Attack | R-Attack | S-Attack | C-Attack |
|---|---|---|---|---|---|
| Non-private | 68.56% | 0.76 | 0.78 | 0.77 | 0.12% |
| AdvReg | 57.56% | 0.70 | 0.70 | 0.66 | 0.08% |
| SELENA | 64.31% | 0.63 | 0.73 | 0.66 | 0.07% |
| HAMP | 61.66% | 0.64 | 0.74 | 0.64 | 0.07% |
| DP-SGD ($\varepsilon$=8) | 47.61% | 0.56 | 0.56 | 0.56 | 0.08% |
| PD-SGD (PS 1) | 64.83% | 0.63 | 0.72 | 0.64 | 0.06% |
| PD-SGD (PS 2) | 61.16% | 0.61 | 0.59 | 0.60 | 0.06% |

## G.8 Variants of Privacy Tests

Table 12: **Performance of PD-SGD under different privacy tests:** We evaluate different privacy tests on CIFAR-10 and follow the setting as Table 1. We change the $\gamma$ to 2.3 for the integer test and 8.0 for the clique test.

| Privacy Test | $\gamma$ | Test acc | P-attack | R-attack | S-attack | C-attack |
|---|---|---|---|---|---|---|
| Simple | 2.0 | 94.22% | 0.53 | 0.49 | 0.53 | 0.20% |
| Integer | 2.3 | 94.35% | 0.54 | 0.53 | 0.54 | 0.20% |
| Clique | 8.0 | 94.79% | 0.52 | 0.53 | 0.52 | 0.22% |

We discuss privacy test counting methods in Section 4.2. Here, we measure their performance empirically on CIFAR-10 and show the results in Table 12. We find that PD-SGD achieves a similar privacy utility trade-off for different tests.

We also analyze the effect of threshold randomization and ceiling. We consider a "comprehensive" test which uses clique counting and threshold randomization and ceiling, and compare it to the simple (non-randomized, no ceiling) test.

We evaluate this on Purchase-100. Because the threshold is now perturbed, we must raise it to offset the added noise, so we decrease the batch size to 500 to increase the total number of batches to 50 and set $T = 20$. We use the geometric mechanism based on the implementation of differential-privacy-library[10]. We set the $\beta = e$ (i.e., $\varepsilon_0 = 1$) and $\psi = 0.2$. Since we changed the batch size, we reset the $\gamma$ to 3800. We show the performance comparison to the original privacy test in Table 13. We can observe that even after introducing additional randomness in testing, PD-SGD delivers a comparable privacy–utility trade-off once the hyperparameters are properly calibrated.

## G.9 Computational Resources

We evaluate the running time of PD-SGD for one training step. We conduct experiments using CIFAR-10 by fine-tuning the ViT model, and train the WRN-16-4 model from scratch following the same setup as described for Table 1. The time is averaged over three consecutive steps taken from

---

[10]https://github.com/IBM/differential-privacy-library/blob/main/diffprivlib/mechanisms/geometric.py

Table 13: **Performance of PD-SGD under comprehensive privacy tests:** We evaluate different privacy tests on Purchase-100 and follow the setting as Table 1.

| Privacy Test | Test acc | P-attack | R-attack | S-attack | C-attack |
|---|---|---|---|---|---|
| Simple | 61.16% | 0.61 | 0.59 | 0.60 | 0.06% |
| Comprehensive | 62.31% | 0.60 | 0.59 | 0.61 | 0.07% |

Table 14: **Computational Time per step:** We measure the GPU time for SGD, DP-SGD, and our proposed PD-SGD for one step with the same model and the same amount of data. We report the average time among 3 steps. For CIFAR-10 (Fine-tuning), we use ViT model and for CIFAR-10 (From scratch), we train WRN-16-4 from scratch. We can observe that although PD-SGD is slower than SGD, it takes less time than DP-SGD.

| Dataset | Method | Time (ms) |
|---|---|---|
| CIFAR-10 (FT) | DP-SGD | 18.86 ($\pm$0.08) |
| | PD-SGD | 7.70 ($\pm$0.10) |
| | SGD | 0.49 ($\pm$0.03) |
| CIFAR-10 (FS) | DP-SGD | 2492.11 ($\pm$8.06) |
| | PD-SGD | 1780.16 ($\pm$15.72) |
| | SGD | 344.47 ($\pm$0.20) |

Table 15: **GPU Memory Usage Comparison:** Evaluation of GPU memory usage of PD-SGD against DP-SGD and non-private (SGD) training with the Wide-ResNet-16-4 model on CIFAR-10 dataset

| Method | alloc | reserved | peak_alloc | peak_reserved |
|---|---|---|---|---|
| SGD | 37.9 MB | 6562.0 MB | 4918.2 MB | 6562.0 MB |
| DP-SGD | 10345.9 MB | 18612.0 MB | 10345.9 MB | 18612.0 MB |
| PD-SGD | 145.8 MB | 6712.0 MB | 5014.5 MB | 6712.0 MB |

the middle of the training process. For comparison, we also measure the time of standard SGD and DP-SGD under the same conditions. The results are summarized in Table 14. PD-SGD is noticeably slower than standard SGD but notably faster than DP-SGD for a single training step.

We also conducted experiments to measure GPU memory usage of PD-SGD against DP-SGD and non-private (SGD) training with the Wide-ResNet-16-4 model on the CIFAR-10 dataset and show results in Table 15. PD-SGD's memory footprint is slightly higher but comparable to SGD and **far below DP-SGD**. Thus, on memory-constrained GPUs where DP-SGD may exceed capacity, PD-SGD remains viable.

### G.10 Vulnerable Data points recognized by PD-SGD

We hypothesized that data samples most frequently rejected by PD-SGD are inherently more vulnerable to membership inference attacks. To validate this hypothesis, we first identified the six images with the highest rejection rates across three independent PD-SGD trainings. For each of these samples, we ran ten independent per-example MIA trials and recorded the fraction of trials in which the adversary correctly classified the sample as a "member." The resulting per-sample MIA success rates are shown in Fig. 8. Under standard SGD, these six images exhibit an average MIA success rate of 91.67 %, confirming their high vulnerability. When using PD-SGD, however, the average success rate falls to 56.67 %, demonstrating that PD-SGD not only delivers strong per-example privacy protection but also serves as an effective mechanism for detecting the most privacy-sensitive points.

### G.11 Training with more data points

To extend our evaluation, we have finetuned ViT model with a larger subset of CIFAR-100 i.e., using 10K for training and 10K for testing and the rest of the data for shadow datasets. We report results in Table 16. We can observe that without any defense, the attack AUC is around 0.56, while with PD-SGD, the attack AUC decreases to 0.52-0.51. For utility, PD-SGD can achieve 80.2% test accuracy while DP-SGD with $\varepsilon = 8$ can only achieve 77.13% test accuracy.

Table 16: **Evaluate PD-SGD on CIFAR-100 with more training data points:** Finetune Vit model with large subset of CIFAR-100 (10K). We can observe the similar thing that PD-SGD can achieve better privacy utility trade-off than DP-SGD.

| Method | Test Acc | P-Attack | R-Attack | S-Attack | C-Attack |
|--------|----------|----------|----------|----------|----------|
| Non-Private | 82.94% | 0.56 | 0.57 | 0.56 | 0.19% |
| PD-SGD (PS 1) | 80.29% | 0.52 | 0.52 | 0.51 | 0.11% |
| PD-SGD (PS 2) | 78.25% | 0.51 | 0.51 | 0.51 | 0.08% |
| DP-SGD ($\varepsilon = 8$) | 77.13% | 0.52 | 0.54 | 0.52 | 0.13% |

Table 17: **Evaluate PD-SGD on CIFAR-100 for training from scratch:** Train WRN-28-2 from scratch with PD-SGD on CIFAR-100. We can observe that: PD-SGD achieves a better privacy-utility trade-off than DP-SGD even with large $\varepsilon$.

| Method | Test Acc | P-Attack | R-Attack | S-Attack | C-Attack |
|--------|----------|----------|----------|----------|----------|
| Non-Private | 56.27% | 81.71% | 81.91% | 81.85% | 0.37% |
| DP-SGD($\varepsilon = 8$) | 18.24% | 52.29% | 49.58% | 51.03% | 0.11% |
| DP-SGD($\varepsilon = 100$) | 29.50% | 53.04% | 50.57% | 51.76% | 0.12% |
| DP-SGD($\varepsilon = 500$) | 30.55% | 53.84% | 50.88% | 51.84% | 0.14% |
| PD-SGD (PS 1) | 53.63% | 58.80% | 52.81% | 57.46% | 0.15% |
| PD-SGD (PS 2) | 47.07% | 54.27% | 50.56% | 50.00% | 0.12% |

Table 18: **Impact of Clipping Threshold of DP-SGD**

| Clipping Threshold | Test Acc | P-Attack | R-Attack | S-Attack | C-Attack |
|--------------------|----------|----------|----------|----------|----------|
| 0.1 | 93.49% | 0.54 | 0.56 | 0.54 | 0.18% |
| 1 | 93.56% | 0.54 | 0.56 | 0.54 | 0.18% |
| 10 | 93.54% | 0.54 | 0.57 | 0.54 | 0.20% |

## G.12 Train from scratch on CIFAR-100

We used small training set sizes for these experiments to ensure the resulting models would be vulnerable to MIA so that it would be clear if the desired level of protection was indeed achieved. However, we also included other experiments in our paper where we used much larger training set sizes (e.g., Table 1). In addition, we conducted further experiments using a larger subset of CIFAR-100. We follow the experiment setting in [50] which trains a WRN-28-2 from scratch on 25k samples of CIFAR-100. We show the results in Table 17. It can be observed that PD-SGD can successfully defend different MIA attacks for example, Attack AUC is decreased significantly from around 81% to 54% by using parameter setting 2 of PD-SGD.

Compared to DP-SGD, PD-SGD consistently provides substantially better utility, even for large values of $\varepsilon$. For instance, with $\varepsilon = 100$ or $\varepsilon = 500$, DP-SGD achieves only about 30% test accuracy. In contrast, our proposed PD-SGD attains significantly higher accuracy — 47.07% (with a slightly higher Attack AUC) or approximately 37% (with a comparable Attack AUC), as shown in Fig. 5.

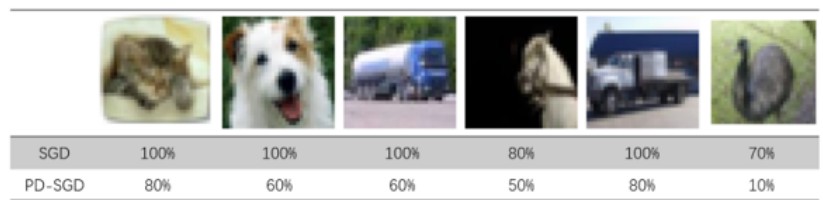

Figure 8: **Samples frequently rejected by PD-SGD face higher MIA risk under standard SGD:** Average per-example membership inference attack success rates are 91.67% but lead to reduced MIA success (56.67%) when using PD-SGD. This shows the usefulness of the method to provide per-example privacy protection, as well as its potential for detecting vulnerable points.

### G.13 Impact of clipping threshold of DP-SGD

To further investigate the impact of the clipping threshold in DP-SGD on privacy protection, we fixed all other parameters and varied the clipping threshold, as shown in Table 18. We can observe that even as the threshold changes, the model's utility and privacy remain almost the same. However, during these experiments, we do find that if the threshold is changed, the learning rate also needs to be tuned properly to get the optimal utility.

## H Proofs

### H.1 Proof of Lemma 1

*Proof of Lemma 1.* We need to prove that if the likelihoods $l_b$ satisfy Definition 1 for some $\lambda \geq 1$ and $\lambda' \geq 0$, then the advantage is bounded:

$$\text{adv} \leq \frac{\lambda - 1}{\lambda + 1} + \frac{\lambda'}{\lambda + 1} .$$

For this first observe that the advantage is bounded by the total variation distance: $\text{adv} \leq TV(l_1, l_0) = \frac{1}{2}|l_1 - l_0|_1$. Therefore, we will show that $\upsilon = TV(l_1, l_0) \leq \frac{\lambda - 1}{\lambda + 1} + \frac{\lambda'}{\lambda + 1}$.

For conciseness, write $p = l_1$ and $q = l_0$. From Eq. (1) we have that for any $x$ in the range of the distribution (of $p$ and $q$):

$$p(x) \leq \lambda q(x) + c(x) \quad \text{and} \quad q(x) \leq \lambda p(x) + c(x) ,$$

for $c(x) \geq 0$ and $\int c(x)dx \leq \lambda'$.

Define $A = \{x : p(x) \geq q(x)\}$. We have $\upsilon = TV(p, q) = \int_A (p - q) = p(A) - q(A)$. Decompose the TV over $A$ and $A^c$:

$$\upsilon = \int_A (p(x) - q(x))dx \leq (\lambda - 1) \int_A q(x)dx + \int_A c(x)dx ,$$

and

$$\upsilon = \int_{A^c} (q(x) - p(x))dx \leq (\lambda - 1) \int_{A^c} p(x)dx + \int_{A^c} c(x)dx .$$

Write $p(S)$, $q(S)$, $c(S)$ to denote the integral over a set $S$ we get: $\upsilon \leq (\lambda - 1)q(A) + c(A)$ and $\upsilon \leq (\lambda - 1)p(A^c) + c(A^c)$. Also, $\upsilon = p(A) - q(A)$, we can plug in $p(A^c) = q(A^c) - \upsilon$ in the second inequality. We obtain:

$$\upsilon \leq (\lambda - 1)(1 - q(A) - \upsilon) + c(A^c) ,$$

which can be reorganized to:

$$\upsilon \leq \lambda^{-1} \left[ (\lambda - 1)(1 - q(A)) + c(A^c) \right]$$

Combining this with the first inequality, we have that:

$$\upsilon \leq \min \left[ (\lambda - 1)q(A) + c(A), \lambda^{-1} \left( (\lambda - 1)(1 - q(A)) + c(A^c) \right) \right] .$$

Writing $y = q(A)$ the above expression upper bounds TV as the minimum of $(\lambda - 1)y + c(A)$ and $\lambda^{-1}[(\lambda - 1)(1 - y) + c(A^c)]$. These can be viewed as lines/functions of $y \in [0, 1]$. Clearly, the maximum is reached when two lines intersect. It can be seen that the intersection point $y^\star$ satisfies:

$$y^\star (\lambda - 1) = \frac{\lambda}{\lambda + 1} \left[ \frac{c(A^c)}{\lambda} - c(A) + \frac{\lambda - 1}{\lambda} \right] .$$

Replacing the RHS in the first inequality yields:

$$\upsilon \leq \frac{\lambda - 1}{\lambda + 1} + \frac{c(A) + c(A^c)}{\lambda + 1} \leq \frac{\lambda - 1}{\lambda + 1} + \frac{\lambda'}{\lambda + 1} ,$$

which completes the proof. $\qquad \square$

## H.2 Proofs of Privacy Testing

*Proof of Lemma 2.* Because $T - \tau$ decreases with $\tau$, the event $\{z \geq T - \tau\}$ becomes easier as $\tau$ grows; hence

$$p_{\tau+1} \geq p_\tau.$$

**Upper ratio.** The geometric tail satisfies $\Pr[z \geq k] = \frac{\beta}{\beta+1}\beta^{-k}$, so for $p_{\tau+1} < 1 - \psi$ we have

$$\frac{p_{\tau+1}}{p_\tau} = \frac{\beta^{-(T-\tau-1)}}{\beta^{-(T-\tau)}} = \beta.$$

If $p_{\tau+1} = 1 - \psi$, then $p_{\tau+1}/p_\tau \leq 1 < \beta$

**Lower bound on** $(1 - p_{\tau+1})/(1 - p_\tau)$**.** Based on upper ratio, we have

$$\frac{1 - p_{\tau+1}}{1 - p_\tau} \geq \frac{1 - p_{\tau+1}}{1 - \frac{p_{\tau+1}}{\beta}}$$

Since $\frac{1 - p_{\tau+1}}{1 - \frac{p_{\tau+1}}{\beta}}$ is monotonically decreasing and $p_{\tau+1} \leq 1 - \psi$, We get:

$$\frac{1 - p_{\tau+1}}{1 - \frac{p_{\tau+1}}{\beta}} \geq \frac{\psi}{1 - \frac{1 - \psi}{\beta}}$$

$\square$

*Proof of Lemma 3.* For $1 \leq t < T$:

$$p_t = (1 - \psi)\Pr(z \geq T - t) = (1 - \psi)\frac{\beta - 1}{\beta + 1}\sum_{i \geq T-t}\beta^{-i}$$

$$\leq (1 - \psi)\beta^{-(T-t)} = (1 - \psi)\exp\left(-\varepsilon_0(T - t)\right).$$

$\square$

## H.3 PD Criterion for PD-SGD

*Proof of Lemma 4.* Take $\mathcal{T}$ as in Lemma 4, that is the algorithm maps onto a single training iteration of Algorithm 1 with a privacy test with parameters $\alpha > 1$, randomization $\mathrm{Geom}(\beta)$ and ceiling $1 - \psi$.

We need to prove that $\mathcal{T}$ satisfies the PD criterion (Definition 1) with $\lambda$, $\lambda'$ as in the lemma. The pointwise bounds need to hold for any partition $B_1, \ldots, B_m$, target batch $B_t$, and any $g \in \mathrm{Range}(\mathcal{T})$.

Consider an arbitrary partition $\mathfrak{B} = (B_1, \ldots, B_m)$ and target batch $B^\star$. Let $\mathfrak{B}' = (B_1, \ldots, B_m, B^\star) = \mathfrak{B}, B^\star$. The likelihood terms are $l_1(g) = \Pr_\mathcal{T}(g|\mathfrak{B}, B^\star)$ and $l_0(g) = \Pr_\mathcal{T}(g|\mathfrak{B})$.

PD-SGD sometimes rejects a gradient. We formalize this as the algorithm outputting $\bot$. Its range is $\mathbb{R}^d \cup \{\bot\}$ for a $d$ parameters model. Consider the case for an arbitrary $g \in \mathbb{R}^d$ ($g \neq \bot$) first.

To produce an output, Algorithm 1 first selects a seed batch uniformly random, then produces $g$ by adding noise to the seed batch gradient, and finally runs the privacy test. Therefore:

$$\Pr_\mathcal{T}(g|\mathfrak{B}) = \sum_{B \in \mathfrak{B}}\Pr(B) \cdot \Pr(g|B, \mathfrak{B}) = \sum_{B \in \mathfrak{B}}\Pr(B) \cdot p(g|B) \cdot p_{\tau(g,B,\mathfrak{B})}.$$

Here $\Pr(B)$ is the probability of $B$ being selected as seed, $\Pr(g|B, \mathfrak{B})$ is the probability of outputting $g$ conditional on $B$ being the seed batch (and $\mathfrak{B}$ is the partition). The latter term, $\Pr(g|B, \mathfrak{B})$, requires that $g$ is sampled from batch $B$'s gradient ($p(g|B)$) and that the test passes $p_{\tau(g,B,\mathfrak{B})}$. We

use $\tau = \tau(g, B, \mathfrak{B})$ to denote the number of alternatives. This is because the probability of passing the test only depends on $\tau$ (Lemmas 2 and 3).

Breaking down the sum for $l_1(g)$ over the seed batches, we have:

$$l_1(g) = \frac{1}{m+1}\left[\sum_{B \in \mathfrak{B}} \Pr(g|B, \mathfrak{B}') + \Pr(g|B^\star, \mathfrak{B}')\right]$$

$$\geq \frac{1}{m+1}\sum_{B \in \mathfrak{B}} \Pr(g|B, \mathfrak{B}') = \frac{1}{m+1}\sum_{B \in \mathfrak{B}} p(g|B)\, p_{\tau(g,B,\mathfrak{B}')}$$

$$\geq \frac{1}{m+1}\sum_{B \in \mathfrak{B}} p(g|B)\, p_{\tau(g,B,\mathfrak{B})} = \frac{1}{m+1}\sum_{B \in \mathfrak{B}} \Pr(g|B, \mathfrak{B})$$

$$= \frac{m}{m+1}l_0(g)\,.$$

The first inequality uses $\Pr(g|B^\star, \mathfrak{B}') \geq 0$ and the second uses the fact that adding a batch does not decrease the probability of passing the test for other batches.

This gives the first direction: $l_1(g)/l_0(g) \geq \frac{m}{m+1}$.

For the second direction, we proceed similarly:

$$l_1(g) = \frac{1}{m+1}\left[\sum_{B \in \mathfrak{B}} \Pr(g|B, \mathfrak{B}') + \Pr(g|B^\star, \mathfrak{B}')\right]$$

$$\leq \frac{1}{m+1}\left[\beta\sum_{B \in \mathfrak{B}} \Pr(g|B, \mathfrak{B}) + \Pr(g|B^\star, \mathfrak{B}')\right]$$

$$= \frac{1}{m+1}(\beta\, m\, l_0(g) + \Pr(g|B^\star, \mathfrak{B}'))\,. \tag{†}$$

The inequality uses the fact that the rate of increase of the probability of passing the test is at most $\beta$ when adding a batch (Lemma 2), thus: $\Pr(g|B, \mathfrak{B}') \leq \beta\Pr(g|B, \mathfrak{B})$.

To proceed from (†), we divide the analysis into two cases based on $\tau(g, B^\star, \mathfrak{B}')$., i.e., the number of alternatives available to $B^\star$.

Case 1 ($\tau(g, B^\star, \mathfrak{B}') > t$): We have $\tau(g, B^\star, \mathfrak{B}) \geq t$ since $t + 1$ alternatives in $\mathfrak{B}'$ means at least $t$ such batches exist in $\mathfrak{B}$. Thus (using the definition of $\alpha$-similarity):

$$p(g|B^\star) \leq \frac{\alpha}{t}\sum_{\substack{B \in \mathfrak{B} \\ \tau(g,B,\mathfrak{B}) \geq t}} p(g|B)\,.$$

For each alternative batch $B$ in this sum, we also have that: $p_{\tau(g,B,\mathfrak{B})} \geq \beta^{-1}p_{\tau(g,B^\star,\mathfrak{B}')}$ (Lemma 2). Therefore:

$$\Pr(g|B^\star, \mathfrak{B}') \leq \frac{\alpha\beta}{t}\sum_{\substack{B \in \mathfrak{B} \\ \tau(g,B,\mathfrak{B}) \geq t}} \Pr(g|B, \mathfrak{B}) \leq \frac{\alpha\beta}{t}\sum_{B \in \mathfrak{B}} \Pr(g|B, \mathfrak{B}) = \frac{\alpha\beta}{t}m\, l_0(g)\,.$$

So for case 1, we have: $l_1(g) \leq \frac{m}{m+1}\beta(1 + \frac{\alpha}{t})l_0(g)$.

Case 2 ($\tau(g, B^\star, \mathfrak{B}') \leq t$): In this case, there are fewer than $t$ alternatives in $\mathfrak{B}$ (there could be none). But (by Lemma 3) the probability of passing the test is exponentially small in $T - t$. That is:

$$p_\tau(g, B^\star, \mathfrak{B}') \leq p_t \leq (1 - \psi)\beta^{-(T-t)}\,.$$

Thus: $\Pr(g|B^\star, \mathfrak{B}') \leq p(g|B^\star)(1 - \psi)\beta^{-(T-t)}$ from which we get (for case 2):

$$l_1(g) \leq \frac{m}{m+1}\beta\, l_0(g) + \frac{1}{m+1}\lambda'(g)\,,$$

where $\lambda'(g) = p(g|B^\star)(1 - \psi)\beta^{-(T-t)}$.

Observe that for any $S \subseteq \mathbb{R}^d$: $\int_S \lambda'(g) \leq (1 - \psi)\beta^{-(T-t)}$ since $\int_S p(g|B^\star) \leq 1$.

It remains to consider the case $g = \perp$.

For this observe that since the test has a ceiling $1 \geq l_b(\perp) \geq \psi > 0$ for $b = 0, 1$. We have:

$$\frac{(m+1)\, l_1(\perp)}{m\, l_0(\perp)} = \frac{\sum_{B \in \mathfrak{B}} \Pr(\perp|B, \mathfrak{B}') + \Pr(\perp|B^\star, \mathfrak{B}')}{\sum_{B \in \mathfrak{B}} \Pr(\perp|B, \mathfrak{B})}\,.$$

Write $s(\mathfrak{B}) = \sum_{B \in \mathfrak{B}} \Pr(\perp|B, \mathfrak{B})$, $s(\mathfrak{B}') = \sum_{B \in \mathfrak{B}} \Pr(\perp|B, \mathfrak{B}')$ and $s^\star = \Pr(\perp|B^\star, \mathfrak{B}')$. From Lemma 2 we get that $1 \geq \frac{s(\mathfrak{B}')}{s(\mathfrak{B}')} \geq \frac{\beta\psi}{\beta - 1 + \psi}$. We also have that for any batch $B$ and partition $\mathfrak{B}$: $1 \geq \Pr(\perp|B, \mathfrak{B}) \geq \psi$. Thus $1 \geq s^\star \geq \psi$ and also $s(\mathfrak{B}) \geq m\psi$. It follows that:

$$\frac{\beta\psi}{\beta - 1 + \psi} + \frac{\psi}{m} \leq \frac{(m+1)\, l_1(\perp)}{m\, l_0(\perp)} \leq 1 + \frac{1}{m\psi}\,.$$

Dividing by $\frac{m}{m+1}$ yields:

$$\frac{m}{m+1}\left(\frac{\beta\psi}{\beta - 1 + \psi} + \frac{\psi}{m}\right) \leq \frac{l_1(\perp)}{l_0(\perp)} \leq \frac{m}{m+1}\left(1 + \frac{1}{m\psi}\right) = \frac{m + \psi^{-1}}{m+1}\,.$$

Aggregating the bounds over all the cases completes the proof. $\square$

## H.4 PD-SGD satisfies DP

The proof of Lemma 5 is similar to that of Lemma 4, except that it must consider a random partition and consequently there is no additional target batch $B_t$, so we have to couple the partitions under $D_1$ and $D_2$ and leverage the observation that the additional example $(x, y)$ such that $D_1 = D_2 \cup \{(x, y)\}$ (or $D_2 = D_1 \cup \{(x, y)\}$) falls into exactly one batch.

*Proof of Lemma 5.* Consider adding an example $(x, y)$ to $D$. Let $D' = D \cup \{(x, y)\}$. Observe that no matter how $D'$ gets partitioned (assuming only $m$ batches) the example $(x, y)$ only falls into exactly one batch. Furthermore, we can couple partitions on $D$ and $D'$ as follows. If $\mathfrak{B} = (B_1, \ldots, B_m)$ is a partition on $D$ and $D$ has $n$ data points, there are $1 \leq q \leq m$ batches of size $r + 1$ and $m - q$ batches of size $r$ for integer $r$ such that $n = mr + q$. Under $D'$ we can take the partition $\mathfrak{B}$ and obtain a partition $\mathfrak{B}'$ by selecting a uniformly random batch of size $r$ (i.e., one of the $m - q$ batches of size $r$) and adding $(x, y)$ to it. (If $m = q$, i.e., all partitions have size exactly $n/m$ then we pick a uniformly random batch and add $(x, y)$ to it).

Fix an arbitrary partition $\mathfrak{B} = (B_1, \ldots, B_m)$ of $D$ and let $\mathfrak{B}' = (B'_1, \ldots, B'_m)$ denote the associated partition under $D'$. There exists $j$ such that $B'_j \neq B_j$ (and $B'_j$ includes $(x, y)$) and for $i \neq j$: $B_i = B'_i$. Write $B^\star \,(= B'_j)$ to denote the differing batch.

Fix an arbitrary $g \neq \perp$. Since the two partitions have the same number of batches, we can relate the probability of producing $g$ under $\mathfrak{B}$ to that under $\mathfrak{B}'$ by considering the case where the selected seed batch is $B^\star$ or not. When selecting a seed batch, the probability that we select $B^\star$ is exactly $1/m$, therefore:

$$\Pr(g|\mathfrak{B}') = \frac{1}{m}\Pr(g|B^\star, \mathfrak{B}') + \frac{m-1}{m}\Pr(g|B, \mathfrak{B}')\,, \tag{6}$$

We proceed similarly to the proof of Lemma 4. For the first direction, we immediately get the first direction:

$$\Pr(g|\mathfrak{B}') > \frac{m-1}{m}\Pr(g|B, \mathfrak{B}') \geq \frac{m-1}{m}\Pr(g|B, \mathfrak{B}) = \frac{m-1}{m}\Pr(g|\mathfrak{B})$$

where the first inequality used the fact that $\Pr(g|B^\star, \mathfrak{B}') > 0$ and the second used Lemma 2 (the probability of passing the test is non-decreasing).

For the second direction, observe that the term $\Pr(g|B, \mathfrak{B}')$ is related to $\Pr(g|B, \mathfrak{B})$ as follows:

$$\Pr(g|B, \mathfrak{B}') = p(g|B) \cdot p_{\tau(B, \mathfrak{B}')}$$
$$\leq \beta p(g|B) \cdot p_{\tau(B, \mathfrak{B})} = \beta\Pr(g|B, \mathfrak{B})\,,$$

where we again used Lemma 2. Since $\mathfrak{B}'$ differs in only one batch from $\mathfrak{B}$ the number of alternatives increases by at most one and thus the probability of passing the test by a factor of at most $\beta$.

Case 1: $\tau(B^\star, \mathfrak{B}') > t$. In this case for any batch $B \in \mathfrak{B}'$ that is $\alpha$-similar to $B^\star$ we have $\tau(B^\star, \mathfrak{B}) \geq t$ (and there are at least $t$ such batches in $\mathfrak{B} \setminus B_j$). Therefore:

$$\Pr(g|B^\star, \mathfrak{B}') \leq \frac{\alpha\beta}{t} \sum_{B \in \mathfrak{B}: B^\star \simeq_\alpha B} \Pr(g|B, \mathfrak{B})$$

$$\leq \frac{\alpha\beta}{t} \sum_{B \in \mathfrak{B}} \Pr(g|B, \mathfrak{B}) \,,$$

where the first inequality applies the $\alpha$-similarity and Lemma 2 to the average batch $B \in \mathfrak{B}$ that is $\alpha$-similar to $B^\star$. The second inequality simply uses the fact that $\Pr(g|B, \mathfrak{B}) \geq 0$ for any batch $B$.

Since $\Pr(g|\mathfrak{B}) = \frac{1}{m} \sum_{B \in \mathfrak{B}} \Pr(g|B, \mathfrak{B})$:

$$\frac{1}{m} \Pr(g|B^\star, \mathfrak{B}') \leq \frac{1}{m} \frac{\alpha\beta}{t} \sum_{B \in \mathfrak{B}} \Pr(g|B, \mathfrak{B}) = \frac{\alpha\beta}{t} \Pr(g|\mathfrak{B}) \,.$$

Putting this together, we have for case 1 that:

$$\Pr(g|\mathfrak{B}') \leq \frac{\alpha\beta}{t} \Pr(g|\mathfrak{B}) + \frac{m-1}{m} \beta \Pr(g|\mathfrak{B}) \leq \beta \left( \frac{m-1}{m} + \frac{\alpha}{t} \right) \Pr(g|\mathfrak{B}) \,.$$

Case 2: $\tau(B^\star, \mathfrak{B}') \leq t$. In this case, there are less than $t$ alternatives in $\mathfrak{B}$ (e.g., there could be none). However, since the term $\Pr(g|B^\star, \mathfrak{B}')$ is bounded by the probability of passing the test according to Lemma 3. We have:

$$\frac{1}{m} \Pr(g|B^\star, \mathfrak{B}') \leq \frac{1}{m} (1 - \psi) \exp\left(-\varepsilon_0(T - t)\right) p(g|B^\star) \,.$$

Therefore:

$$\Pr(g|\mathfrak{B}') \leq \frac{1}{m} (1 - \psi) \exp\left(-\varepsilon_0(T - t)\right) \cdot p(g|B^\star) + \frac{m-1}{m} \beta \Pr(g|\mathfrak{B}) \,.$$

Since $\frac{m-1}{m}\beta \leq \beta(\frac{m-1}{m} + \frac{\alpha}{t})$, this completes the second direction and we get:

$$\Pr(g|\mathfrak{B}') \leq \beta(\frac{m-1}{m} + \frac{\alpha}{t}) \Pr(g|\mathfrak{B}) + \delta(g, \beta^\star) \,,$$

where $\delta(g, \beta^\star) = \frac{1}{m}(1 - \psi) \exp\left(-\varepsilon_0(T - t)\right) \cdot p(g|B^\star)$.

To meet the definition for both directions, we need to consider an arbitrary set $Y$. It suffices to integrate over the previous results. Since: $\Pr(Y|\mathfrak{B}') = \int_{g \in Y} \Pr(g|\mathfrak{B}')$, it follows that:

$$\frac{m-1}{m} \Pr(Y|\mathfrak{B}) \leq \Pr(Y|\mathfrak{B}') \leq \beta(\frac{m-1}{m} + \frac{\alpha}{t}) \Pr(Y|\mathfrak{B}) + \delta(Y, \beta^\star) \,,$$

where $\delta(Y, \beta^\star) = \frac{1}{m}(1 - \psi) \exp\left(-\varepsilon_0(T - t)\right) \cdot \int_{g \in Y} p(g|B^\star) \leq \frac{1}{m}(1 - \psi) \exp\left(-\varepsilon_0(T - t)\right)$ because $\int_{g \in Y} p(g|B^\star)$ is at most 1 since it is a probability.

Noting that $\Pr(Y|\mathcal{M}(D)) = \frac{1}{|\mathcal{B}(D)|} \sum_{\mathfrak{B} \in \mathcal{B}(D)} \Pr(Y|\mathfrak{B})$ and the number of partitions in the sum from the coupling is the same for $D$ and $D'$ yields the result with $\varepsilon = \ln(\beta(1 + \frac{\alpha}{t}))$ and $\delta \leq \frac{1}{m}(1 - \psi) \exp\left(-\varepsilon_0(T - t)\right)$.

Finally, consider the case $g = \bot$. From the ceiling, we have that for any batch $B$ and partition $\mathfrak{B}$: $1 \geq \Pr(\bot|B, \mathfrak{B}) \geq \psi > 0$. From Lemma 2, we have that for any batch in $B \in \mathfrak{B}$: $1 \geq \Pr(\bot|B, \mathfrak{B}')/\Pr(\bot|B, \mathfrak{B}) \geq \frac{\beta\psi}{\beta - 1 + \psi}$.

Write:

$$\frac{\Pr(\bot|\mathcal{M}(D'))}{\Pr(\bot|\mathcal{M}(D))} = \frac{\sum_{B \in \mathfrak{B}'} \Pr(\bot|B, \mathfrak{B}')}{\sum_{B \in \mathfrak{B}} \Pr(\bot|B, \mathfrak{B})} = \frac{s(\mathfrak{B}') + \Pr(\bot|B^\star, \mathfrak{B}')}{s(\mathfrak{B}) + \Pr(\bot|B_j, \mathfrak{B})} \,, \tag{†}$$

with $s(\mathfrak{B}') = \sum_{B \in \mathfrak{B}':B \neq B^\star} \Pr(\bot|B, \mathfrak{B}')$ and $s(\mathfrak{B}) = \sum_{B \in \mathfrak{B}:B \neq B_j} \Pr(\bot|B, \mathfrak{B})$. Both sums are over the same batches and we have that: $1 \geq \frac{s(\mathfrak{B}')}{s(\mathfrak{B})} \geq \frac{\beta\psi}{\beta - 1 + \psi}$.

From this taking the worst case for the terms involving $B^\star$ and $B_j$ in both directions, we get:

$$\frac{(\frac{\beta\psi}{\beta-1+\psi})s + \psi}{s+1} \leq (\dagger) \leq \frac{s+1}{s+\psi} \,,$$

for $s = s(\mathfrak{B})$. Optimizing as a function of $s \in [(m-1)\psi, (m-1)]$, we see that the upper and lower bounds are reached for $s$ at its maximum and minimum respectively, from which we conclude:

$$\frac{\beta\psi}{(\beta-1)+\psi} \frac{m}{(m-1)+\psi^{-1}} \leq \frac{\Pr(\bot|\mathcal{M}(D'))}{\Pr(\bot|\mathcal{M}(D))} \leq \frac{m}{(m-1)+\psi} \,,$$

which concludes the proof. We get the result as stated in the lemma when the bounds for the $g \neq \bot$ case dominate (those for $g = \bot$). For this, it suffices to choose $\psi$ appropriately. $\qquad \square$

### H.5 Proofs of Appendix D

We now prove Lemma 6.

*Proof of Lemma 6.* Consider the ratio of probabilities bounded by Eq. (2) and expand using the Gaussian PDF. We get:

$$\frac{p(\tilde{g} - g_s)}{p(\tilde{g} - g)} = \frac{\exp\left(-(2\sigma^2)^{-1} \sum_{j=1}^{k} Z_j^2\right)}{\exp\left(-(2\sigma^2)^{-1} \sum_{j=1}^{k} (Z_j + (g_{s,j} - g_{i,j}))^2\right)}$$

$$= \exp\left(-(2\sigma^2)^{-1} \sum_{j=1}^{k} \left[Z_j^2 - (d_j + Z_j)^2\right]\right)$$

$$= \exp\left(-(2\sigma^2)^{-1} \left[-d - 2\sum_{j=1}^{k} d_j Z_j\right]\right) \,,$$

where $d_j = g_{s,j} - g_{i,j}$ and $d = \sum_{j=1}^{k} d_j^2 = ||g_s - g_i||_2^2$.

Plugging this into the inequality, taking the log and some reorganization we get that the candidate gradient is plausibly deniable with respect to $g_i$ iff:

$$-\frac{\gamma\sigma}{\sqrt{d}} \leq \frac{\sqrt{d}}{2\sigma} + \sum_{j=1}^{k} \frac{d_j}{\sqrt{d}} \frac{Z_j}{\sigma} \leq \frac{\gamma\sigma}{\sqrt{d}} \,.$$

Since $Z_j \sim \mathcal{N}(0, \sigma^2)$, the summand for $j$ is distributed as $\mathcal{N}(0, d^{-1}d_j^2)$. Further, since the sum of i.i.d. Gaussian random variable is distributed a Gaussian random variable with the sum of the means and the sum of the variance, we recognize that $Y = \sum_{j=1}^{k} \frac{d_j}{\sqrt{d}} \frac{Z_j}{\sigma} \sim \mathcal{N}(0, 1)$.

Thus reducing the plausibility of a candidate gradient to:

$$\frac{\sqrt{d}}{2\sigma} - \frac{\gamma\sigma}{\sqrt{d}} \leq Y \leq \frac{\gamma\sigma}{\sqrt{d}} + \frac{\sqrt{d}}{2\sigma} \,, \tag{7}$$

and further to

$$\frac{d - 2\gamma\sigma^2}{2\sigma\sqrt{d}} \leq Y \leq \frac{d + 2\gamma\sigma^2}{2\sigma\sqrt{d}} \tag{8}$$

where we have used symmetry so that $-Y$ has the same distribution as $Y$.

Therefore, $Y$ needs to be within a band of width $\frac{\tilde{\gamma}}{\sigma\sqrt{d}}$ around $\sqrt{d}/2\sigma$ where $\tilde{\gamma} = 2\sigma^2\gamma$, which completes the proof. $\qquad \square$

The proof of Lemma 7 relies on the following standard normal upper and lower tail bounds:

**Lemma 8.** *Let $X \sim N(0,1)$. For $t > 0$, we have:*

$$\frac{t}{t^2 + 1}(\sqrt{2\pi})^{-1}\exp\left(-t^2/2\right) < \Pr(X > t) < (t\sqrt{2\pi})^{-1}\exp\left(-t^2/2\right).$$

Note that tighter bounds are available ([11, 15]).

*Proof of Lemma 7.* Let $a = \frac{\sqrt{d}}{2\sigma}$ and $b = \frac{\gamma\sigma}{\sqrt{d}}$. We have from Lemma 6 that $q(s,i) = \Pr(a - b \leq X \leq a + b)$ for $X \sim N(0,1)$. Thus:

$$q(s,i) = \Pr(X > a - b) - \Pr(X > a + b)$$

$$< \frac{1}{(a-b)\sqrt{2\pi}}e^{-(a-b)^2/2} - \frac{(a+b)}{((a+b)^2+1)\sqrt{2\pi}}e^{-(a+b)^2/2}$$

$$= \frac{1}{\sqrt{2\pi}}\left[\frac{1}{a-b}e^{-(a-b)^2/2} - \frac{(a+b)}{(a+b)^2+1}e^{-(a+b)^2/2}\right]$$

$$= \frac{e^{\frac{-(a^2+b^2)}{2}}}{\sqrt{2\pi}}\left[\frac{e^{ab}}{a-b} - \frac{(a+b)}{(a+b)^2+1}e^{-ab}\right].$$

Substituting back $a$ and $b$ in terms of $d, \sigma, \gamma$ yields the result. $\qquad\square$

The following corollary of the lemma provides a simple upper bound whenever $d > \tilde{\gamma}$.

**Corollary 1.** *Let $d \geq \frac{\tilde{\gamma}}{f}$ for some $0 < f < 1$. Then:*

$$q(d) < \frac{e^{-\left(\frac{d}{8\sigma^2} + \frac{\gamma^2\sigma^2}{2d}\right)}}{\sqrt{2\pi d}}2\sigma\left[\frac{e^{\gamma/2}}{1-f} - \frac{e^{-\gamma/2}}{2+f}\right] \qquad (9)$$

*Proof of Corollary 1.* Let $d \geq 2\gamma\sigma^2$ which implies $a - b \geq 0$. When $d$ increases, $a$ increases but $b$ decreases. So, we can bound $a - b$ and $a + b$ as follows:

Suppose $b \leq fa$ where $0 \leq f < 1$ and $a > 1$, then

$$\frac{1}{a-b} \leq \frac{1}{a(1-f)}$$

$$\frac{a+b}{(a+b)^2+1} \geq \frac{1}{a(2+f)}$$

Based on this, we can get:

$$q(s,i) < \frac{e^{\frac{-(a^2+b^2)}{2}}}{\sqrt{2\pi}}\left[\frac{e^{ab}}{a-b} - \frac{(a+b)}{(a+b)^2+1}e^{-ab}\right]$$

$$< \frac{e^{\frac{-(a^2+b^2)}{2}}}{\sqrt{2\pi}a}\left[\frac{e^{ab}}{1-f} - \frac{e^{-ab}}{2+f}\right].$$

Observe that $ab = \gamma/2$, $a^2 = \frac{d}{4\sigma^2}$, $b^2 = \frac{\gamma^2\sigma^2}{d}$

So:

$$q(s,i) < \frac{e^{-\left(\frac{d}{8\sigma^2} + \frac{\gamma^2\sigma^2}{2d}\right)}}{\sqrt{2\pi d}}2\sigma\left[\frac{e^{\gamma/2}}{1-f} - \frac{e^{-\gamma/2}}{2+f}\right].$$

$\qquad\square$

