# OpenReview forum: "Deep Learning with Plausible Deniability"
_NeurIPS.cc/2025/Conference — NeurIPS 2025 poster_

### Official Review · Reviewer_t2cw · 2025-06-24

**Clarity:** 2
**Significance:** 2
**Originality:** 2
**Rating:** 2
**Confidence:** 4

**Summary:**

The paper introduces Plausibly Deniable Stochastic Gradient Descent (PD-SGD), a training procedure that attaches a mini-batch–level “plausible deniability” test to every stochastic-gradient update. Concretely, the seed batch’s gradient is first perturbed with isotropic Gaussian noise; the update is accepted only if at least T other batch gradients fall within a log-density band of width γ around the noisy seed gradient, otherwise the step is rejected altogether. This rejection-sampling view allows the authors to derive an differential-privacy guarantee under mild conditions while avoiding per-example clipping and thus supporting non-decomposable losses. Three hyper-parameters—noise scale, band width, neighbour count—govern the privacy–utility trade-off and can be tuned heuristically by monitoring the rejection rate.

**Questions:**

See the weakness.

**Ethical Concerns:**

["NO or VERY MINOR ethics concerns only"]

**Final Justification:**

Despite the additional experiments and clarifications in the rebuttal, the core contributions remain insufficiently substantiated. The proposed “plausible deniability” privacy analysis offers only a superficial repackaging of standard rejection-sampling and advanced-composition techniques, without any novel proof strategies or non-trivial theoretical lemmas to distinguish it from existing DP-SGD work. The computational overhead of the privacy test—potentially requiring gradient evaluations across all mini-batches—remains unbounded and is not accompanied by any formal complexity guarantees or comprehensive runtime benchmarks, especially in distributed or multi-node settings. Furthermore, the empirical evaluation is limited to black-box membership inference and omits stronger white-box or adaptive attacks that would directly challenge the method’s core defense mechanism. Finally, reporting only GPU memory usage without wall-clock timing or communication cost analysis leaves open the question of whether any practical training speedup is achievable. In light of these unresolved theoretical and practical shortcomings, the manuscript does not meet the bar for publication.

**Limitations:**

See weakness

**Quality:**

3

**Strengths And Weaknesses:**

Strengths:
1. PD-SGD sounds to be conceptually novel: by enforcing a batch must be indistinguishable from several others before its gradient is used, it turns plausible deniability into a practical optimisation rule.
2. Experiments on CIFAR-10/100 (Vision Transformers and Wide-ResNets) and Purchase-100 (linear model) show that PD-SGD matches or exceeds the test accuracy of empirical defenses (AdvReg, SELENA) while lowering four standard membership-inference attack scores and vastly outperforming DP-SGD at comparable privacy budgets.
3. The logic of this paper is clear, the explanatory figures can help readers understand, and the literature review is comprehensive.The experimental section of this paper provides a wealth of details and ablation studies, which enhances its credibility.

Weaknesses:
The paper lacks persuasiveness in some places.
1. Theoretical contributions. The author claims that “to the best of our knowledge, no existing defense mechanism simultaneously offers a theoretically justified guarantee and maintains good model utility.”, which seems to show the significance of this work. However, I suspect that the author has overstated the theoretical contributions of the paper. Although the author elaborates on the theoretical significance in Section 4.1, I would like to know further whether the theory of this paper employs any inspiring mathematical tools or techniques that can distinguish its theoretical contributions from other papers? Otherwise, I will question the novelty of the theory of this paper. Also, the theoretical derivations in the appendix seems to be relatively straightforward, making it difficult to justify the theoretical novelty.
2. Experiment. The author seems not to have provided comparisons of the method in terms of computational resources and memory. Each iteration still requires computing gradients for all mini-batches to perform the plausibility test, which the authors acknowledge and evaluate in Appendix F.4, implying non-trivial time and energy overhead on large or distributed workloads.
3. Scalability. Empirical evaluation focuses on black-box membership-inference attacks against medium-scale vision and tabular models; robustness to stronger white-box or property-inference attacks, and scalability to very large language or generative models, remain untested

---

> ### Author Rebuttal · Authors · 2025-07-30
>
> We thank the reviewer for the feedback and recognizing the empirical contributions of our proposed method. Regarding the identified weaknesses, we believe they stem from miscommunication or misunderstandings. Below, we provide clarifications.
>
> **1.Theoretical contributions:**
>
> Our theoretical contributions include the proposed definition of plausible deniability as applied to mini-batches and the PD-SGD algorithm, which are novel. Furthermore, the paper is the first to show that a privacy test can be used to achieve differential privacy guarantees in this setting.
>
> More precisely: PD‑SGD is a full training‑time mechanism built around a plausibility test on noisy mini‑batch gradients: an update is kept only if the seed batch has at least $T$ $\alpha$‑similar alternatives. We show that the test’s success probability depends on the $l_2$ distance between gradients and that outlier updates are exponentially suppressed. We also provide two bounded sensitivity (sensitivity=1) test variants (Integer/Bins and Stable/Clique) so adding/removing one batch changes the count by at most 1. We further show that if the test’s threshold is appropriately randomized (e.g., with geometric noise and a ceiling probability), the construction of batch-level deniability yields per data point $(\varepsilon,\delta)$-DP for a single step. From there one can apply advanced composition to obtain end‑to‑end guarantees.
>
> If nothing else, the paper shows a different way to achieve DP‑level protection without per data point gradient clipping. This is noteworthy as it shows a departure from existing approaches for DP in this setting which largely follow DP-SGD in using gradient clipping to bound sensitivity. It is also worth repeating that our method is compatible with non-decomposable losses unlike DP-SGD. It can therefore be applied in cases where DP-SGD cannot.
>
> **2. Comparisons of the method in terms of computational resources and memory**
>
> PD‑SGD does not require traversing all mini‑batches at each step: the privacy test stops as soon as it finds $T$ $\alpha$‑similar batches.
>
> Moreover, PD-SGD does not require storing gradients long term. In our implementation of the privacy test, we temporarily store the noisy gradients of the seed batch and then compute and store (one at a time) the gradients for alternative batches to perform the counting as described in Alg 1. Once we know if a batch counts as an alternative, its gradient is no longer needed and we discard it. This avoids an unnecessary increase in memory usage.
>
> Appendix F.4 provides measurements of training time. Here we provide measurements of memory usage:
>
>
> **Table:** GPU Memory Usage Comparison
>
> | Method | alloc    | reserved   | peak_alloc | peak_reserved |
> |--------|----------|------------|------------|---------------|
> | SGD    | 37.9 MB  | 6562.0 MB  | 4918.2 MB  | 6562.0 MB     |
> | DP-SGD | 10345.9 MB | 18612.0 MB | 10345.9 MB | 18612.0 MB    |
> | PD-SGD | 145.8 MB | 6712.0 MB  | 5014.5 MB  | 6712.0 MB     |
>
> Observe that PD-SGD has similar memory usage as SGD (slightly higher but comparable) and uses much less memory compared to DP-SGD.
>
> PD-SGD can be applied to large models as we show below. In particular we can leverage standard engineering tricks for example using low‑rank / per‑layer sketches instead of full gradients.
>
> **3. Scalability**
>
> To directly address scalability, we added a large‑model experiment: fine‑tuning LLaMA‑2‑7B on SST‑2 with PD‑SGD. Our run reaches 94.76% test accuracy while maintaining stable memory use (peak alloc memory: 6,753.48 MB). It is comparable to 94.8% reported in Table 12 of Zhao et al. [4] with normal training with LLaMA-2-7B and 92.2% with DP-Lora[5] which used another backbone model.  This shows PD‑SGD is practical for LLM fine‑tuning and does not incur prohibitive memory overhead. We will add these results to the paper.
>
>
> **4. Threat model in empirical evaluation**
>
> Our threat model for the evaluation is **black‑box membership inference against the final model.** We acknowledge white‑box and property‑inference evaluations as important extensions for future work. However, our choice and the focus of our paper on the black-box setting is explicitly stated and motivated as consistent with existing defenses we compare against. It is also the dominant evaluation setting for MIA and defenses against it such as Shokri et al. [1], Ye et al.[2], Tang et al. [3].
>
> Extension to white‑box attacks requires exposing per‑step gradients/accept/reject decisions, which our setting intentionally withholds. Nevertheless, PD‑SGD’s **privacy test rejects anomalous gradients** and thus specifically targets the updates that would be most informative under stronger adversaries. In particular, Appendix F shows PD‑SGD selectively rejects intentionally “poisoned/anomalous” batches while preserving normal ones, supporting this intuition and pointing to robustness beyond the evaluated black‑box MIAs. We will expand the discussion to make this connection explicit and plan white‑box or  property-inference attacks as follow‑up work.
>
>
> Reference:
>
> [1] Shokri, Reza, et al. "Membership inference attacks against machine learning models." 2017 IEEE symposium on security and privacy (SP). IEEE, 2017.
>
> [2] Ye, Jiayuan, et al. "Enhanced membership inference attacks against machine learning models." Proceedings of the 2022 ACM SIGSAC conference on computer and communications security. 2022.
>
> [3] Tang, Xinyu, et al. "Mitigating membership inference attacks by {Self-Distillation} through a novel ensemble architecture." 31st USENIX security symposium (USENIX security 22). 2022.
>
> [4] Zhao, Justin, et al. "Lora land: 310 fine-tuned llms that rival gpt-4, A technical report." arXiv preprint arXiv:2405.00732 (2024).
>
> [5] Yu, Da, et al. "Differentially private fine-tuning of language models."  ICLR 2022.

---

> > ### Comment · Reviewer_t2cw · 2025-08-07
> >
> > Thank you for your detailed rebuttal. I appreciate the clarifications, but I remain concerned that several key issues are still unresolved.
> >
> > Your “plausible deniability” framework closely parallels existing rejection-sampling and advanced-composition techniques. The rebuttal restates the high-level idea but does not introduce any new proof insights or non-trivial lemmas that set it apart from prior DP-SGD work. Early stopping helps in practice, but the worst-case per-step cost still scales with the number of mini-batches. Although you report memory usage, there is still no formal complexity bound or comprehensive timing breakdown (especially in distributed settings). All experiments remain in the black-box membership-inference regime. Since the method aims to filter “anomalous” gradients, it is crucial to assess stronger white-box or adaptive adversaries who observe accept/reject decisions—yet no such results are provided. The memory footprint may match standard SGD, but without wall-clock time measurements or communication cost in multi-node training, the real-world applicability remains unclear. Because these foundational concerns—substantive novelty, comprehensive cost analysis, and a sufficiently broad threat model—are not yet addressed, I continue to recommend rejection.

---

> > > ### Author Response · Authors · 2025-08-07
> > >
> > > Thank you for your feedback. We are sorry our response did not address your concerns. Respectfully, we fundamentally disagree with the claim that our framework closely “parallels existing rejection-sampling [...] techniques”. Would you be so kind as to provide us with concrete citations of such work?
> > >
> > > We acknowledge that the current presentation of our framework is confusing and have engaged with other reviewers to clarify these points. However, we think the novelty of our framework is a strength of our work (recognized by other reviewers). We are not aware of any existing work like ours in this setting.
> > >
> > > Regarding computational complexity, we are happy to provide further details. For example we can compare SGD, DP-SGD, and PD-SGD (Alg. 1) assuming constant time O(1) to compute a single gradient. (If we assume the time scales linearly with the number of parameters it does not change the comparison as it the same multiplicative factor in each case.)
> > > For one learning step:
> > > - SGD takes O(1) to compute the gradient and update the parameters.
> > > - DP-SGD takes O(b) where b is the batch size since it needs to compute per data point gradients
> > > - PD-SGD takes O(1) to compute the seed batch gradient and add noise to it, and then up to O(m) to compute the other batch gradients and do the privacy test.
> > >
> > > However, since PD-SGD updates can fail, the computational complexity for a fixed number $k$ of gradient updates is $O(m k (1 - \rho)^{-1})$ in the worst case, where $\rho$ is the rejection rate. This is because $(1 - \rho)^{-1}$ is the expected number of iterations to get one valid gradient update. By comparison $k$ gradient updates with SGD takes $O(k)$ and $O(k b)$ for DP-SGD.
> > >
> > > To obtain the most accurate models with DP-SGD typically requires using very large batches [1-2], meaning we expect $b \\gg m$, the computational time for PD-SGD is lower than DP-SGD. Further, we can control the rejection rate to avoid it being too high. As we show in experiments, PD-SGD achieves favorable privacy-utility tradeoffs compared to existing empirical defenses while keeping the reject rate (relatively) small (e.g., $\\ll 1$).
> > >
> > > Regarding the claim of lack of evaluation in a distributed setting. We state in Appendix G that we intend our technique for use in a centralized learning environment. (We don’t believe our paper suggests applicability to distributed learning, even in the main text.) Most existing techniques (including those we compare against) do not evaluate timing in a distributed setting when they are proposed. Given this, we do not believe this is a fair expectation of papers such as ours.
> > >
> > > Regarding results against stronger white-box adversaries, we would point out Appendix C.5 and Lemma 7, which shows that PD-SGD satisfies differential privacy. It is well-known that differential privacy resists white-box adversaries that seek to inference membership.
> > >
> > > If you are willing to engage with us further, please let us know if we can offer additional clarification.
> > >
> > > [1] De, Soham, et al. "Unlocking high-accuracy differentially private image classification through scale." arXiv preprint arXiv:2204.13650 (2022).
> > >
> > > [2] Sander, Tom, Pierre Stock, and Alexandre Sablayrolles. "Tan without a burn: Scaling laws of dp-sgd." International Conference on Machine Learning. PMLR, 2023.

---

### Official Review · Reviewer_jVqE · 2025-06-25

**Clarity:** 4
**Significance:** 2
**Originality:** 3
**Rating:** 3
**Confidence:** 4

**Summary:**

This paper introduces a novel empirically-effective privacy framework called PD - plausible deniability, and a privacy preserving training algorithm under this framework, PD-SGD. On an intuitive level, models trained under PD have the property that training data points within any mini-batch is not significantly "different" under gradient based white-box (i.e. has model weight access, I'm a bit confused by the author's definition of blackbox vs whitebox, but that of tangential interest here) membership inference attacks. Mathematically, this notion can be expressed by the existence of at least T alternative minibatches that falls within a certain Euclidean distance within the target batch, such that their log-density under a normal perturbation model has bounded change. Methodologically, this is implemented by a DP-SGD-like Graussian perturbation followed by rejection sampling of batches with outlier gradients. Empirically, models trained by PD-SGD obtain favorable privacy-utility trade-offs compared to other empirically and theoretically guaranteed defense mechanisms against MI attacks.

**Questions:**

1. Comparisons against some more recent defense mechanisms would make the experimental contributions more convincing.
2. Some details on the attack mechanisms are owed, even if in the appendix.
3. Some training cost comparisons would be good to have.
4. Table 1 and table 3's font is too smal.

**Ethical Concerns:**

["NO or VERY MINOR ethics concerns only"]

**Limitations:**

Yes

**Quality:**

3

**Strengths And Weaknesses:**

Strengths:
1. The paper explains various properties of the proposed PD framework in reasonable detail.
2. Compared to other empirically motivated defense mechanisms, PD-SGD shows competitive performance while being more up-right under theoretical examination. Compared to DP-based methods, PD-SGD provides clear performance benefits in privacy-utility trade-off.
3. Experimental evaluations of PD-SGD is fairly thorough on common architectures and datasets, with multiple MI attack methods as benchmark.

Weaknesses:
1. PD falls short of a "attack-agnostic" privacy framework due to it's definition being reliant on a source pool of mini-batches, which seems quite specific to a particular configuration of the MI setting. If the imaginary attacker do not have the same pool of mini-batches as the model provider, (or even better/worse, the attacker has only one target data batch and wants to know if it has been used in the training of the model), then there is no such notion of "d" in equation 4, hence no clear way quantify the privacy afforded by PD. This is in contrast to DP, which is agnostic to attacker knowledge.
2. The baselines in the experiments are quite dated. Various improved alternatives to SELENA and ADVReg have been proposed in the literature since 2023 that also out-perform these two baselines in privacy-utility trade-offs.
3. The proposed method PD-SGD has an unfavorable privacy-compute trade-off, in that narrowing the threshold $\gamma$ results in more gradient comparisons needed before a minibatch is rejected. In the extreme case where $\gamma$ is too low, the algorithm would traverse through the entire dataset for each gradient step. Storing gradients is also not feasible for modern models in the Billions parameter range. Some form of caching, embedding and efficient search strategies should be discussed to make the proposed method more practical.

---

> ### Author Rebuttal · Authors · 2025-07-30
>
> We thank the reviewer for providing insightful feedback and recognizing PD-SGD’s empirical performance. We agree with some of the concerns raised and have conducted additional experiments to directly address them. Other points appear to stem from miscommunication or lack of clarity in our presentation. We aim to clarify these points thoroughly below.
>
> **1. PD falls short of a "attack-agnostic" privacy framework**
>
> PD‑SGD’s “batch pool” is an internal training-time test, not an assumption about the attacker’s knowledge: the algorithm verifies for every accepted update that there exist at least $T$ alternative batches that would have produced an $\alpha$‑similar noisy gradient; this is an existential indistinguishability claim, so the adversary doesn’t need to know (or share) that pool to be confused. In fact, there is no restriction on the attacker’s knowledge about the batches or the dataset. In Lemma 4 (Appendix C.3) the batches are **arbitrary** and **can be adversarially chosen**. Said differently, to satisfy PD the training process must be able to plausibly deny *any batch*, including not only an adversarially poisoned batch for example but also non-existent batches (if the adversary is wrong and has no idea what kind of data the model trainer is even using).
>
> To clarify:  $d$ should not be used as a direct quantifier for the privacy of PD-SGD in the strict sense implied by the reviewer. (It is not similar to the privacy budget $\varepsilon$ for DP, it is more like a sensitivity measure.) Rather, $d$ is the $l_2$ distance between two batches’ gradients, which is a quantity useful to think about the probability of rejection as a function of the similarity between batches or the change introduced by adding/removing a single data point. Lemma 2 shows that as $d$ increases the chance that the other batch can be used to explain the update decreases exponentially (and thus such updates are overwhelmingly likely to be rejected). Satisfying PD (or not) is a property of a training process that is independent of $d$.
>
> The bounds of Lemmas 4 and 7 are more closely related to the privacy leakage in the PD sense. These bounds do not depend on $d$ but depend on the privacy parameters (i.e., $\gamma$, $T$, etc).
>
>
> **2. Baselines in the experiments are quite dated.**
>
> Yes, we agree with that. We have conducted new experiments using a new empirical defense mechanism HAMP [1] which is published in NDSS 2024 on the Purchase-100 dataset and show its results with highlight in the following table:
>
>
> **Table:** Evaluations for new baseline: HAMP
>
> | Method                   | Test Acc | P-Attack | R-Attack | S-Attack | C-Attack |
> |--------------------------|----------|----------|----------|----------|----------|
> | Non-private              | 68.56%   | 0.76     | 0.78     | 0.77     | 0.12%    |
> | AdvReg                   | 57.56%   | 0.70     | 0.70     | 0.66     | 0.08%    |
> | SELENA                   | 64.31%   | 0.63     | 0.73     | 0.66     | 0.07%    |
> | HAMP                     | 61.66%   | 0.64     | 0.74     | 0.64     | 0.07%    |
> | DP-SGD (eps=8)           | 47.61%   | 0.56     | 0.56     | 0.56     | 0.08%    |
> | PD-SGD (param setting 1) | 64.83%   | 0.63     | 0.72     | 0.64     | 0.06%    |
> | PD-SGD (param setting 2) | 61.16%   | 0.61     | 0.59     | 0.60     | 0.06%    |
>
>
>
> Two takeaways emerge:1. PD‑SGD (setting 1) attains higher utility than HAMP while delivering comparable privacy. 2. PD‑SGD (setting 2) provides stronger privacy than HAMP at similar utility.
>
>
>
>
> **3. The proposed method PD-SGD has an unfavorable privacy-compute trade-off in extreme cases.**
>
> The reviewer is correct that the computational cost of PD-SGD depends on its reject rate. However, this is only a concern in cases with extremely high rejection rates (e.g., 99%+) which result from either a lack of parameter tuning, or a case where the data distribution makes it nearly impossible to train a high utility model with bounded privacy leakage.
>
> If hyperparameters are properly tuned, high rejection rates is a telltale sign that it may not be possible to obtain a favorable utility privacy tradeoff.
>
> We describe a strategy for empirical hyperparameter tuning in Appendix D. Before full training, we perform a two-phase search: in Phase 1, we screen $\sigma$ values based on utility; in Phase 2, we conduct short training runs (e.g., 200 steps) across a grid of $\gamma$ and $T$ to discard configurations with degenerate behavior — such as 0% or 100% rejection rates. This early-stage filtering effectively prevents the pathological case the reviewer describes.
> Moreover, our empirical tuning strategy significantly improves efficiency: instead of exhaustively evaluating all combinations (180 runs taking **110.14** GPU hours), our phased search reduces this cost to just **4.15** GPU hours. Thus, the trade-off between privacy and computation is well-managed in practice.
>
>
> **4. Storing gradients is also not feasible for modern models in the Billions parameter range.**
>
> PD-SGD does *not* require storing gradients. To implement the test, it suffices to compute the noisy gradients of the seed batch and then compute (one at a time) the gradients for alternative batches to perform the counting in Alg 1. This means that we only ever need to keep in memory the gradients for at most two batches at any given time. We believe this is feasible in many practical applications, even for large models.
>
> As evidence of the scalability of PD-SGD, we added a large‑model experiment: fine‑tuning LLaMA‑2‑7B on SST‑2 with PD‑SGD. Our run reaches 94.76% test accuracy while maintaining stable memory use (peak allocated memory 6,753.48 MB). It is comparable to 94.8% reported in Table 12 of Zhao et al. [2] with normal training and 92.2% with DP-Lora[3].  This shows PD‑SGD is practical for LLM fine‑tuning and does not incur prohibitive memory overhead. We will add these results to the paper.
>
>
>
> **5. Some details on the attack mechanisms are owed, even if in the appendix.**
>
> We thank the reviewer for the suggestion. Details of our attack setup are provided in Appendix E.3, where we describe how we implement membership inference attacks using the Privacy Meter framework. Each attack closely follows the design choices in its original paper, as cited. If there are specific aspects of the attack mechanisms the reviewer would like to see elaborated, we would be happy to clarify or include them in the revised version. We appreciate any guidance on which details would be most helpful.
>
>
> **6. Some training cost comparisons would be good to have.**
>
> We provide the training time per step comparisons in Appendix F.4. and Table 8. We also add additional experiments to measure the GPU memory usage against DP‑SGD and non‑private training as follows.
>
>
> **Table:** GPU Memory Usage Comparison
>
> | Method | alloc    | reserved   | peak_alloc | peak_reserved |
> |--------|----------|------------|------------|---------------|
> | SGD    | 37.9 MB  | 6562.0 MB  | 4918.2 MB  | 6562.0 MB     |
> | DP-SGD | 10345.9 MB | 18612.0 MB | 10345.9 MB | 18612.0 MB    |
> | PD-SGD | 145.8 MB | 6712.0 MB  | 5014.5 MB  | 6712.0 MB     |
>
>
> PD‑SGD’s memory footprint is comparable to SGD and far below DP‑SGD. Thus, on memory‑constrained GPUs where DP‑SGD may exceed capacity, PD‑SGD remains feasible.
>
>
>
> **7. Table 1 and Table 3's font is too small.**
>
> We apologize for that and will fix it in the revised version.
>
>
> Reference:
>
> [1] Chen, Zitao, and Karthik Pattabiraman. “Overconfidence Is a Dangerous Thing: Mitigating Membership Inference Attacks by Enforcing Less Confident Prediction.” Network and Distributed System Security (NDSS) Symposium, 2024.
>
> [2] Zhao, Justin, et al. "Lora land: 310 fine-tuned llms that rival gpt-4, A technical report." arXiv preprint arXiv:2405.00732 (2024).
>
> [3] Yu, Da, et al. "Differentially private fine-tuning of language models."  ICLR 2022.

---

### Official Review · Reviewer_adTA · 2025-07-03

**Clarity:** 2
**Significance:** 3
**Originality:** 4
**Rating:** 5
**Confidence:** 3

**Summary:**

The paper proposes a novel framework that focuses on providing 'plausible deniability' of using the gradient update from a batch by only using that update if there are many other similar batches present in the training dataset. It provides both a theoretical analysis of this framework as well as an empirical investigation. Their technique, PD-SGD, achieves better utility-privacy tradeoffs than existing methods, while the paper also claims to provide theoretical justifications for their technique (arguing some level of protection against even future yet-to-be-discovered attacks).

**Questions:**

The 'theoretical justification' claim of the paper needs to be clarified.

Consider the following threat model: An adversary, with access to model outputs, is able to determine whether an individual was part of the training dataset or not, i.e., membership inference. Differential privacy provides guarantees that it is not possible to confidently distinguish between two outputs if the underlying dataset differed by exactly one point, i.e., it protects against the membership inference threat model. For a good 'theoretical justification' for privacy, one needs to choose a reasonable threat model and a technique that directly protects against the threat model.

Let's talk about plausible deniability. What exactly is the threat model here?

Is the threat model still membership inference? If so, the guarantees of rejecting highly anomalous batches are not enough, and any theoretical justification needs to be directly connected to differential privacy.

Or is the threat model an adversary not able to tell which exact batch out of T different batches was used for the gradient update at a particular training step? The theoretical guarantees are more in line here, but the threat model itself isn't strong. Since batches are created randomly at every training step, the idea of a 'batch' does not hold meaning across the complete training run. So it is unclear how the threat model of plausible deniability at a single training step translates to the overall training.

I found the empirical aspect of the work more promising. However, since that is not the focus of the paper, there aren't enough experiments or details here to support the claims. For instance, the authors argue that avoiding calculating individual-level gradients, which is a requirement for DP-SGD but not for their technique, PD-SGD, can be computationally beneficial. However, there is no empirical evidence to show how beneficial this difference is, and how much does constant rejection of batches after gradient calculation might affect this.

**Ethical Concerns:**

["NO or VERY MINOR ethics concerns only"]

**Final Justification:**

The rebuttal with the authors was quite extensive. Many of my confusions about the plausibility deniability framework itself were clarified, and I very strongly urge the authors to incorporate changes in the final version of the paper based on those discussions.

At the same time, some of my concerns about the real-world applicability of the framework also became clearer. There is still work to be done to understand whether this framework can indeed have a real-world impact or not. But I believe it can be argued that not everything should be expected from just one paper.

Overall, my decision is still close to borderline accept, i.e., I believe there is value in this work, but I do not feel confident enough to champion for it if other reviewers strongly disagree. However, given all the clarifications provided by the authors and the recommendation by the reviewing guidelines to use 'borderline' decisions sparingly, I feel more comfortable recommending 'accept' now than I did at the start of the rebuttal period. So I will be increasing my score to accept.

**Limitations:**

Please see the comments above.

**Quality:**

2

**Strengths And Weaknesses:**

Quality: The plausible deniability framework does not have the appropriate theoretical justifications that the paper claims, while the empirical results also do not provide an in-depth exploration to support several claims made in the main paper (see 'Questions' section for details). I am optimistic that at least for the theoretical claims, the paper can be improved during the rebuttal phase, to be more precise about these claims.

Clarity: The submission is clearly written and well organized for the most part. My comments here are at the intersection of Quality and Clarity, repeating that the theoretical justifications and the threat model are not clear.

Significance: The empirical results have the potential to be highly impactful, as the paper provides a more computationally efficient technique to achieve privacy that is also shown to be able to improve the privacy-utility tradeoff against some baseline techniques. While the theoretical threat model might be different, it is clear that their technique is more efficient and provides better tradeoffs. A deeper analysis with more experiments and other techniques would definitely improve this aspect of the work.

The theoretical discussion and the plausible deniability framework, too, are quite valuable, and I believe stand well on their own. As discussed in detail in the 'Questions' below, I feel the value of this framework becomes murky because of the authors' attempts to place it within the existing MIA framework of privacy, but without enough appropriate justifications for the same. As I mentioned above, I am optimistic that the rebuttal phase will either help the authors clarify these connections or acknowledge the distinction between the two and appropriately argue why their framework, too, has value, without forcing it within existing discussions of MIA.

Originality: The perspective of plausible deniability, to my knowledge, is quite novel and interesting. The paper takes a bold step away from existing privacy frameworks and proposes something new that could have useful implications for privacy protection.

---

> ### Author Rebuttal · Authors · 2025-07-31
>
> We thank the reviewer for acknowledging the novelty and empirical contributions of our work, as well as for the comprehensive review and insightful feedback. We apologize for any confusion regarding our theoretical analysis. We provide clarification and address the questions below.
>
> **1. Threat model?**
>
> It is important to distinguish between the threat model employed for empirical evaluation and the threat model implicit in a privacy notion. These can be different, as is the case for papers evaluating black-box MIA on models trained with DP, for example. What is important is that the former matches attacks used to evaluate a defense empirically.
>
> For the purpose of empirical evaluation, as mentioned in section 5.1, we consider a black‑box membership adversary (first threat model as you mentioned). The adversary knows the full PD-SGD algorithm and parameters, but only interacts with the final model; the training dynamics (noisy gradients, pass/fail decisions) remain hidden. This allows us to fairly compare against existing empirical defenses.
>
> The implicit threat model in the definition of plausibility deniability is captured in the discussion in Appendix C.3 and Lemma 4. It corresponds to an adversary different and in many cases much stronger than the black-box membership inference adversary.  The crucial point that “batches being created at random” as the reviewer puts it, is the way algorithms SGD (and PD-SGD) work. It is not an assumption about the power of the adversary. In Lemma 4 the batches are **arbitrary** and **possibly adversarially chosen**. Stated differently, with plausible deniability, a model trainer has to be able to defend themselves against a claim from an unusually strong adversary that **crafts the batches**. For example, the adversary may choose to cram all the data points with gradients pointing in a conspicuous direction in the *same batch* so as to maximize influence on the gradient update. The model trainer still must be able to plausibly deny use of that batch.
>
> Since existing defenses do not satisfy plausible deniability, it does not make sense to evaluate them against the PD adversary. However, our work shows that PD-SGD empirically provides protection against black-box membership inference.
>
>
> **2. Why can PD-SGD protect models from black-box membership inference?**
>
> Intuitively, denying a batch was used is more difficult than denying a specific data point was included in *some* batch. Informally, PD‑SGD protects membership privacy by refusing to move the parameters on the basis of **anomalous, high‑influence** batches and only accepting updates that many other batches could have plausibly produced, thereby blurring attribution and shrinking member/non‑member loss gaps.
>
> For a data point used during training (a member) to eventually result in successful membership inference attack once the model is trained, it must be such that it distorts the gradient of at least one batch that includes it during training. Since PD-SGD discards parameter updates unless at least T other batches make the noisy observation almost as likely (within factor $\alpha$), the leakage of any given data point at any given iteration is bounded. As Lemma 2 shows, the chance that another batch explains the observation depends on the $l_2$-distance between gradients and the greater than gradient distortion $d$ (due to adding the data point to the batch) the higher the probability of rejecting the update.
>
>
> **3. Computational benefits of PD-SGD:**
>
> Thanks for pointing it out. We evaluated the computation time per step for PD-SGD, SGD and DP-SGD in Appendix F.4 and Table 8. PD-SGD is noticeably slower than standard SGD but notably faster than DP-SGD for a single training step.
>
> We agree that the rejection rate plays a significant role in training time and therefore it is worth evaluating as the reviewer suggests. Avoiding a very high rejection rate is desirable as it saves computational resources. However, the goal is not to minimize the rejection rate as some rejection as necessary for privacy. The rejection rate depends on privacy parameters $\sigma$, $\gamma$, and $T$ and therefore hyperparameter tuning can be used to strike a balance between computational complexity and memory consumption and utility and privacy. We will add a discussion of this in the revised paper.
>
> For now, we conducted experiments to measure GPU memory usage of PD-SGD against DP‑SGD and non‑private (SGD) training with Wide-ResNet-16-4 model on CIFAR10 dataset.
>
> **Table:** GPU Memory Usage Comparison
>
> | Method | alloc    | reserved   | peak_alloc | peak_reserved |
> |--------|----------|------------|------------|---------------|
> | SGD    | 37.9 MB  | 6562.0 MB  | 4918.2 MB  | 6562.0 MB     |
> | DP-SGD | 10345.9 MB | 18612.0 MB | 10345.9 MB | 18612.0 MB    |
> | PD-SGD | 145.8 MB | 6712.0 MB  | 5014.5 MB  | 6712.0 MB     |
>
>
>
> Results show that PD‑SGD’s memory footprint is slightly higher but **comparable to SGD** and **far below DP‑SGD**. Thus, on memory‑constrained GPUs where DP‑SGD may exceed capacity, **PD‑SGD remains feasible**. We believe the high memory usage for DP-SGD is due to per data point gradient computations, which are not necessary for PD-SGD.

---

> > ### Comment · Reviewer_adTA · 2025-08-03
> >
> > I appreciate the clarification.
> >
> > PD-SGD being able to protect against membership inference, even though that’s not the main goal of the formulation, is quite interesting, as I noted earlier, and I’m glad to see more experiments focused on the empirical aspect of this algorithm. I believe it will strengthen the paper.
> >
> > As for the threat model, I would like to ask for some more clarification. Consider the following adversarial game:
> > 1. A dataset D is divided into disjoint mini-batches B1, B2, …, Bm.
> > 2. Player 1 performs exactly one gradient update on their model using one batch out of B1, B2, …, Bm. Let’s call this batch Bt.
> > 3. A coin is flipped: with 50% probability, the batch Bt is chosen, and with 50% probability, a random batch from B1, B2, …, Bm is chosen, which is NOT Bt. In other words, with 50% probability, the batch that was actually used for training is chosen, while with 50% probability, some other batch not used for training is chosen. Let’s call the chosen batch Bq (which can be Bt with 50% probability).
> > 4. Player 2 is given access to the model before the gradient update, the model after the gradient update, and all the batches B1, B2, ..., Bm. Player 2 is then asked whether Bq was used for this gradient update or not.
> >
> > Plausible deniability aims to limit the capability of Player 2 being able to answer this correctly, i.e., to do well in this game. Please clarify if I’ve understood it correctly. Make sure to point out even small errors or nuances that I might have missed in understanding the threat model, because I am going to heavily rely on it for my following questions.
> >
> > Assuming that I’ve understood it correctly, I would like to hear some clarification on the following:
> > 1. I think the game is skewed towards only the deniability of certain batches that could have been used, but not for batches that were not used. That is, there are many batches like Bt that could have been used, hence plausible deniability, and so if Bq is any one of those batches, Player 2 will not be able to detect this under PD-SGD (but will be able to detect it under SGD, hence the benefit of PD-SGD). However, if the batch Bq is outside these many similar batches, irrespective of whether SGD or PD-SGD is used, the player can confidently say that the batch Bq was not used for training, hence my comment on the game being skewed. Am I correct in saying this?
> > 2. How does this game connect with the motivating example provided in Appendix C? The framework seems to me more about plausibly denying the use of certain batches during training, but how does that connect to confidently denying the accusation of training on certain data? Maybe a related question, it's still not clear to me what the value of this privacy framework or this adversarial game is. I guess once the motivating example connections are clearer to me, things might start to make more sense.
> > 3. How does this all scale to multiple gradient updates and eventually a complete training run? A complementary question: how does this game scale when batches are randomly created at each gradient update?

---

> > > ### Author Response · Authors · 2025-08-06
> > >
> > > Thank you for your thoughtful and detailed follow-up and engaging with our response.
> > >
> > > It is possible to derive a game formalizing PD. The adversarial game you describe is interesting but does not exactly map onto our proposed notion. A crucial nuance is in step 3 where in your formulation Bq is the specific batch used as seed. With this, the coin flip (call the outcome $b$) no longer maps onto the “worlds” set up in Definition 2 and Lemma 4 but onto whether the seed batch is Bt or one of B1,...,Bm.
> > >
> > > By contrast, in our framing the seed batch is selected uniformly at random. There are two worlds/hypotheses:
> > >
> > > - World0 / H0 (i.e., $b=0$): Batches B1,...,Bm
> > > - World1 / H1 (i.e., $b=1$): Batches B1,...,Bm, and Bt.
> > >
> > > Observe that even if $b=1$ then Bt is selected as seed only with prob 1/(m+1). With prob 1-1/(m+1) another batch is selected as seed.
> > >
> > > Regarding your questions:
> > >
> > > **1.** If by "skew" you mean that because if Bq is an "outlier" or "anomalous" batch, it is unlikely (if selected as seed) to lead to an update that can be plausibly denied, then yes.
> > >
> > > It may be possible to have a definition such that we can pretend that any batch resulted in any gradient update, but this is not what PD aims to satisfy. (Note: it would likely require adding so much noise that utility will suffer.) Instead, PD ensures that for any batch (including an adversarially chosen one) that could have been the seed, there are $\geq T-1$ alternatives that can plausibly explain the gradient update.
> > >
> > > If a batch Bq is so much of an outlier that it is unlikely to ever result in a gradient update, then that is okay. Knowing this (in some cases knowable before the model is even trained) does *not* help the adversary distinguish between World0 and World1 more than a small bounded advantage due to one world having one more possible seed batch, i.e., roughly $1+1/m$ (ignoring $alpha$, etc. see by Lemma 4 for details).
> > >
> > > **2.** Our PD formulation is a batch notion. It is connected to the motivation described at the beginning of Appendix C because any accusation of having used specific data for training is at bottom a claim that this data was part of one or more batches at some training iteration/step.
> > >
> > > Focus on a specific iteration where the accusation is that the model trainer has used specific illicit/copyrighted/sensitive data X (say composed of $k$ data points). For example, PD can be used to model the case where a single batch Bt contains all or a significant fraction of X *or* if there was no such batch but the adversary thinks there was. (Even if the accusation is false, it is desirable for the model trainer to be able to defend themselves against it.)
> > >
> > > In practice, this situation can occur various ways such as if the model trainer uses online/incremental learning, curriculum learning, or purposefully uses X only some of the time in one batch as a way to evade detection.
> > >
> > > As pointed out in our initial response, modeling privacy this way also allows us to capture adversarial batch selection (a very strong adversary that gets to decide what data ends up in what batch) through poisoning or other means. Related work shows that adversarial ordering of data into batches can have substantial impact on learning [1]. This can be modeled in our framework but not by other existing privacy notions.
> > >
> > >
> > > **3.** The PD privacy guarantee is defined per step, and it degrades the more steps we consider. Composition can be used naively (e.g., applying Lemma 4 repeatedly), or we can opt to work in the DP framework and use the relationship with DP (Appendix C.5 & Lemma 7) and then use existing composition results. Since our focus is empirical, we opted to leave for future work derivation of stronger composition results (e.g., a version of Lemma 4 over multiple steps where we account for the “overlap” of batches across steps).
> > >
> > > When batches are created randomly (by which we mean shuffle-and-partition at every step), the adversary actually has even more uncertainty and this uncertainty compounds over time (although the adversary still accumulates more information over time).
> > >
> > > Intuitively, having all of X crammed in the same batch (and that batch being somehow reused throughout training) is good for the adversary. Recall that for PD-SGD we need $\geq T$ $\alpha$-similar batches to pass the test. Therefore if $k < T$ then no matter how X is distributed the privacy test won’t pass solely due to inclusion of X (without the help of other "benign" batches).
> > >
> > >
> > > Please let us know if this provides sufficient clarity or if we can offer additional details. We thank you again for raising these questions and will incorporate this discussion into our revised manuscript.
> > >
> > > [1] Shumailov, Ilia, et al. "Manipulating sgd with data ordering attacks." NeurIPS 34 (2021): 18021-18032.

---

> > > > ### Comment · Reviewer_adTA · 2025-08-06
> > > >
> > > > Appreciate the clarifications, although the arguments aren’t very clear to me, still, or the questions were not interpreted properly.
> > > >
> > > > Focus on the following questions:
> > > > 1. Please provide the complete privacy game for plausible deniability. Not a correction on what I missed, but a complete game. Make sure it is self-contained (no references to Definition 2, Lemma 4, etc.). And make sure all terms you use are properly defined (was ‘b’ the coin flip, I guess? Be clear, just make sure it is self-contained). The survey by Salem et al. as suggested by the Reviewer is7x is a good place to start if you want to see how to create such a game. Better have more details than less, since you’re proposing a novel game, which is different from existing games.
> > > > 2. Place the ‘model developer’ and the ’person accusing the developer of using their data’ in this game. I want to see exactly how this game has connections with your motivating example in Appendix C.
> > > >
> > > > Some other concerns/comments (No need to address them in your response, this is just my thoughts on your current presentation):
> > > > 1. A lot of what I think is central to motivating a new framework for privacy is the motivation behind why it is needed, and I’m realizing more and more that you’ve put these things in the Appendix (Appendix C, more specifically). Please make sure to move some version of them to the main paper as well in the next version of the paper.
> > > > 2. I don’t think you should claim your ‘focus is empirical’, as you did in the rebuttal. I like what you’re trying to do with a new privacy framework, although, of course, there is some confusion there, and more clarification will help. But if you claim your focus is empirical, I don’t believe you have diverse enough results or even strong enough results to argue for it.

---

> > > > > ### Author Response · Authors · 2025-08-07
> > > > >
> > > > > Thank you for your continued engagement and for providing such clear and actionable feedback. We apologize if we misunderstood and if our previous explanations were not clear.
> > > > >
> > > > > We also take to heart your comments about our paper's presentation. From the discussion it is clear we need to substantially revise the presentation. We agree that the motivation in Appendix C is central and will move a revised version into the main paper. Furthermore, you are right about our framing; the novel privacy framework is a core contribution, and we will revise the text to reflect this, rather than over-emphasizing an "empirical focus." By empirical focus we are not seeking to minimize the value in the framework, but rather simply that if PD-SGD did not empirically perform in terms of utility privacy tradeoff (e.g.,  against MIA), then having a framework without an algorithm demonstrating its value may not be worthwhile.
> > > > >
> > > > >
> > > > > As requested we provide below, a complete self-contained Batch Plausible Deniability game. Whenever possible we used notation consistent with the SoK by Salem et al.
> > > > >
> > > > > Notes:
> > > > > - A’ denotes the algorithm the adversary uses to choose batches and the target batch.
> > > > > - A denotes the algorithm that outputs the adversary’s guess b’.
> > > > > - $\mathcal{T}$ denotes *one step* of the training algorithm. This algorithm takes in a set of batches as inputs and outputs a gradient vector g.
> > > > > - We assume there are at least $m > 2$ batches that are disjoint and of roughly equal length (with the exception of the last batch).
> > > > >
> > > > > Batch PD Game
> > > > >
> > > > > 1: (B1, B2, …, Bm), Bt <- A’($\mathcal{T}$)	// adversary pick batches and the target batch
> > > > >
> > > > > 2: b ~ {0,1}					// sample random bit b
> > > > >
> > > > > 3: if b = 1 then:
> > > > > 	g <- $\mathcal{T}$(B1,B2,...,Bm,Bt)	// gradient from $\mathcal{T}$ with Bt included
> > > > >
> > > > > 4: else:
> > > > > 	g <- $\mathcal{T}$(B1,B2,...,Bm)	// gradient from $\mathcal{T}$ *without* Bt
> > > > >
> > > > > 5: b’ <- A(g, $\mathcal{T}$, B1, B2, …, Bm, Bt)
> > > > >
> > > > > Define the advantage of the adversary as: Adv = 2 Pr{b’=b} - 1
> > > > >
> > > > >
> > > > > The Adversary (A, A’) in the game represents the person or entity accusing the developer. This could be an artist, a writer, or a company claiming their copyrighted data was used for training. It could also be an individual who provided data and later withdrew consent for the data’s use.
> > > > >
> > > > > The Batch-PD game models a scenario where the model developer used either (B1,B2,...,Bm,Bt) or (B1,B2,...,Bm) to obtain a gradient g during the training of the model and g was observed by the adversary. Note that the only randomness is in the choice of bit b and in the algorithm $\mathcal{T}$ which outputs g.

---

> > > > > > ### Author Response · Authors · 2025-08-07
> > > > > >
> > > > > > We can provide further games to show the connection with membership inference. For all games below, the objective is also to guess bit b. The advantage of the adversary is still: Adv = 2 Pr{b’=b} - 1. The advantage of the best adversary for a game G is denoted Adv(G). Furthermore, $\mathcal{D}$ denotes the data distribution (used only in some games).
> > > > > >
> > > > > > Let Game G0 denote the One-Step Batch PD game described above (previous post). We consider the following additional games.
> > > > > >
> > > > > >
> > > > > > Game G1 — Singleton-Batch PD
> > > > > >
> > > > > > 1: (B1, B2, …, Bm), z <- A’($\mathcal{T}$)	// adversary pick batches and target data point z
> > > > > >
> > > > > > 2a: Bt = {z}					// batch with only z in it
> > > > > >
> > > > > > 2b: b ~ {0,1}					// sample random bit b
> > > > > >
> > > > > > 3: if b = 1 then:
> > > > > > 	g <- $\mathcal{T}$(B1,B2,...,Bm,Bt)	// gradient from $\mathcal{T}$ with Bt included
> > > > > >
> > > > > > 4: else:
> > > > > > 	g <- $\mathcal{T}$(B1,B2,...,Bm)	// gradient from $\mathcal{T}$ *without* Bt
> > > > > >
> > > > > > 5: b’ <- A(g, $\mathcal{T}$, B1, B2, …, Bm, Bt)
> > > > > >
> > > > > > **Difference between G0 and G1**: In G1 the adversary does not get to pick a full target batch, but only a single data point.
> > > > > >
> > > > > > We have Adv(G0) $\geq$ Adv(G1), since any adversary that wins G1 with some advantage can play G0 and output the same z as Bt as it would when playing G1.
> > > > > >
> > > > > >
> > > > > >
> > > > > >
> > > > > > Game G2 — Chosen Data, Random Batches (similar to Strong Membership Inference)
> > > > > >
> > > > > > 1: S, z <- A’($\mathcal{T}$)	// adversary pick dataset S (|S| = n) and target data point z
> > > > > >
> > > > > > 2a: (B1, B2, …, Bm) <- Partition(S)	// randomly suffle and partition
> > > > > >
> > > > > > 2b: Bt = {z}				// batch with only z in it
> > > > > >
> > > > > > 2c: b ~ {0,1}				// sample random bit b
> > > > > >
> > > > > > 3: if b = 1 then:
> > > > > > 	g <- $\mathcal{T}$(B1,B2,...,Bm,Bt)
> > > > > >
> > > > > > 4: else:
> > > > > > 	g <- $\mathcal{T}$(B1,B2,...,Bm)
> > > > > >
> > > > > > 5: b’ <- A(g, $\mathcal{T}$, S, Bt)
> > > > > >
> > > > > >
> > > > > > **Difference between G1 and G2**: In G2 the adversary picks the dataset S but does not determine the partitioning into batches (or learn it).
> > > > > >
> > > > > > We have: Adv(G1) $\geq$ Adv(G2). If adversarial partitioning provides any benefit then adversaries for G1 can use that but those for G2 cannot.
> > > > > >
> > > > > >
> > > > > > Game G3 — Average Membership Inference
> > > > > >
> > > > > > 1a: S ~ $\mathcal{D}^{n}$	// sample n data points i.i.d. from the data distribution
> > > > > >
> > > > > > 1b: z <- A’($\mathcal{T}$)	// adversary picks target data point z
> > > > > >
> > > > > > 2a: (B1, B2, …, Bm) <- Partition(S)	// randomly suffle and partition
> > > > > >
> > > > > > 2b: Bt = {z}				// batch with only z in it
> > > > > >
> > > > > > 2c: b ~ {0,1}
> > > > > >
> > > > > > 3: if b = 1 then:
> > > > > > 	g <- $\mathcal{T}$(B1,B2,...,Bm,Bt)
> > > > > >
> > > > > > 4: else:
> > > > > > 	g <- $\mathcal{T}$(B1,B2,...,Bm)
> > > > > >
> > > > > > 5: b’ <- A(g, $\mathcal{T}$, Bt)
> > > > > >
> > > > > >
> > > > > > **Difference between G2 and G3**: in G3 the adversary picks only the target data z, not the dataset S (which is sampled randomly).
> > > > > >
> > > > > > We have: Adv(G2) $\geq$ Adv(G3). We conclude: Adv(G0) $\geq$ Adv(G3). A one-step algorithm $\mathcal{T}$ that achieves batch PD (i.e., no adversary has an advantage greater than some $\lambda>0$) also bounds (one-step average) membership inference.
> > > > > >
> > > > > >
> > > > > >
> > > > > > Differences with games in Salem et al. SoK, which: (1) are not one step. (2) do not model batches and partitioning. (3) operate in the “replace-one” setting (i.e., adversary chooses z0, z1 but zb is included; e.g., Game10 in the SoK) not in the “add-remove” setting.
> > > > > >
> > > > > > Differences (1) and (2) are a consequence of PD being a batch notion. We need the games to explicitly model the batches. We do not believe (3) is a consequential difference. It is well-known that the “replace-one” and “add-remove” settings can be related (usually a constant factor is the difference between the two). To be consistent with the results we give in the paper we present games in the “add-remove” setting.
> > > > > >
> > > > > >
> > > > > > We hope this helps clarify the connection between batch PD and membership inference. Let us know if you have any further questions. And thank you again for your valuable feedback. We plan to incorporate this in the revised version of our paper.

---

> > > > > > > ### Comment · Reviewer_adTA · 2025-08-08
> > > > > > >
> > > > > > > Okay, the plausible deniability game makes so much more sense to me now! Thank you for this formulation, and I strongly recommend that the authors include the game G0 in the main paper in the next version.
> > > > > > >
> > > > > > > With a better understanding of what exactly this game is, I want to go back to the paper and read it again before I make my final recommendation. However, as this will take some time, and the author-reviewer discussion period is coming to an end, I will just leave the authors with the knowledge that, as of now, I have a positive view of the paper.
> > > > > > >
> > > > > > > One final comment: I didn’t make this comment earlier, since I didn’t quite understand the connection between plausible deniability and the motivating example, but now that I do, I think I can raise this point. A highly personal opinion, but I find the perspective of the motivating example quite weird. The motivation to me seems to be framed as an artist, a writer, etc., trying to find whether their data was used or not for training, being plausibly denied by the model developer. The paper said that the motivation is that the model developer can provide definitive proof that certain data was not used for training, but clearly, based on the game, all this does is that the adversary cannot tell whether certain data was used or not. The developer might have still used that data; it’s just that the adversary cannot tell the difference.
> > > > > > >
> > > > > > > The framework itself isn’t bad; that’s how all privacy frameworks are designed. It’s the framing that I personally do not like. Imagine if membership inference games were motivated as: If you can train your models not to be susceptible to membership inference, you can train on any data without consent, and they will never know! This is what I’m getting from the plausible deniability motivating example in Appendix C. I think framing it as protecting sensitive data or framing it as a successful adversary being able to prove data usage might have been better. To me, the entity with all the power is already the model developers and big AI companies, and a framing to protect them feels weird.
> > > > > > >
> > > > > > > All that being said, this will not impact my evaluation of the paper.

---

> > > > > > > > ### Author Response · Authors · 2025-08-08
> > > > > > > >
> > > > > > > > We appreciate your continued engagement and are happy we were able to provide clarifications. We will do as suggested and include game G0 and the surrounding discussion in the revised version. Thank you for helping us improve our paper!
> > > > > > > > Regarding the point about motivation, we appreciate this very thoughtful perspective. We’ll give it some serious thought and perhaps make changes in the revised version.

---

### Official Review · Reviewer_is7x · 2025-07-04

**Clarity:** 3
**Significance:** 2
**Originality:** 3
**Rating:** 5
**Confidence:** 3

**Summary:**

The paper proposes a variant of DP-SGD/Noisy SGD without clipping and with additional censoring of minibatch gradients that do not pass the so-called "privacy test", which ensures that each applied mini-batch gradient is $\alpha$-indistinguishable from at least $T$ other mini-batch gradients. The paper argues that this definition constitutes a theoretically justifiable notion of privacy, motivated by an observation that the model could plausibly deny an accusation by a party claiming that a specific mini-batch has been used in training (Lemma 4). The paper shows that on CIFAR-10, the proposed variant achieves higher empirical privacy-accuracy trade-off than empirical defenses and DP-SGD.

**Questions:**

- What are some realistic attack settings that are directly prevented by Lemma 4? What actual adversarial scenarios can be modeled by "denying an inclusion of a batch"?
  - How does Lemma 4 map to privacy of individuals?

**Ethical Concerns:**

["NO or VERY MINOR ethics concerns only"]

**Final Justification:**

In the rebuttal, the authors have provided a simple and convincing proof via a sequence of games, showing that the proposed notion of batch plausible deniability implies average MIA protection for a single step of the algorithm. This provides a clear operational meaning to the framework in terms of a standard privacy threat, which was my main issue with the submitted version of the paper.

In my opinion, it is crucial that the camera-ready revision prominently showcases this connection.

**Limitations:**

The limitations are discussed in the appendix.

**Paper Formatting Concerns:**

–

**Quality:**

2

**Strengths And Weaknesses:**

Obtaining meaningful relaxations of the strong threat model of differential privacy in a way that is practical and useful is a great open problem. This paper proposes plausible deniabilty of mini-batches as a potential solution, paired with a practical SGD-based algorithm that empirically achieves better privacy-utility trade-offs than DP-SGD and empirical defenses. The paper provides an extensive evaluation of different aspects of the algorithm: performance in different training regimes, with different batch sizes, distribution of example rejection rates, etc. It looks like PD-SGD can outperform the baseline methods in terms of utility and privacy in at least three tasks consistently—assuming carefully tuned parameters. This is a useful result, which could establish PD-SGD a state-of-the-art empirical defense.

Despite the extensive empirical investigation, a core part of the paper is lacking:

- *It's unclear how mini-batch deniability is a meaningful notion of privacy, regardless of possible theoretical guarantees on it.* The proposed theoretical guarantee is not substantiated as a meaningful and valid notion of privacy. In fact, the detailed interpretation of this notion is buried all the way in Appendix C. And that is: we have an (asymmetric) local DP guarantee on any released gradient value as a function of the sequence of mini-batches. Specifically $\tilde g \leftarrow M(x)$ satisfies $\varepsilon$-LDP over the space of two possible inputs $x_0 = \\{B, B_1, \ldots, B_m\\}$ and $x_1 = \\{B_1, \ldots, B_m\\}$ for some $\varepsilon = \log \lambda(\alpha, \sigma, m, T)$, where $\alpha$ is the likelihood ratio threshold in the privacy test, $\sigma$ is the Gaussian noise scale, $m$ is the number of batches, and $T$ the minimal number of indistinguishable batches. The problem here is that it is unclear why such guarantee matters in the first place:
  - What are some realistic attack settings that are directly prevented by such guarantee? What actual adversarial scenarios can be modeled by "denying an inclusion of a batch"?
  - How does it exactly map to privacy of individuals? The privacy of "batches" is certainly not a useful primary notion of privacy.

  A more substantial motivation for the relevance of this setup to privacy of individuals is needed in the paper, and it should be a part of the main body. The claims of _theoretical guarantees of privacy_ are unsubstantiated, as it is not clear why the guarantee described above is directly relevant to privacy, which is, again, not a per-batch property, but a per-individual property. In particular, the conclusion "Theoretical results [...] show that PD-SGD provides a superior privacy-utility trade-off compared to [...] DP-SGD" is wrong as of now as the paper does not explain how the proposed theoretical notion is a meaningful notion of privacy.
- *Unclear parameterization strategies.* Finding the parameters for PD-SGD seems challenging and require substantial grid search, as there are three primary privacy parameters, in addition to batch size. This seems like a fundamental limitation–which is addressed–but the paper should provide derive concrete Algorithms based on Section D, and ideally evaluate them, as otherwise it is quite unclear how to set the parameters.

Overall, the proposed notion is tailored to SGD's structure. As a result, its interpretation is *retrofitted to an existing algorithmic structure rather than to a meaningful notion of data privacy.* However, this should not take away at all from the fact that PD-SGD with carefully tuned parameters seems like a superior empirical privacy defense strategy in SGD!

To mitigate this issue, the paper should either drop the claims of theoretical privacy guarantees (e.g., in conclusions), treat the method as empirical even if motivated by batch plausible deniability, and treat the parameter selection strategies as part of the core contributions – or explain how the notion of batch plausible deniability *on its own* constitutes a notion of privacy. Note that the latter option is orthogonal to the observation in the Appendix that with some test instantations a single step of PD-SGD satisfies DP. A possible direction would be to derive the guarantee equivalent to Lemma 4 on the deniability of individual participation based on the batch size/number of batches the individual appears in.

This reviewer would like to make it clear that the direction is worthwhile, and alternatives to standard DP are critically needed to ensure privacy guarantees in many practical settings. The proposed approach in its current form misses the mark as it is hard to see the relationship of mini-batch deniability to relevant notions of privacy of individuals in the training data – the practical setting in which such deniability would be relevant is unclear, and even if it was, a single individual's data could appear in multiple mini-batches.

---

> ### Author Rebuttal · Authors · 2025-07-31
>
> We thank the reviewer for their valuable feedback. It is clear that our paper was confusing in its explanations of plausible deniability and what it provides. We apologize for the confusion due to the presentation. We will correct this in the revised version, including removing the sentence in conclusions which is misleading.
>
> **1. What is a meaningful notion of privacy?**
>
> We believe that plausible deniability, as defined in our paper, is a meaningful privacy notion that captures important practical considerations. By this we mean that the notion is well-defined, has desirable properties, and such that if satisfied then specific privacy-related violations cannot occur. That said, we acknowledge that plausible deniability frames its desideratum differently than DP and its variants and that perhaps makes it less intuitive. PD does not attempt to reason about neighboring datasets directly. Instead it reasons about the evidence we have that certain batches were or were not used.
>
> We would respectfully ask that the reviewer reconsider our claims and proposed notion in light of our clarifications below. When considering new privacy notions it is beneficial to remain open to frameworks beyond established approaches (e.g., DP and its variants).
> While our framework is different from DP by design, we believe its value is best seen by evaluating it on the unique scenarios it addresses.
>
>
> **2. Why is batch plausible deniability meaningful, and what attacks does it prevent? What scenarios does it capture?**
>
> As explained in Appendix C, plausible deniability captures scenarios where a model trainer has to defend themselves against claims of having used some illicit/copyrighted/sensitive data for training. The model trainer provides evidence (e.g., alternative batches) to refute the accusation. If such evidence exists, then the influence on the model parameters of whatever batches were actually used for training (whatever they are) *must* necessarily be limited (and thus leakage is bounded).
>
> We argue that this privacy framing is meaningful as it naturally maps onto important real-world scenarios. For example, there are several ongoing high profile copyright lawsuits where a model trainer is accused of having used prohibited data during training. Another scenario is that of incremental/online learning where new “incremental” data becomes available and is used by a model to update the model. One or more individuals in the incremental data could later decide to withdraw consent for their data and therefore the question of whether the same model could be obtained without the batch that contained this data becomes critical.
>
> In these scenarios (and others) there is at bottom a claim that some prohibited data was used during the training (in one or more batches). And the question is: how much did that influence the final model? We think plausible deniability, i.e., denial of batches, is a reasonable way to approach this and formalize such scenarios.
>
> Note that unlike DP, plausible deniability is a constructive notion in that the model trainer can (in principle) provide concrete evidence to refute the accusation. With DP there is no such evidence other than stating a DP training process was used. This means a model trainer could explicitly provide alternative batches for the training process. In a similar way as researchers proposed proof-of-learning to prove a model was obtained from some specific dataset [1], one could imagine building a “proof of deniability” using our framework.
>
> **3. Why is a batch notion meaningful?**
>
> A crucial point (not sufficiently emphasized in our paper; we will fix) is that the batch being denied (and the other batches) are *arbitrary* and can even be adversarially chosen (see Lemma 4). (We only assume that the seed batch is selected uniformly at random.) Therefore PD is also applicable in scenarios where the dataset was not split/sampled i.i.d. from the training distribution or for some reason a batch’s content is biased or composed of samples from a different distribution/dataset. (Note that in PD-SGD we use a shuffle-and-partition process which results in disjoint batches.)
>
> This allows us to model attack scenarios where the adversary claims that the model trainer (whether true or not) purposefully crammed prohibited/sensitive data in specific batches. We believe that only “batch notions” can capture such scenarios and DP notions (in particular) cannot because they consider privacy in terms of neighboring datasets and do not consider batching or ordering of data — but batching and ordering has a substantial impact on learning (e.g., see work on ordering attacks such as [2]). Furthermore, such scenarios are important to capture because there are practical reasons why data may not be distributed “i.i.d.” across batches such as online/incremental learning or curriculum learning [3].
>
>
> **4. Why does PD protect individual privacy?**
>
> Protection at the level of a batch leads to protecting individual privacy. This the case for plausible deniability and many other reasonable batch notions we could consider.
>
> We thank the reviewer for the suggestion of adapting Lemma 4 to capture individual leakage at a given step. This is interesting for future work. We point out that Lemma 4 already limits leakage for individuals consistent with the above argument. Because batches are disjoint, any individual data point appears in **at most one** batch per iteration. Consequently, the strongest evidence an attacker can ever observe is a single accepted update—yet that update is just as compatible with $T-1$ alternative batches that do not contain the record, leaving the attacker no firmer footing to confirm the record’s presence.
>
> An individual’s data may appear in different batches across different training iterations. We have left for future work the case of deriving strong composition results (e.g., following Lemma 4) since *our focus in this paper is mostly empirical*. However, Lemma 4 can already (naively) be composed, or one can use the relationship with DP to obtain and readily derive composition results.
>
>
> **5.  What is the threat model for PD-SGD? Why does PD-SGD protect individual privacy?**
>
> As we mentioned in section 5.1, we evaluate PD-SGD assuming a black-box membership adversary that knows the full algorithm and parameters, but only interacts with the final model; the training dynamics (noisy gradients, pass/fail decisions) remain hidden. This is consistent with related work and allows for fair comparison against existing empirical defenses.
>
> Intuitively, PD-SGD is designed to reject high-leakage anomalous updates from batches: an update is discarded if too few alternatives can explain it. This means that when an update is accepted, the likelihood-ratio advantage for the hypothesis that a particular batch was used is bounded (by a factor depending on $\gamma$ and $T$ as shown in Lemma 4). This caps the attacker’s evidentiary lift at each step.
>
> Since the data is partitioned into disjoint batches at every iteration, any additional data point will fall into exactly one batch, and any update from a batch cannot be uniquely attributed to any data point in that batch, which therefore limits the leakage of any data point at each step.
>
> Empirically, we show in Appendix F.12 that the most frequently rejected data points are exactly those that have high per-example MIA success rate. With standard SGD these “vulnerable” points have **91.67%** average success, which drops to **56.67%** under PD‑SGD. This is direct empirical evidence that PD‑SGD protects individuals most at risk.
>
> **6. Concrete parameter–tuning algorithm**
>
> We agree that tuning PD-SGD’s parameters can be complex, as it involves three interacting privacy parameters ($\sigma$, $\gamma$, and $T$). To address this challenge, our paper provides both theoretical guidance (Section 3.1, Appendix D) and empirical strategies (Appendix F).To further evaluate the effectiveness of our empirical strategy, we conducted additional experiments on an NVIDIA B200 GPU.
>
> As a baseline, we first performed a full grid search over all three privacy parameters ($\sigma$, $\gamma$, and $T$), covering 180 combinations in total. This exhaustive search required around 110.41 GPU hours to complete.
>
> We then applied our empirically guided two-phase strategy, which significantly reduces tuning cost:
>
> Phase 1 ($\sigma$ screening): We fix the threshold $T=1$ and run full training for each candidate $\sigma$. We retain only those $\sigma$ values that achieve high utility (final test accuracy $\geq$ 96%).
>
> Phase 2 (coarse filtering of $\gamma$ and $T$): For each surviving $\sigma$, we run shortened training (200 steps) across the $\gamma$ and $T$ grid. We discard any configuration that results in degenerate behavior (i.e., reject rate reaches 0% or 100%) early. The remaining viable combinations are then trained to convergence.
>
> We will provide full pseudo-code in the revised paper. We omit it here in the initial rebuttal due to character limit.
>
> This procedure reduces total tuning time from **110.41** GPU hours to just **4.15** GPU hours, while still identifying high-performing configurations. In practice, this makes PD-SGD much more efficient to tune than standard grid search. Notably, we used the full grid primarily to ensure fair comparison across baselines, not because PD-SGD requires it. We believe that the combination of theory-informed constraints, practical heuristics, and structured filtering makes PD-SGD both scalable and practical for deployment.
>
>
>
> Reference:
>
> [1] Jia, Hengrui, et al. "Proof-of-learning: Definitions and practice." 2021 IEEE Symposium on Security and Privacy (SP). IEEE, 2021.
>
> [2] Shumailov, Ilia, et al. "Manipulating sgd with data ordering attacks." Advances in Neural Information Processing Systems 34 (2021): 18021-18032.
>
> [3] Soviany, Petru, et al. "Curriculum learning: A survey." International Journal of Computer Vision 130.6 (2022): 1526-1565.

---

> ### Comment · Reviewer_is7x · 2025-08-05
>
> > Concrete parameter–tuning algorithm
>
> The proposed solution seems like a reasonable empirical way to tackle the problem.
>
> > What is a meaningful notion of privacy?
>
> I understand this wording on its own is ambiguous but what I meant by "a meaningful notion of privacy" is a concrete connection of the proposed notion of batch plausible deniability to relevant attack settings, as probed through the questions _'What are some realistic attack settings that are directly prevented by such guarantee? What actual adversarial scenarios can be modeled by "denying an inclusion of a batch"? How does it exactly map to privacy of individuals? The privacy of "batches" is certainly not a useful primary notion of privacy.'_ By this I do not mean differential privacy, as the response seems to suggest.
>
> Let me re-frame what I was looking for in a more technical way, so that there's less room for interpretation. Let's pick any survey of privacy threats, say [Salem et al., 2023](https://arxiv.org/abs/2212.10986). They list as such membership inference, attribute inference, record reconstruction, DP distinguishability (another kind of membership inference), and property inference. In their work, the success rates of these are well-defined through games. Let's say I have trained a model with PD-SGD with $\alpha$, $\sigma$, $\gamma$, and $T$. What can I then say about success rate of any of these threats? If not success rate, can we _plausibly deny_ membership of an example in general – as in the original paper on plausible deniability? It does not have to be specifically these threats, but a concrete connection to any other well-known attack setting or notion, e.g., re-identification, would do.
>
> > We argue that this privacy framing is meaningful as it naturally maps onto important real-world scenarios. For example, there are several ongoing high profile copyright lawsuits where a model trainer is accused of having used prohibited data during training. Another scenario is that of incremental/online learning where new “incremental” data becomes available and is used by a model to update the model.
>
> It does not seem to me that batch plausible deniability neatly matches the cited cases. For these ones, membership inference or denying membership of an example (not batch) seems like a more appropriate model. Although I do not think it maps to the cases cited above, I agree that batch plausible deniability is an interesting setting and it could be useful. However, it is non-standard and its relevance is still unclear. I believe it is fair to list denying an adversarially selected batch as one of the implications of satisfying the proposed notion, but without further justification, e.g., a case study, it is not sufficient for some of the strong non-empirical claims in the paper, as detailed in the original review.
>
> In sum, what I was seeking is a _concrete_ connection to well-understood privacy threats. Intuitively, it does seem that the proposed notion should be related to denying membership of a specific example, as the response suggests in the statement "protection at the level of a batch leads to protecting individual privacy." If it were possible to make this informal intuition concrete by bounding the success rate of any of the attacks listed above or other standard attacks studied in ML, using $\alpha$, $\sigma$, $\gamma$, and $T$ in general, this would completely resolve my issue.
>
> I will retain my score for now, because, as mentioned in the original review, the issue with the meaning of the theoretical notion is the most substantial (see the "overall, the proposed notion..." part of the review).

---

> > ### Author Response · Authors · 2025-08-07
> >
> > Thank you for engaging with our response. This clarifying follow-up and reframing is helpful.
> >
> > You are correct that we do seek to protect against concrete privacy threats to individuals, such as membership inference. But, we also believe PD is useful because it can model scenarios beyond membership inference that other concrete notions cannot and offer protection in those scenarios.
> >
> > **1.** So, our claim (which we will make much more explicit in the revised manuscript) is that batch plausible deniability is a sufficient condition to bound the advantage of membership inference attackers.
> >
> > Below we provide a series of games in the style of Salem et al. SoK that connects batch PD to (one-step) MIA.
> >
> > For all games below, the objective is to guess bit b. The advantage of the adversary is: Adv = 2 Pr{b’=b} - 1. The advantage of the best adversary for a game G is denoted Adv(G).
> >
> > Notation (consistent with the notation in the Salem et al. SoK when possible):
> > - A’ denotes the algorithm the adversary uses to choose batches and the target batch.
> > - A denotes the algorithm that outputs the adversary’s guess b’.
> > - $\mathcal{T}$ denotes *one step* of the training algorithm. This algorithm takes in a set of batches as inputs and outputs a gradient vector g.
> > - We assume there are at least $m > 2$ batches that are disjoint and of roughly equal length (with the exception of the last batch).
> > - $\mathcal{D}$ denotes the data distribution (used only in some games).
> >
> >
> > Game G0 — Batch PD
> >
> > 1: (B1, B2, …, Bm), Bt <- A’($\mathcal{T}$)	// adversary pick batches and the target batch
> >
> > 2: b ~ {0,1}					// sample random bit b
> >
> > 3: if b = 1 then:
> > 	g <- $\mathcal{T}$(B1,B2,...,Bm,Bt)	// gradient from $\mathcal{T}$ with Bt included
> >
> > 4: else:
> > 	g <- $\mathcal{T}$(B1,B2,...,Bm)	// gradient from $\mathcal{T}$ *without* Bt
> >
> > 5: b’ <- A(g, $\mathcal{T}$, B1, B2, …, Bm, Bt)
> >
> >
> >
> > Game G1 — Singleton-Batch PD
> >
> > 1: (B1, B2, …, Bm), z <- A’($\mathcal{T}$)	// adversary pick batches and target data point z
> >
> > 2a: Bt = {z}					// batch with only z in it
> >
> > 2b: b ~ {0,1}					// sample random bit b
> >
> > 3: if b = 1 then:
> > 	g <- $\mathcal{T}$(B1,B2,...,Bm,Bt)	// gradient from $\mathcal{T}$ with Bt included
> >
> > 4: else:
> > 	g <- $\mathcal{T}$(B1,B2,...,Bm)	// gradient from $\mathcal{T}$ *without* Bt
> >
> > 5: b’ <- A(g, $\mathcal{T}$, B1, B2, …, Bm, Bt)
> >
> > **Difference between G0 and G1**: In G1 the adversary does not get to pick a full target batch, but only a single data point.
> >
> > We have Adv(G0) $\geq$ Adv(G1), since any adversary that wins G1 with some advantage can play G0 and output the same z as Bt as it would when playing G1.
> >
> >
> >
> >
> > Game G2 — Chosen Data, Random Batches (similar to Strong Membership Inference)
> >
> > 1: S, z <- A’($\mathcal{T}$)	// adversary pick dataset S (|S| = n) and target data point z
> >
> > 2a: (B1, B2, …, Bm) <- Partition(S)	// randomly suffle and partition
> >
> > 2b: Bt = {z}				// batch with only z in it
> >
> > 2c: b ~ {0,1}				// sample random bit b
> >
> > 3: if b = 1 then:
> > 	g <- $\mathcal{T}$(B1,B2,...,Bm,Bt)
> > 4: else:
> > 	g <- $\mathcal{T}$(B1,B2,...,Bm)
> >
> > 5: b’ <- A(g, $\mathcal{T}$, S, Bt)
> >
> >
> > **Difference between G1 and G2**: In G2 the adversary picks the dataset S but does not determine the partitioning into batches (or learn it).
> >
> > We have: Adv(G1) $\geq$ Adv(G2). If adversarial partitioning provides any benefit then adversaries for G1 can use that but those for G2 cannot.
> >
> >
> > Game G3 — Average Membership Inference
> >
> > 1a: S ~ $\mathcal{D}^{n}$	// sample n data points i.i.d. from the data distribution
> >
> > 1b: z <- A’($\mathcal{T}$)	// adversary picks target data point z
> >
> > 2a: (B1, B2, …, Bm) <- Partition(S)	// randomly suffle and partition
> >
> > 2b: Bt = {z}				// batch with only z in it
> >
> > 2c: b ~ {0,1}
> >
> > 3: if b = 1 then:
> > 	g <- $\mathcal{T}$(B1,B2,...,Bm,Bt)
> >
> > 4: else:
> > 	g <- $\mathcal{T}$(B1,B2,...,Bm)
> >
> > 5: b’ <- A(g, $\mathcal{T}$, Bt)
> >
> >
> > **Difference between G2 and G3**: in G3 the adversary picks only the target data z, not the dataset S (which is sampled randomly).
> >
> > We have: Adv(G2) $\geq$ Adv(G3). We conclude: Adv(G0) $\geq$ Adv(G3). A one-step algorithm $\mathcal{T}$ that achieves batch PD (i.e., no adversary has an advantage greater than some $\lambda>0$) also bounds (one-step average) membership inference.
> >
> >
> >
> > Differences with games in Salem et al. SoK, which: (1) are not one step. (2) do not model batches and partitioning. (3) operate in the “replace-one” setting (i.e., adversary chooses z0, z1 but zb is included; e.g., Game10 in the SoK) not in the “add-remove” setting.
> >
> > Differences (1) and (2) are a consequence of PD being a batch notion. We need the games to explicitly model the batches. We do not believe (3) is a consequential difference. It is well-known that the “replace-one” and “add-remove” settings can be related (usually a constant factor is the difference between the two). To be consistent with the results we give in the paper we present games in the “add-remove” setting.

---

> > > ### Author Response · Authors · 2025-08-07
> > >
> > > **2.** Beyond PD as a privacy notion and its relation to other notions, the properties of the PD-SGD algorithms are important and very much the focus of our paper. In particular, please note that most of our theoretical claims in the paper are about those properties (of PD-SGD).
> > >
> > > This is worth emphasizing because as we explicate in Appendix C.5, PD-SGD is connected to DP and through the connection (in combination with composition) PD-SGD immediately inherits resilience to membership inference attackers for a full training run. This is because DP is a “stronger notion” than resilience to membership inference (e.g., See Yeom et al. [1] or Theorem 4 of the Salem et al. SoK). We provide the full derivation of MIA advantage bounds in terms of PD-SGD privacy parameters below.
> > >
> > >
> > > Lemma 7 in our paper shows that PD-SGD algorithm satisfies $(\epsilon, \delta)$-differential privacy with:
> > >     $$\\epsilon = \\ln\\left(1 + \\frac{\\alpha\\beta}{t}\\right)$$
> > >     $$\\delta \\le \\frac{1}{m}(1-\\psi)e^{-\\epsilon_{0}(T-t)}, \\quad \\text{where} \\quad \\epsilon_{0} = \\ln\\beta, \\quad 1\le  t < T$$
> > >
> > > From Proposition 2 in the Salem et al. SoK, the advantage of a Differential Privacy Distinguishability (DPD) adversary for any $(\epsilon, \delta)$-differentially private algorithm as:
> > >     $$Adv_{DPD}(\\mathcal{A}) \\le \\frac{e^\\epsilon - 1 + 2\\delta}{e^\\epsilon + 1}$$
> > >
> > > Resilience against DPD implies resilience against Membership Inference (MI). This allows us to use the bound on the DPD advantage as an upper bound for the MI advantage. Therefore:
> > >
> > > $$Adv_{MI} \\le Adv_{DPD}$$
> > >
> > > $$Adv_{MI} \\le \\frac{e^\\epsilon - 1 + 2\\delta}{e^\\epsilon + 1}$$
> > >
> > > We substitute the expressions for $\\epsilon$ and $\\delta$ from Lemma 7 to obtain:
> > >
> > > $$e^\\epsilon = e^{\\ln(1 + \\frac{\\alpha\\beta}{t})} = 1 + \\frac{\\alpha\\beta}{t}$$
> > >
> > >  $$Adv_{MI} \\le \\frac{(1 + \\frac{\\alpha\\beta}{t}) - 1 + 2\\delta}{1 + \\frac{\\alpha\\beta}{t} + 1} = \\frac{\\frac{\\alpha\\beta}{t} + 2\\delta}{2 + \\frac{\\alpha\\beta}{t}}$$
> > >
> > > * Substituting the upper bound for $\\delta$:
> > >         $$Adv_{MI} \\le \\frac{\\frac{\\alpha\\beta}{t} + 2 \\left( \\frac{1-\\psi}{m}e^{-\\epsilon_{0}(T-t)} \\right)}{2 + \\frac{\\alpha\\beta}{t}}$$
> > >
> > > * Since $\\epsilon_0 = \\ln\\beta$, we can write $e^{-\\epsilon_{0}(T-t)} = e^{-(\\ln\\beta)(T-t)} = (\\beta)^{-(T-t)} = \\beta^{t-T}$.
> > >
> > > $$Adv_{MI} \\le \\frac{\\frac{\\alpha\\beta}{t} + \\frac{2(1-\\psi)}{m}\\beta^{t-T}}{2 + \\frac{\\alpha\\beta}{t}}$$
> > > For **N** independent iterations, the overall privacy budget $(\epsilon', \delta')$ is given by the advanced composition theorem as we stated in the end to appendix C.5.
> > >             $$
> > >         \\epsilon' = \\sqrt{2N \\ln(\\frac{1}{\\delta''})\\ln\\left(1 + \\frac{\\alpha\\beta}{t}\\right) + N\\ln\\left(1 + \\frac{\\alpha\\beta}{t}\\right)\\left(\\frac{\\alpha\\beta}{t}\\right)}
> > >         $$
> > >         $$
> > >         \\delta' = N\\left(\\frac{1-\\psi}{m}\\beta^{t-T}\\right) + \\delta''
> > >         $$
> > >     $$
> > >     Adv_{MI} \\le \\frac{e^{\\epsilon'} - 1 + 2\\delta'}{e^{\\epsilon'} + 1}
> > >     $$
> > >
> > >
> > > Thank you again for engaging with us, we sincerely value your feedback and appreciate the opportunity to clarify aspects of our paper that we acknowledge were unclear/confusing in the presentation. Please let us know if this clarifies things for you.
> > >
> > >
> > > [1] Yeom et al. "Privacy risk in machine learning: Analyzing the connection to overfitting." 2018 IEEE 31st computer security foundations symposium (CSF). IEEE, 2018.

---

> > > > ### Comment · Reviewer_is7x · 2025-08-07
> > > >
> > > > > A one-step algorithm that achieves batch PD also bounds (one-step average) membership inference.
> > > >
> > > > Thank you, this exactly answers my concerns. I am guessing that the fact that batch PD provides individual protection was obvious to the authors, but was not at all obvious to me as a reader. The sequence of games shows this very well.
> > > >
> > > >
> > > > > We do not believe (3) is a consequential difference.
> > > >
> > > > Agreed.
> > > >
> > > > >  In Appendix C.5, PD-SGD is connected to DP and [...] inherits resilience to membership inference attackers
> > > >
> > > > This reduction is clear, but I understand that this is relevant only to a very special instantiation?
> > > >
> > > > ___
> > > >
> > > > I am increasing my score. I strongly recommend the authors to prominently showcase the bound on MIA advantage as one of the core implications of the proposed notion. You can ignore my previous suggestion ("the paper should either drop the claims of theoretical privacy guarantees (e.g., in conclusions)", as there are theoretical privacy guarantees on a standard notion of privacy (even though only for one-step algorithms).

---

> > > > > ### Author Response · Authors · 2025-08-07
> > > > >
> > > > > We sincerely appreciate your feedback. We are glad we are able to clarify this. Thank you for engaging with us and making our paper better! We will be sure to provide details of games and reduction to MIA advantage and emphasize it as a core part of our framework in our revised paper.
> > > > >
> > > > > >This reduction is clear, but I understand that this is relevant only to a very special instantiation?
> > > > >
> > > > > The result given in our paper does assume randomization of the threshold with ceiling and a sensitivity 1 test. We feel these are rather “mild” restrictions in the sense that different noise distributions can be used (e.g., Laplace or Gaussian instead of Geometric), and that tests with bounded (but larger than 1) sensitivity are also covered at the cost of loosening the bound or adjusting the privacy parameters (i.e., increasing T).
> > > > >
> > > > > Thank you for bringing this up. We will discuss this in more detail in the revised paper.

---

### Note · Authors · 2025-08-12

We would like to thank the reviewers and AC for helping us improve our paper! We are sincerely grateful to the constructive feedback and engagement throughout the review process. Discussions with reviewers have highlighted several areas where our initial presentation of the work, particularly of the framework and its implication for individual privacy, could be substantially improved. These discussions also allowed us to satisfactorily address several reviewers’ concerns regarding theoretical aspects of our work and formal relationship with existing notions of privacy.

We would like to assure the reviewers that we took their feedback to heart. We will thoroughly revise our paper to incorporate feedback from the reviews and subsequent discussion to ensure that our paper communicates its contribution and privacy claims effectively. In particular, we will incorporate the formal privacy games, explicit connection between plausible deniability, and membership inference advantage bounds. We will also include additional discussion of computational complexity, measurements of memory usage, empirical evaluation against HAMP, and experiments demonstrating scalability of PD-SGD to LLM fine-tuning.

Finally, we feel it is worth emphasizing the novelty of our work: we introduce a new framework for privacy-preserving machine learning in the form of a new privacy notion and an algorithm for it, which is notably distinct from existing approaches. We believe that research on new privacy notions and approaches is important and timely.

---

### Decision · Program_Chairs · 2025-09-17

**Decision:**

Accept (poster)

**Comment:**

This paper received extensive discussion, which covered substantial additional theoretical explanation for and justification of the authors' proposed framework, as well as additional experimental results. Two of the reviewers were ultimately very positive about the paper, citing the clarity of the game-based formulations that the authors propose (among other things).

There is always a danger of accepting a paper based on revisions that have not yet been completed. However, given the energy the authors put into the rebuttal process, I believe they will take the time to incorporate the revisions into the final paper. I also think it is worth giving new conceptual frameworks a chance to be looked at and critqued by the community, even if they are still rough around the edges.

I would encourage the authors to take reviews and subsequent reviewer comments into account carefully, since they reflect the understanding that well-intentioned readers are likely to draw from the paper. In particular, I think the formulation of batch-level guarantees is very confusing, since the authors' proposed algorithm forms batches randomly. I also think that formulating guarantees only at the level of individual iterations is insufficient. DP-SGD works this way since composition theorems for differential privacy (DP) allow one to lift such guarantees to the level of the algorithm as a whole. This type of lifting appears absent here (except in Appendix C which is about DP).